# Theoretical Hardness and Tractability of POMDPs in RL with Partial Online State Information

## Abstract

Partially observable Markov decision processes (POMDPs) have been widely applied to capture many real-world applications. However, existing theoretical results have shown that learning in general POMDPs could be intractable, where the main challenge lies in the lack of latent state information. A key fundamental question here is how much online state information (OSI) is sufficient to achieve tractability. In this paper, we establish a lower bound that reveals a surprising hardness result: unless we have full OSI, we need an exponentially scaling sample complexity to obtain an $\epsilon$-optimal policy solution for POMDPs. Nonetheless, inspired by the key insights in our lower bound design, we find that there exist important tractable classes of POMDPs even with only partial OSI. In particular, for two novel classes of POMDPs with partial OSI, we provide new algorithms that are proved to be near-optimal by establishing new regret upper and lower bounds.

## 1 Introduction

Partially observable Markov decision processes (POMDPs) model reinforcement learning (RL) systems, where an agent interacts with the environment sequentially without observing the latent state. In these systems, the agent only has access to a noisy observation randomly generated by the latent state via an emission probability distribution. The goal of the agent is to achieve a large expected cumulative reward. POMDPs generalize the classic (fully observable) MDPs, and have been applied to capture many real-world applications. For example, an AI-trained robot often receives only noisy observations of the environment from its sensors due to sensory noise (Akkaya et al., 2019); autonomous cars typically do not have a global view of traffic conditions due to their limited reception (Levinson et al., 2011). Similar scenarios can occur in games (Berner et al., 2019), healthcare (Hauskrecht & Fraser, 2000), recommendation systems (Li et al., 2010), economic systems (Zheng et al., 2020), and so forth.

Existing information-theoretic results have shown that learning in general POMDPs is intractable and PSPACE-complete (Papadimitriou & Tsitsiklis, 1987; Mundhenk et al., 2000; Vlassis et al., 2012; Krishnamurthy et al., 2016). This is in contrast to classic MDPs, where many efficient algorithms have been developed, e.g., Azar et al. (2017); Jin et al. (2018); Agarwal et al. (2019); Jin et al. (2020); Ayoub et al. (2020); Xie et al. (2020); Foster et al. (2021); Jin et al. (2022); Bai et al. (2019); Cai et al. (2020), among others. The challenge of POMDPs mainly lies in the lack of latent state information, such that the Markov property that simplifies classic MDPs does not hold any more.

Despite the intractability in general POMDPs, recent studies have identified some **tractable classes** of POMDPs, for which efficient algorithms with polynomial dependency (on the number of actions $A$, number of states $S$ and episode length $H$) can be developed, e.g., $m$-step decodable POMDPs (Efroni et al., 2022), reactive POMDPs (Jiang et al., 2017), POMDPs with block MDPs (Zhang et al., 2022) or latent MDPs (Kwon et al., 2021), and POMDPs with reachability (Xiong et al., 2022) or observability (Golowich et al., 2022). Due to page limits, we relegate more discussions about related work in Appendix A. One prominent tractable class is identified based on weakly revealing conditions (Liu et al., 2022; 2023) or predictive state representations (Chen et al., 2022a; Zhong et al., 2022). However, these conditions may not hold in practical cases, e.g., resource allocation (Sinclair et al., 2023; Lee et al., 2023) and robotics (Pinto et al., 2018; Lee et al., 2023).

Moreover, the regret obtained there can be arbitrarily large if the emission probability differences of different underlying states are small.

To circumvent the dependency and strong assumptions on the emission probability measure, recent work has exploited hindsight state information (Sinclair et al., 2023; Lee et al., 2023), where full state information is revealed only *at the end of each episode*. This line of work is motivated by the fact that, although the precise information about the true underlying state is not available before the agent takes an action, some information may become available in hindsight. However, these studies have assumed *full* hindsight state information. Thus, a natural question one may ask is: what would happen if the state information was *not fully* revealed *at the end* of the episode? In fact, this can happen often in practice. For example, in classic wireless channel scheduling formulated by POMDPs (Zhao et al., 2007; Chen et al., 2008; Ouyang et al., 2015), only the feedback about the scheduled or sensed channels will be available to the users; in autonomous driving Levinson et al. (2011); Pinto et al. (2018); Jennings & Figliozzi (2019), only the condition of the located or probed path will be known to the car. Further, it can be trivially shown (based on the existing lower bounds in Krishnamurthy et al. (2016); Liu et al. (2022)) that such a situation becomes intractable.

This thus motivates us to investigate the value of *partial (i.e., not full) state information *inside (i.e., not at the end of)* the episode. We call this **partial "Online State Information" (OSI)**. In order to model such partial OSI more concretely, we provide a novel formulation. Specifically, we consider vector-structured states Jin et al. (2020); Agarwal et al. (2019); Ayoub et al. (2020), which are motivated by the aforementioned practical examples. In other words, the state is given by a $d$-dimension vector with each element representing an abstract feature, such as the feedback about a wireless channel Zhao et al. (2007) and the condition of a path in autonomous driving Jennings & Figliozzi (2019). Partial OSI means that at each step of an episode, a subset of $\tilde{d}$ ($1 \le \tilde{d} < d$) elements in the state-vector will be revealed to the agent *after* her query. Note that such a model allows the agent to *actively query* partial OSI for different elements at different times. This prevents the trivial case, where one state-element cannot be known throughout the process (so that the problem becomes equivalent to a POMDP problem with that specific unknown state-element being the hidden state). Therefore, the key fundamental open questions are:

> **With such partial OSI, can POMDPs be tractable/learnable? If not, are there any specific classes of POMDPs that can be tractable under partial OSI?**

**Our Contributions:** In this paper, we study the important problem of POMDPs with partial OSI and provide in-depth answers to the above key open questions.

**First,** we establish a **lower bound** in Theorem 1 that reveals a *surprising hardness* result: unless we have full OSI, we need an exponentially scaling sample complexity of $\tilde{\Omega}(\frac{A^H}{\epsilon^2})$ to find an $\epsilon$-optimal policy for POMDPs, where $A$ and $H$ are the number of actions and episode length, respectively. This result indicates a sharp gap between POMDPs with *partial* OSI and those with *full* OSI or *full* hindsight state information (Lee et al., 2023). This may seem somewhat counter-intuitive, because by combining multiple partial OSI from different steps, one may construct full information of a state, and thus enjoy similar performance as that with full OSI. In fact, in Sec. 3, we design a hard instance with special state representations and transitions, under which partial OSI at each step and even a combination of partial OSI from different steps are not sufficient to achieve an $\epsilon$-optimal solution with polynomial complexity.

Nonetheless, inspired by the key insights in our design of the hard instance for establishing the lower bound, we identify two intriguing **tractable classes** of POMDPs with only partial OSI.

**Second,** inspired by our state-transition design for the lower bound, in Sec. 4 we identify a novel tractable class of POMDPs with partial OSI, where the transitions of the sub-states (i.e., elements) in the state-vector are independent of each other. This class is motivated by many practical examples ranging from wireless scheduling (Zhao et al., 2007; Chen et al., 2008; Ouyang et al., 2015) to Martian rock-sampling (Levinson et al., 2011) and autonomous driving (Pinto et al., 2018; Jennings & Figliozzi, 2019). We provide two new **near-optimal algorithms** for this class. The regrets of both algorithms achieve a polynomial dependency on all parameters (please see Theorem 2 and Theorem 6). In addition, the regret of our second algorithm for the case with $\tilde{d} > 1$ shows that the regret can be *further reduced* as $\tilde{d}$ increases. To achieve such results, our algorithm design includes important novel ideas to determine (i) which partial OSI is more informative, and (ii) the

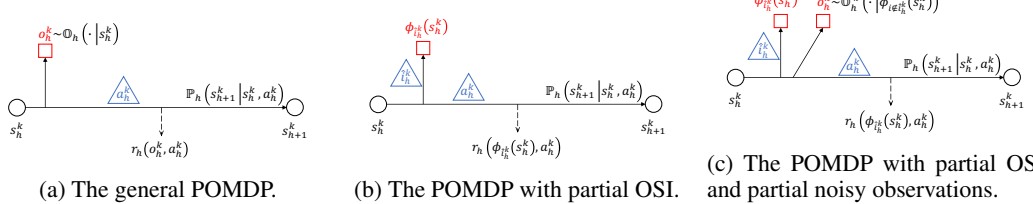

Figure 1: A sketch of one step in different classes of POMDPs: the squares represent the feedback of the partial OSI actively queried by the agent, and/or the (partial) noisy observations

action policy that relies on the queried partial OSI at each step. These also require new technical developments in the regret analysis (see Appendix E and Appendix F).

**Third,** inspired by our state-representation design for the lower bound, in Sec. 5 we identify another novel tractable class of POMDPs with partial OSI, where additional noisy observations for the sub-states in the state-vector that are not actively queried are available. We provide a new algorithm with a **near-optimal regret** in Theorem 3. Our regret analysis involves a non-trivial generalization of the observable operator method (Jaeger, 2000; Liu et al., 2022) to handle the case with partial OSI of different sub-states that are actively queried by the agent. In addition, we provide a new regret *lower-bound* in Theorem 4 that demonstrates the near-optimality of the regret that we achieve.

## 2 PROBLEM FORMULATION

In this section, we first introduce the general episodic partially observable Markov decision process (POMDP) for clarity, which is intractable in the worst case. Then, we introduce the POMDP setting with partial online state information (OSI) that we study in this paper.

### 2.1 THE GENERAL EPISODIC POMDP

Episodic POMDPs are usually modelled by a tuple $\mathcal{M} = (\mathcal{S}, \mathcal{A}, \mathcal{O}, H, \Delta_1, \mathbb{P}, \mathbb{O}, r)$ (Liu et al., 2022; 2023; Chen et al., 2022a;b; Cai et al., 2022), where $\mathcal{S}$, $\mathcal{A}$ and $\mathcal{O}$ denote the state space with $S$ states, the action space with $A$ actions and the observation space with $O$ observations, respectively; $H$ denotes the number of steps in an episode; $\Delta_1 : \mathcal{S} \to [0, 1]$ denotes a probability measure supported on the state space $\mathcal{S}$ and determines the randomness of the initial state at the beginning of an episode; $\mathbb{P} = \{\mathbb{P}_h : \mathcal{S} \times \mathcal{S} \times \mathcal{A} \to [0, 1]\}_{h=1}^{H-1}$ and $\mathbb{O} = \{\mathbb{O}_h : \mathcal{O} \times \mathcal{S} \to [0, 1]\}_{h=1}^{H}$ denote the *unknown* transition and emission probability measures, respectively; and $r = \{r_h : \mathcal{O} \times \mathcal{A} \to [0, 1]\}_{h=1}^{H}$ denotes the known reward function. Specifically, an online agent interacts with the environment in $K$ episodes. At each step $h = 1, ..., H$ of an episode, the agent receives a noisy observation $o_h^k$ that is generated according to the emission probability $\mathbb{O}_h(\cdot|s_h^k)$, where $s_h^k$ is the *unknown* true latent state. Next, the agent takes an action $a_h^k$ and receives the reward $r_h(o_h^k, a_h^k)$. Then, the environment transits to the next state $s_{h+1}^k$, which is drawn according to the transition probability $\mathbb{P}_h(\cdot|s_h^k, a_h^k)$. The goal of the agent is to find a near-optimal policy that achieves an expected cumulative reward close to that of the optimal policy. Please see Fig. 1a for a sketch of one step. Due to the lack of latent state information, the observation is non-Markovian and the policy needs to maintain memory.

### 2.2 THE EPISODIC POMDP WITH PARTIAL OSI

As discussed in Sec. 1, we make the first effort to investigate the impact of partial OSI on POMDPs in this paper. We provide a formulation for studying POMDPs with partial OSI. Specifically, we consider the vector-structured states (Jin et al., 2020; Ayoub et al., 2020; Agarwal et al., 2019). Each state $s$ is represented by a $d$-dimension feature vector $\vec{\phi}(s) = [\phi_1(s), ..., \phi_d(s)]^{\mathrm{T}} \in \tilde{\mathbb{S}}^d$, where $\tilde{\mathbb{S}}$ is the universal set of the values for each element/sub-state in $\vec{\phi}(s)$, and $[\cdot]^{\mathrm{T}}$ denotes the transpose of a vector. We use $|\tilde{\mathbb{S}}|$ to denote the cardinality of the set $\tilde{\mathbb{S}}$. Then, at each step $h = 1, ..., H$ of an episode $k = 1, ..., K$, the agent interacts with the environment as follows (please see Fig. 1b for a sketch of one step of the POMDP with partial OSI):

(Step-i) The agent actively queries a subset of $\tilde{d}$ (where $1 \leq \tilde{d} < d$) sub-states (let $\hat{i}_h^k$ denote the indices of these queried sub-states); (Step-ii) the partial OSI, i.e., the precise information of the

queried sub-states $\{\phi_i(s)\}_{\{i \in \hat{i}_h^k\}}$, is revealed to the agent; (Step-iii) the agent takes an action $a_h^k$ and receives the reward $r_h(\phi_{\hat{i}_h^k}(s_h^k), a_h^k)$, where the reward $r_h : \tilde{S} \times \mathcal{A} \to [0,1]$ is a function of the partial OSI and $\tilde{S} \triangleq \{\phi_{\hat{i}}(s) : |\hat{i}| = \tilde{d}, s \in \mathcal{S}\}$ is the sub-state space for any union of $\tilde{d}$ sub-states.; (Step-iv) the environment transits to the next state $s_{h+1}^k$.

This model is motivated by various practical scenarios, e.g., wireless scheduling (Chen et al., 2008; Ouyang et al., 2015), autonomous driving (Levinson et al., 2011; Pinto et al., 2018; Jennings & Figliozzi, 2019), robotics (Akkaya et al., 2019; Lee et al., 2023; Silver & Veness, 2010) and healthcare (Hauskrecht & Fraser, 2000). Below, we elaborate on two important motivating examples.

**Motivating example 1:** In an autonomous delivery system (Jennings & Figliozzi, 2019), in order to deliver the product to the destination, a robot explores multiple paths and chooses one path at each intersection. Here, each sub-state $\phi_i(s)$ of $s$ represents the condition, e.g., traffic intensity, of one path. At each step, the robot agent first actively queries and observes the condition of several paths, i.e., the partial OSI. However, due to delay requirements, unknown dynamics in the environment, and occlusion, the precise conditions of other paths may not be available to the robot. Then, she chooses one path to follow, i.e., the action that will incur a reward.

**Motivating example 2:** Consider a cognitive MAC (medium access control) system (Ouyang et al., 2015), where a secondary user, i.e., an agent, wishes to search for spectrum-access opportunities. Here, the state $s$ characterizes the conditions of multiple channels available for an agent to use. Substate $\phi_i(s)$ represents the condition, e.g., busy or idle, of the $i$-th channel. At each step, the agent first probes the conditions of a number of channels. After this query, the conditions of the sensed channels will be observed, i.e., the partial OSI. However, due to energy constraints and latency requirements, the agent cannot sense all the channels. Then, she transfers the packets using one channel, i.e., the action that will incur a reward.

## 2.3 PERFORMANCE METRIC

In POMDPs with partial OSI, at each step $h$ of episode $k$, the feedback revealed to the agent is $\Phi_h^k = (\phi_{\hat{i}_1^k}(s_1^k), a_1^k, ..., \phi_{\hat{i}_{h-1}^k}(s_{h-1}^k), a_{h-1}^k)$. We use $\hat{\Phi}_h$ to denote the feedback space of $\Phi_h^k$ *before* the partial OSI for step $h$ is revealed, and use $\tilde{\Phi}_h = \{\hat{\Phi}_h \cup \{\phi_{\hat{i}_h}(s_h)\}_{\hat{i}_h}\}$ to denote the feedback space *after* the partial OSI for step $h$ has been revealed. Then, the query $\hat{i}_h^k$ is made according to a **query policy** $\pi_{q,h}^k \in \{\pi_{q,h} : \hat{\Phi}_h \to \hat{\Delta}_h(\{\hat{i}\}|\tilde{d})\}$, which maps from $\hat{\Phi}_h$ to a conditional probability measure $\hat{\Delta}_h(\{\hat{i}\}|\tilde{d})$ supported on the query space $\{\hat{i} : |\hat{i}| = \tilde{d}\}$. Next, **after** receiving the partial OSI $\phi_{\hat{i}_h^k}(s_h^k)$, the action $a_h^k$ is taken according to an **action policy**[1] $\pi_{a,h}^k \in \{\pi_{a,h} : \tilde{\Phi}_h \to \tilde{\Delta}_h(\mathcal{A})\}$, which maps from $\tilde{\Phi}_h$ to a probability measure $\tilde{\Delta}_h(\mathcal{A})$ supported on the action space $\mathcal{A}$. We use the $V$-value $V^{\pi^k} \triangleq \mathbb{E}_{\{\pi_q^k, \pi_a^k, \mathbb{P}, \Delta_1\}}[\sum_{h=1}^H r_h(\phi_{\hat{i}_h^k}(s_h^k), a_h^k)]$ to denote the expected total reward in episode $k$ by following $\pi_q^k = \{\pi_{q,h}^k\}_{h=1}^H$ and $\pi_a^k = \{\pi_{a,h}^k\}_{h=1}^H$, where $\pi^k = (\pi_q^k, \pi_a^k)$. We take the regret as the performance metric, which is the difference between the expected cumulative reward using the online joint policies $\pi^{1:K}$ and that of using the optimal policy, i.e.,

$$Reg^{\pi^{1:K}}(K) \triangleq \sum_{k=1}^K \left[ V^* - V^{\pi^k} \right], \tag{1}$$

where $V^* \triangleq \sup_\pi V^\pi$ denotes the expected total reward of the optimal policy in an episode. The goal of the online agent is to find a policy that achieves a sub-linear regret with respect to $K$. Hence, the main challenge and new difficulty here is how to design the query policy $\pi_q$, such that an action policy $\pi_a$ can also be intelligently developed to achieve a near-optimal regret.

## 3 PERILS OF NOT HAVING FULL OSI: A NEW LOWER BOUND

In this section, we answer the long-standing open question: whether POMDPs with online state information are tractable without *full* OSI? In Theorem 1 below, we establish a lower bound that reveals a *surprising hardness result*: unless we have full OSI, we need an exponential sample complexity to find an $\epsilon$-optimal policy for POMDPs, where a policy $\pi$ is $\epsilon$-optimal if $V^\pi \geq V^* - \epsilon$.

---

[1]Recall that $\phi_{\hat{i}_h^k}(s_h^k) \in \tilde{\Phi}_h$. Thus, the action policy $\pi_{a,h}^k$ relies on the output of the query policy $\pi_{q,h}^k$.

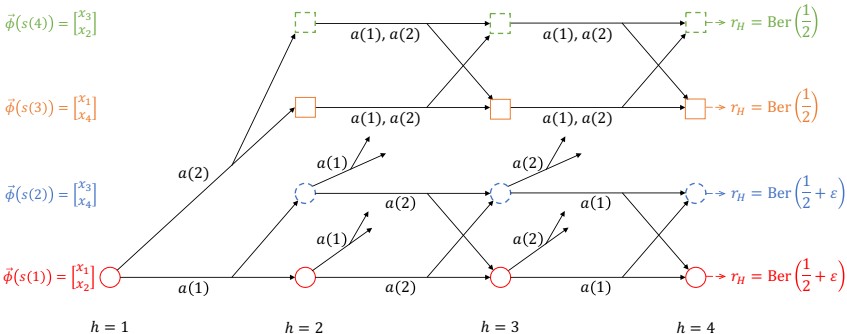

Figure 2: A hard instance for developing the lower bound in POMDPs with only partial OSI. States $s(1)$, $s(2)$, $s(3)$ and $s(4)$ are represented by solid circles, dashed circles, solid squares and dashed squares, respectively. $\text{Ber}(1/2)$ represents the Bernoulli distribution with mean $1/2$

**Theorem 1.** *(Intractability for not having full OSI) For POMDPs with only partial online state information introduced in Sec. 2.2, there exists hard instances, such that with a probability $p \geq 1/3$, any algorithm needs at least $\Omega(A^H/\epsilon^2)$ samples to find an $\epsilon$-optimal policy.*

Theorem 1 demonstrates the hardness of POMDPs without full OSI: a polynomially scaling sample complexity $\text{Poly}(A, H, S, K)$ is impossible. The result in Theorem 1 may seem counter-intuitive, because by combining multiple partial OSI collected from different steps, one may construct full observations and then enjoy similar performance as that with full OSI. Below, we design an important hard instance and provide our key proof ideas of Theorem 1, which shows why this is not true.

**Remark 1.** *The intractability result in Theorem 1 still holds even if in addition to partial OSI, there exist noisy observations (please see our discussion in Sec. 5). This is because we can construct a hard instance directly based on the one that we construct in this section, while letting the emission probabilities of the additional noisy observations to be exactly the same for all underlying states, such that the additional observations do not provide any useful statistical information.*

### 3.1 Our Key Proof Ideas for Theorem 1

For simplicity, we focus on the simpler case with $d = 2$ and $\tilde{d} = 1$, which makes it easier to understand our key proof ideas. Please see Appendix C for the complete proof. The important parts in our proof are to design special state representations and transitions, such that partial OSI cannot help the learner to improve her statistical knowledge about the true underlying state. Towards this end, we construct a hard instance with four states, i.e., $s(1)$, $s(2)$, $s(3)$ and $s(4)$ (see Fig. 2).

**Idea I (Special state representations):** Our first key idea is to construct special state representations, such that by only observing $\tilde{d} = 1$ sub-state, it is still impossible for the learner to infer the true latent state. Specifically, we let $\vec{\phi}(s(1)) = [x_1, x_2]^\text{T}$, $\vec{\phi}(s(2)) = [x_3, x_4]^\text{T}$, $\vec{\phi}(s(3)) = [x_1, x_4]^\text{T}$ and $\vec{\phi}(s(4)) = [x_3, x_2]^\text{T}$, where $x_1, ..., x_4$ are sub-states (see Fig. 2).

We introduce the high-level reason for constructing the state representations in this way. Let us consider states $s(1)$ and $s(2)$ as a group of states, and we call it group $a$. Similarly, we call states $s(3)$ and $s(4)$ group $b$. Under our construction of the state representation, each state in group $a$ (i.e., $s(1)$ and $s(2)$) must contain a same sub-state as that in each state of group $b$ (i.e., $s(3)$ and $s(4)$). For example, the first sub-states of both state $s(1)$ and state $s(3)$ are $x_1$. This means that, by only querying $\phi_1(s) = x_1$, the learner cannot know whether she is in a state from group $a$ or group $b$. As another example, the second sub-states of both state $s(1)$ and state $s(4)$ are $x_2$. This means that, by only querying $\phi_2(s) = x_2$, the learner cannot know whether she is in a state from group $a$ or group $b$. As a result, if (i) there is only one specific action sequence that guarantees the learner to be in group $a$, and (ii) group $a$ generates a larger reward, then intuitively the learner has to constantly keep trying all exponential number of possible action sequences to figure this out with high probability.

However, as we mentioned before, another question still remains: *whether a combination of the partial OSI from different steps would be enough?* To answer this question, we construct special state transitions using our idea II below. Together with the state representation that we construct above, this state transition causes difficulty for the learner, even when multiple partial OSI are combined.

**Idea II (Special state transitions):** Our second key idea is to construct special state transitions, such that *even by combining the partial OSI from different steps, it is still impossible for the learner to infer the true latent state.* Specifically, in each episode, the learner starts from state $s_1 = s(1)$ (see Fig. 2). At step $h = 1$, (i) if action $a(1)$ is chosen, the state will transition to $s(1)$ and $s(2)$ with the same probability (wsp); (ii) if action $a(2)$ is chosen, the state will transition to $s(3)$ and $s(4)$ wsp. At step $h = 2$, (i) if action $a(1)$ is chosen, both states $s(1)$ and $s(2)$ will transition to $s(3)$ and $s(4)$ wsp; (ii) if action $a(2)$ is chosen, they will transition to $s(1)$ and $s(2)$ wsp. At step $h = 3$, (i) if action $a(1)$ is chosen, states $s(1)$ and $s(2)$ will transition to $s(1)$ and $s(2)$ wsp; (ii) if action $a(2)$ is chosen, they will transition to $s(3)$ and $s(4)$ wsp. For states $s(3)$ and $s(4)$ at step $h = 2$ and $h = 3$, no matter which action is chosen, the states will transition to $s(3)$ and $s(4)$ wsp.

Then, together with the state representation that we constructed, even when the partial OSI about the first and second sub-states from different steps are combined, such a construction for the state transition still prevents the learner from knowing which group of states she is in. For example, at step $h = 1$ of two episodes, the learner can keep taking action $a(1)$ and query the first and second sub-states one-by-one. Then, the partial OSI at step $h = 2$ could be $\phi_1(s_2^k) = x_1$ (i.e., the first sub-state of $s(1)$) and $\phi_2(s_2^{k+1}) = x_4$ (i.e., the second sub-state of $s(2)$). However, note that the first and second sub-states of $s(3)$ are also $x_1$ and $x_4$. Thus, such a combination of partial OSI (i.e., $\phi_1(s_2^k) = x_1$ and $\phi_2(s_2^{k+1}) = x_4$) is not powerful enough for the learner to distinguish whether she is visiting $s(1)$ and $s(2)$ or she is simply visiting $s(3)$. Similar issues occurs at other steps.

**Idea III (Special reward functions):** Up to here, we can see that only with partial OSI, the learner cannot improve her statistical knowledge about the true underlying states. Thus, she can only rely on the statistical relation between the sequence of actions that is chosen and the reward that is received. Hence, to create difficulties, we let (i) the rewards $r_h$ at steps $h = 1, 2, 3$ are all 0; (ii) if the final state is in group $b$, i.e., $s(3)$ or $s(4)$, the reward at step $h = 4$ follows Bernoulli distribution with mean $\frac{1}{2}$; (iii) if the final state is in group $a$, i.e., $s(1)$ or $s(2)$, the reward at step $h = 4$ follows Bernoulli distribution with a slightly higher mean equal to $\frac{1}{2} + \epsilon$. In this way, the optimal policy will take action sequence $(a(1), a(2), a(1))$ for all episodes, so that she can remain in group $a$ and enjoy a larger expected total reward in every episode equal to $\frac{1}{2} + \epsilon$. In contrast, the online learner has to try every possible sequence of actions to figure out which sequence provides larger reward with high probability. Since there are $A^H$ number of possible action sequences, according to the Hoeffding's inequality, we can show that the sample complexity for achieving an $\epsilon$-optimal policy is $\Omega(A^H/\epsilon^2)$.

## 4 Optimality under Partial OSI and Independent Sub-States

While learning in the world of general POMDPs with partial OSI is intractable, inspired by the key insights in our lower-bound design, we identify two rich classes of POMDPs with partial OSI that are tractable, for which we provide new near-optimal algorithms. We leave other potential learnable classes as future work. The tractable class that we study in this section is as follows.

**Class 1.** *(POMDPs with partial OSI and independent sub-states) At each step, (step-i) the agent actively selects sub-states $\hat{i}_h^k$ to query, and receives the partial OSI $\{\phi_i(s_h^k)\}_{\{i \in \hat{i}_h^k\}}$; (step-ii) The agent takes the action $a_h^k$ and receives the reward $r_h(\phi_{\hat{i}_h^k}(s_h^k), a_h^k)$; (step-iii) the next state $s_{h+1}^k$ is drawn according to probability $\mathbb{P}_h(\cdot|s_h^k, a_h^k) = \prod_{i=1}^d \mathbb{P}_{h,i}(\phi_i(\cdot)|\phi_i(s_h^k), a_h^k)$, where the product form indicates that the sub-states have independent transition kernels.*

This class is motivated by many important practical applications. For example, in classic wireless channel scheduling Zhao et al. (2007); Chen et al. (2008); Ouyang et al. (2015), the condition of each channel could change independently; and in Martian RockSampling (Silver & Veness, 2010) or autonomous driving Pinto et al. (2018); Jennings & Figliozzi (2019), the condition of each potential rock or path could also change independently. Notably, as we state in Proposition 1, *without the partial OSI* in step-i of Class 1, even learning under independent sub-states could still be intractable.

**Proposition 1.** *(Intractability if not having partial OSI) There exist POMDPs with independent sub-states, such that learning an $\epsilon$-optimal policy necessarily requires $\tilde{\Omega}(A^H/\epsilon^2)$ samples.*

**Remark 2.** *By replacing partial OSI with noisy observations under certain conditions, POMDPs with independent sub-states could be decoupled into paralleled sub-POMDPs, which may be solved using existing methods. In contrast, the query of the agent for partial OSI in Class 1 couples potential sub-POMDPs together, such that existing solutions do not apply or result in poor performance.*

---

**Algorithm 1** Optimistic-Pessimistic Two-Layer Learning (OP-TLL)

---

    **for** $k = 1 : K$ **do**
        *Step 1:* update the weights $w^k(i)$ and probabilities $p^k(i)$ according to Eq. (2).
        **for** $h = 1 : H$ **do**
            *Step-2:* choose a sub-state $i_h^k$ according to probability $p^k(i)$ and query partial OSI $\phi_{i_h^k}(s_h^k)$.
            *Step-3:* take an action $a_h^k$ that maximizes the updated $Q$-value function in Eq. (3).
        **end for**
    **end for**

---

For Class 1, we develop two new near-optimal algorithm. Due to page limits, we focus on the simpler case with $\tilde{d} = 1$ in this section, and introduce our results for the more challenging case with $\tilde{d} > 1$ in Appendix F. Our new algorithm when $\tilde{d} = 1$ is called Optimistic-Pessimistic Two-Layer Learning (OP-TLL). Please see Algorithm 1. At each step $h$, the optimal policy queries a sub-state $i$ according to a fixed distribution $p$, and receives the partial OSI for this queried sub-state. Then, she takes an action according to $\phi_i(s_h)$. We note that the **new challenge** here is: how to utilize partial OSI to avoid the intractability issue shown in Proposition 1 and achieve optimality? To address this question, our OP-TLL algorithm contains two critical learning layers that involve our two new ideas, and obtains a near-optimal regret.

**Idea-I (Update the query policy pessimistically):** This pessimism is because the query policy updated in "Step-1" of Algorithm 1 affects the choice of action $a_h^k$ in Step-3, which requires complete state information for $V$-value estimation. As a result of this, the relation between the regret and model misspecification error Jin et al. (2020) indicates a *linear-in-$K$* regret if the estimation error due to query is not sufficiently considered. Thus, although the state-transition and reward are stochastic, the query needs to be made sufficiently conservatively. Specifically, at the beginning of each episode $k$, OP-TLL updates the query policy as follows,

$$w^k(i) = w^{k-1}(i) \cdot e^{\frac{\eta_1}{dp^{k-1}(i)} \sum_{h=1}^H \hat{r}_h^{k-1}(\phi_i(s_h^{k-1}), a_h^{k-1})}, \text{ and } p^k(i) = \frac{(1-\eta_1)w^k(i)}{\sum_{i'=1}^d w^k(i')} + \frac{\eta_1}{d}, \quad (2)$$

where the estimated reward $\hat{r}_h^{k-1}(\phi_i(s_h^{k-1}), a_h^{k-1}) = r_h(\phi_i(s_h^{k-1}), a_h^{k-1})$, if $i = i_h^k$; and $\hat{r}_h^{k-1}(\phi_i(s_h^{k-1}), a_h^{k-1}) = 0$, otherwise. Note that this is a new variant of the importance sampling method, where the new development lies in estimating the reward by exploiting partial OSI. Moreover, $\eta_1$ is a key parameter that determines how pessimistic the algorithm is. For example, with a smaller $\eta_1$, the term $e^{\frac{\eta_1}{dp^{k-1}(i)} \sum_{h=1}^H \hat{r}_h^{k-1}(\phi_i(s_h^{k-1}), a_h^{k-1})}$ increases more slowly. As a result, the weight $w^k(i)$ increases more slowly, and thus the algorithm behaves more pessimistically. In "Step-2", OP-TLL chooses the query according to probability $p^k(i)$, where the first term $\frac{w^k(i)}{\sum_{i'=1}^d w^k(i')}$ captures the query importance of sub-state $i$ among all sub-states.

**Idea-II (Update the action policy optimistically):** The intuition for this optimism is to minimize the bias in reward estimates, which is critical because the query policy updated in Step-1 relies on the estimated reward. Specifically, in "Step-3", OP-TLL takes an action that maximizes the $Q$-value function following the optimism-in-face-of-uncertainty principle (the new challenge here is how to design the bonus term to address the impact of partial OSI),

$$Q_h^k(\phi_i(s), a) = \min\{r_h(\phi_i(s), a) + [\mathbb{P}_h^k V_{h+1}^k](\phi_i(s), a) + O(\sqrt{H^2/\mathcal{N}_h^k(\phi_i(s), a)}), H\}, \quad (3)$$

where $\mathbb{P}_h^k(\phi_i(s')|\phi_i(s), a) = \frac{\mathcal{N}_h^k(\phi_i(s), a, \phi_i(s'))}{\mathcal{N}_h^k(\phi_i(s), a)}$ is the estimated transition kernel, $\mathcal{N}_h^k(\phi_i(s), a)$ and $\mathcal{N}_h^k(\phi_i(s), a, \phi_i(s'))$ are the number of times $(\phi_i(s), a)$ and $(\phi_i(s), a, \phi_{\hat{i}}(s'))$ have been visited at step $h$ up to episode $k$, respectively, and $V_h^k(\phi_i(s)) = \max_a Q_h^k(\phi_i(s), a)$ is the estimated $V$-value.

**Theorem 2.** *(Regret) For POMDPs with partial OSI ($\tilde{d} = 1$) and independent sub-states, with probability $1 - \delta$ for any $\delta \in (0, 1)$, the regret of our OP-TLL algorithm with parameter $\eta_1 = O(\sqrt{\frac{d \ln d}{H^2 K}})$ can be upper-bounded as follows,*

$$Reg^{OP\text{-}TLL}(K) \leq \tilde{O}\left(AH^3|\tilde{\mathbb{S}}|^2 d\sqrt{K}\left(\ln(AH^2|\tilde{\mathbb{S}}|K/\delta)\right)^2\right). \quad (4)$$

---

**Algorithm 2** Optimistic Maximum Likelihood Estimation with Partial OSI (OMLE-POSI)

---

**Initialization:** $\Theta^0 = \{\theta \in \Theta : \min_{\{h,\hat{i}\}} \sigma_{\tilde{S}}(\tilde{\mathbb{O}}_h^{\hat{i}}) \geq \alpha\}$.
**for** $k = 1 : K$ **do**
    *Step-1:* estimate the models $\hat{\theta} \triangleq (\hat{\mathbb{P}}, \hat{\tilde{\mathbb{O}}}, \hat{\Delta}_1)$ (including partial emission model) according to

$$\Theta^k = \left\{ \hat{\theta} \in \Theta^0 : \sum_{\tau=1}^{k-1} \log P_{\hat{\theta}}^{\pi^\tau}(\Gamma^\tau) \geq \max_{(\mathbb{P}', \tilde{\mathbb{O}}', \Delta_1') \in \Theta^0} \sum_{\tau=1}^{k-1} \log P_{\mathbb{P}', \tilde{\mathbb{O}}', \Delta_1'}^{\pi^\tau}(\Gamma^\tau) - \beta \right\} \cap \Theta^0. \quad (5)$$

    *Step-2:* update the joint policy $\pi^k \triangleq \arg\max_{\pi : \hat{\theta} \in \Theta^k} \mathbb{E}_{\{\pi_q, \pi_a, \Delta_1, \hat{\theta}\}}[\sum_{h=1}^H r_h(\phi_{\hat{i}_h^k}(s_h^k), a_h^k)]$.
    **for** $h = 1 : H$ **do**
        *Step-3:* query the partial OSI $\{\phi_i(s_h^k)\}_{\{i \in \hat{i}_h^k\}}$ according to the query policy $\pi_{q,h}^k$, collect
        partial noisy observation $\tilde{o}_h^k$, and then take an action $a_h^k$ according to the action policy $\pi_{a,h}^k$.
    **end for**
**end for**

---

Theorem 2 shows that OP-TLL achieves a regret that (i) depends *polynomially* in all parameters $A$, $H$, $|\tilde{\mathbb{S}}|$, $d$ and $K$, and (ii) depends on $\sqrt{K}$, which is *tight. To the best of our knowledge, this is the first such near-optimal result for POMDPs with partial OSI.* Similar to algorithm design, the main difficulty in the proof is how to capture the mutual impact between the query and action policies. Due to page limits, please see Appendix E for details and Appendix F for the case when $\tilde{d} > 1$.

## 5    Optimality under Partial OSI and Partial Noisy Observations

In this section, we identify another tractable class (i.e., Class 2 below) of POMDPs with partial OSI, and provide a new near-optimal algorithm. Please see Fig. 1c for a sketch of one step in this class.

**Class 2. (POMDPs with partial OSI and partial noisy observations)** *At each step, (step-i) the agent actively selects sub-states $\hat{i}_h^k$ to query, and receives the partial OSI $\{\phi_i(s_h^k)\}_{\{i \in \hat{i}_h^k\}}$; (step-ii) the agent receives the partial noisy observation $\tilde{o}_h^k$ for the other $d - \tilde{d}$ sub-states that are not queried, where $\tilde{o}_h^k$ is generated according to the partial emission probability $\tilde{\mathbb{O}}_h^{\hat{i}_h^k}\left(\cdot | \{\phi_i(s_h^k)\}_{\{i \notin \hat{i}_h^k\}}\right)$. The partial emission matrix $\tilde{\mathbb{O}}_h^{\hat{i}} \in \mathbb{R}^{O \times |\tilde{\mathbb{S}}|^{d-\tilde{d}}}$ satisfies the partially revealing condition: there exists a constant $\alpha > 0$, such that $\sigma_{\tilde{S}}(\tilde{\mathbb{O}}_h^{\hat{i}}) \geq \alpha$ for any sub-states $\hat{i}$ and step $h$, where $\tilde{S} = |\tilde{\mathbb{S}}|^{d-\tilde{d}}$ and $\sigma_{\tilde{S}}(\cdot)$ denotes the $\tilde{S}$-th largest singular value of a matrix. Namely, $\min_{\{h,\hat{i}\}} \sigma_{\tilde{S}}(\tilde{\mathbb{O}}_h^{\hat{i}}) \geq \alpha$ holds; (step-iii) the agent takes an action $a_h^k$ and receives the reward $r_h(\phi_{\hat{i}_h^k}(s_h^k), a_h^k)$; (step-iv) the next state $s_{h+1}^k$ is drawn according to the joint transition probability $\mathbb{P}_h(\cdot | s_h^k, a_h^k)$.*

We note that in classic POMDPs (Chen et al., 2022a; Liu et al., 2022; 2023), the noisy observation is independent of the decisions of the agent. In contrast, in Class 2, at each step, the partial noisy observation $\tilde{o}_h^k$ depends on the query $\hat{i}_h^k$ of the agent. This **new dependency** results in new non-trivial challenges in both the algorithm design and regret analysis. For clarity, we use $\Gamma_h^k \triangleq \{\hat{i}_1^k, \phi_{\hat{i}_1^k}(s_1^k), \tilde{o}_1^k, a_1^k, ..., \hat{i}_{h-1}^k, \phi_{\hat{i}_{h-1}^k}(s_{h-1}^k), \tilde{o}_{h-1}^k, a_{h-1}^k\}$ to denote the feedback (including both the partial OSI $\Phi_h^k$ and partial noisy observations $\tilde{o}_{1:h-1}^k$) in this case.

**Remark 3.** *The partially revealing condition in step-ii of Class 2 is milder than the weakly revealing condition in Liu et al. (2022) that requires $\min_h \sigma_S(\mathbb{O}_h) \geq \alpha$, where $S = |\tilde{\mathbb{S}}|^d$ is the total number of states and $\mathbb{O}_h$ is the emission matrix that we introduced in Sec. 2.1. This is because for an $m \times n$ matrix $\mathbb{A}$ and an $m \times (n - l)$ sub-matrix $\mathbb{B}$ of $\mathbb{A}$, we have that $\sigma_{i+l}(\mathbb{A}) \leq \sigma_i(\mathbb{B})$ Horn et al. (1994).*

**Remark 4.** *Without the partially revealing condition in step-ii of Class 2, POMDPs with partial OSI are still intractable in the worst case. This can be shown by letting the partial emission probability $\tilde{\mathbb{O}}_h^{\hat{i}}$ of each $\hat{i}$ be the* same *for all possible sub-states $\{\phi_i(s)\}_{i \notin \hat{i}}$, and then we can show that learning an $\epsilon$-optimal policy in POMDPs with partial OSI still necessarily requires $\tilde{\Omega}(A^H / \epsilon^2)$ samples.*

For Class 2, we develop a new near-optimal algorithms (see Algorithm 2), called Optimistic Maximum Likelihood Estimation with Partial OSI (OMLE-POSI). Recall that the **new challenges** here are: (i) the partial noisy observation $\tilde{o}_h^k$ depends on the query $\hat{i}_h^k$ of the agent; (ii) the performance of the action policy $\pi_{a,h}^k$ depends on both the observation $\tilde{o}_h^k$ and query $\hat{i}_h^k$. Our algorithm is inspired by the idea of OMLE, but extends it to elegantly address the non-trivial joint query policy and action policy optimization. Specifically, OMLE-POSI (in Algorithm 2) differs from OMLE in two aspects. **First,** in "Step-1", OMLE-POSI only collects partial noisy observations $\tilde{o}_{1:H}^k$, which relies on the queries $\hat{i}_{1:H}^k$ determined in Step-2. Due to this new relation, in Eq. (5) we design *a new bonus term* $\beta = O\left(\left(|\tilde{\mathbb{S}}|^{2d}A + |\tilde{\mathbb{S}}|^{d-\tilde{d}}O\right)\ln(|\tilde{\mathbb{S}}|^d AOHK)\right)$ which depends on the size of the non-queried sub-state space $|\tilde{\mathbb{S}}|^{d-\tilde{d}}$, and OMLE-POSI only estimates partial emission model $\tilde{\mathbb{O}}$. **Second,** note that in the joint optimization of "Step-2", the action policy $\pi_a$ is inherently a function of the query policy $\pi_q$, since the action $a_h^k$ taken according to $\pi_{a,h}^k$ relies on the observation $\tilde{o}_h^k$, which further depends on the query $\hat{i}_h^k$ made according to $\pi_{q,h}^k$. Due to page limits, please see Appendix G for more details.

**Theorem 3.** *(Regret) For POMDPs with the partial OSI and partially revealing condition, with probability $1 - \delta$, when $|\tilde{\mathbb{S}}| > (d/\tilde{d})^2$, the regret of OMLE-POSI can be upper-bounded as follows,*

$$Reg^{\text{OMLE-POSI}}(K) \leq \tilde{O}\left(|\tilde{\mathbb{S}}|^{2d-\tilde{d}}OAH^4\sqrt{K(|\tilde{\mathbb{S}}|^{2d}A + |\tilde{\mathbb{S}}|^{(d-\tilde{d})/2}O)/\alpha^2}\right). \tag{6}$$

Theorem 3 above shows that (i) the regret of OMLE-POSI depends on $\sqrt{K}$, which is *tight*; (ii) the regret depends *polynomially* on $A$ and $H$; and (iii) the regret further *decreases* exponentially as $\tilde{d}$ increases. *To the best of our knowledge, this is the first such near-optimal result for POMDPs with partial OSI.* Recall that partial OSI affects both the MLE and policy optimization. Thus, the main difficulty in the proof of Theorem 3 is how to capture such new effects. Indeed, directly applying existing observable operator method (OOM) Jaeger (2000); Liu et al. (2022) will result in a regret that does not decrease with $\tilde{d}$. Please see Appendix G for our new analytical ideas and the proof.

**Theorem 4.** *(Lower bound) For POMDPs with the partial online state information and partially revealing condition, the regret of any algorithm $\pi$ can be lower-bounded as follows,*

$$Reg^\pi(K) \geq \tilde{\Omega}\left(\sqrt{AH} \cdot |\tilde{\mathbb{S}}|^{d/2} \cdot \sqrt{K}\right). \tag{7}$$

Theorem 4 indicates that the dependency on $|\tilde{\mathbb{S}}|^{d/2}$ in the regret of OMLE-POSI is *necessary*. Our key proof idea in Appendix H is to construct a new special state transition, such that even with partial OSI, all combinations of sub-states $\phi_i(s)$ must be explored to achieve a sub-linear regret. We conjecture that a stronger lower bound depending on the *query capability* would be $\tilde{\Omega}\left(\sqrt{AH} \cdot |\tilde{\mathbb{S}}|^{(d-\tilde{d})/2} \cdot \sqrt{K}/\alpha\right)$, and leave this as a future open question.

## 6 DISCUSSION AND CONCLUSION

It is worthwhile to draw connection of our POMDP setting with the standard POMDP and general decision making problem. First, our POMDP setting can be placed under the general decision-making setting (Foster et al., 2021; Chen et al., 2022b; Foster et al., 2023). However, directly instantiating their result to our Classes 1 and 2 will result in worse regret upper bounds than our results here, which exploit our special problem structure such as the dependency of the action policy $\pi_a$ on the query policy $\pi_q$ for developing more refined bounds. Second, our POMDP setting cannot be placed under the standard POMDP setting (Liu et al., 2022; Chen et al., 2022a), mainly due to the special sequential structure of the query, observation, action, and reward in our process. More detailed discussion is provided in Appendix B.

To conclude, this paper answers a fundamental open question: how much online state information (OSI) is sufficient to achieve tractability in POMDPs? Specifically, we establish a lower bound that reveals a *surprising hardness* result: unless we have *full* OSI, we need an exponential complexity to obtain an $\epsilon$-optimal policy for POMDPs. Nonetheless, we identify two novel tractable classes of POMDPs with only *partial* OSI, which are important in practice. For these two classes, we provide three new RL algorithms, which are shown to be *near-optimal* by establishing new regret upper and lower bounds. There are several interesting future work. For example, it would be interesting to study the value of partial OSI in more general POMDPs, e.g., with continuous state spaces (Cai et al., 2022; Liu et al., 2023). Second, the regret upper and lower bounds that we achieved can be further tightened, e.g., improve the dependency on $d$ and $O$ using ideas from Chen et al. (2023).

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

## NOTATIONS

For the convenience of readers, we summarize the notations in Table 1.

Table 1: Notations

| | |
|---|---|
| $\mathcal{S}$ | state space |
| $\mathcal{A}$ | action space |
| $\mathcal{O}$ | observation space |
| $S$ | number of states |
| $A$ | number of actions |
| $O$ | number of observations |
| $H$ | number of steps in each episode |
| $K$ | number of episodes |
| $\Delta_1$ | initial state probability measure |
| $\mathbb{P}$ | state transition probability measure |
| $\mathbb{O}$ | emission probability measure |
| $r$ | reward function |
| $d$ | dimension of the feature vector of each state |
| $\phi_i(s)$ | $i$-th sub-state of state $s$ |
| $\tilde{\mathbb{S}}$ | sub-state set |
| $|\cdot|$ | the cardinality of a set |
| $\tilde{d}$ | query capability: number of sub-states that can be queried |
| $\hat{i}_h^k$ | indices of the queried sub-states |
| $\Phi_h^k$ | feedback *before* the partial OSI for step $h$ of episode $k$ is revealed |
| $\hat{\Delta}_h(\{\hat{i}\}\|\tilde{d})$ | conditional probability measure supported on the query space $\{\hat{i} : \|\hat{i}\| = \tilde{d}\}$ |
| $\tilde{\Delta}_h(\mathcal{A})$ | probability measure supported on the action space $\mathcal{A}$ |
| $\hat{\Phi}_h$ | feedback space of $\Phi_h^k$ *before* the partial OSI for step $h$ is revealed |
| $\tilde{\Phi}_h$ | feedback space *after* the partial OSI for step $h$ has been revealed |
| $\pi_{q,h}^k$ | query policy for step $h$ in episode $k$ |
| $\pi_{a,h}^k$ | action policy for step $h$ in episode $k$ |
| $\pi_h^k$ | joint query-and-action policy for step $h$ in episode $k$ |
| $V^{\pi^k}$ | V-value of the joint policy $\pi^k$ |
| $Reg^{\pi^{1:K}}(K)$ | regret of the online joint policy $\pi^{1:K}$ |
| $\mathbb{P}_h^k(\phi_i(s')\|\phi_i(s),a)$ | the estimated transition kernel |
| $\mathcal{N}_h^k(\phi_i(s),a)$ | number of times $(\phi_i(s),a)$ has been visited at step $h$ up to episode $k$ |
| $\mathcal{N}_h^k(\phi_i(s),a,\phi_{\hat{i}}(s'))$ | number of times $(\phi_i(s),a,\phi_{\hat{i}}(s'))$ has been visited at step $h$ up to episode $k$ |
| $\tilde{o}_h^k$ | partial noisy observation for the $d - \tilde{d}$ sub-states that are not queried |
| $\tilde{\mathbb{O}}_h^{\hat{i}_h^k}\left(\cdot\|\{\phi_i(s_h^k)\}_{\{i \notin \hat{i}_h^k\}}\right)$ | partial emission probability |
| $\Gamma_h^k$ | feedback in Class 2, including both the partial OSI and partial noisy observations |
| $r_{h,i}^k$ | the reward value $r_h(\phi_i(s_h^k),a_h^k)$ of the $i$-th sub-state at step $h$ of episode $k$ |
| $\hat{r}_h^{k-1}$ | the estimated reward at step $h$ of episode $k$ |
| $\theta$ | joint problem model |
| $\bar{\mathbb{O}}^{\hat{i}} \in \mathbb{R}^{O \times \|\tilde{\mathbb{S}}\|^d}$ | augmented partial emission matrix |

## A  MORE RELATED WORK ON POMDPs

Theoretical studies on partially observable Markov decision processes (POMDPs) have had a long history Jaeger (2000); Åström (1965); Smallwood & Sondik (1973); Sondik (1978); Kaelbling et al. (1998); Hauskrecht (2000). For example, Åström (1965) studied limitations of dynamic programming in solving POMDPs. Smallwood & Sondik (1973) studied properties of the optimal control policy and the optimal payoff function in a finite-state discrete POMDP problem. Sondik (1978) studied implementable approximation solutions for stationary policies in POMDPs. Kaelbling et al.

(1998) studied properties of the finite-memory controller in a POMDP problem motivated by robotic navigation. However, these studies did not provide the performance guarantee on the regret or the tractability.

Recently, there has been a significant progress on performance-guaranteed reinforcement learning algorithms for POMDPs. For example, Efroni et al. (2022) studied $m$-step decodable POMDPs, where there is a specific one-to-one mapping from the noisy observation in a history of $m$-steps (i.e., from step $h - m + 1$ to step $h$) to the current true underlying state; Jiang et al. (2017) studied reactive POMDPs, where the optimal value/action at each step is assumed to be independent of any past (i.e., any step before the current step $h$) decisions and feedback; Zhang et al. (2022) studied POMDPs with block MDPs, where each underlying state can be directly learned based on the noisy observation; Kwon et al. (2021) studied POMDPs with latent MDPs, where an unknown MDP model is selected by the environment at the beginning of each episode, and is executed for the episode; Xiong et al. (2022) studied POMDPs with reachability, where the critical exploration is not handled; Golowich et al. (2022) studied POMDPs with $\gamma$-observability, where $\gamma$ characterizes how different the emission probabilities are for different states; Chen et al. (2022a); Zhong et al. (2022) studied predictive state representations, where the probability of each near-term future observations is a weighted linear combination of the probability of a special subset of observations; Liu et al. (2022; 2023) studied $m$-step weakly-revealing POMDPs, where the current state can be statistically decoded based on $m$-step near-term future observations; and Foster et al. (2021); Chen et al. (2022b); Foster et al. (2023) studied general decision making with structured observations, which includes POMDPs as a special case. However, these results typically rely on various assumptions on the emission model or the underlying states, which may not always hold in practice.

To circumvent the dependency and sometimes strong assumptions on the emission probability distribution, Sinclair et al. (2023); Lee et al. (2023) studied the benefit of hindsight state information. Specifically, Sinclair et al. (2023) studied POMDPs with exogenous inputs, where the state transition function and reward function are parameterized by an exogenous input. This exogenous input will be known after the action is taken, i.e., in hindsight. Lee et al. (2023) studied POMDPs with full observability, where the unknown underlying state will be revealed to the agent at the end of each episode, i.e. full hindsight state information. Thus, these recent work studying POMDPs with hindsight state information typically assume full hindsight state information or full observability, which is usually difficult to obtain in practice.

## B  COMPARISON WITH THE STANDARD POMDP SETTING AND COMPARISON BETWEEN CLASS 1 AND CLASS 2

In this section, we provide comparisons with the standard POMDP setting. In addition, we provide a comparison between the identified Class 1 and Class 2.

### B.1  COMPARE WITH THE STANDARD POMDP SETTING

Our POMDP setting with partial OSI cannot be placed under the standard POMDP setting (Liu et al., 2022; Chen et al., 2022a). Recall that in our setting, at each step, the partial noisy observation $\tilde{o}_h^k$ depends on the query $\hat{i}_h^k$ of the agent, whereas standard POMDP does not allow the observation to depend on the action of the same time step. One may resolve this issue by letting the observation at step $h$ be $o_h = (\phi_{\hat{i}_h}(s_h), \tilde{o}_h)$ and letting the action be $A_h = (a_h, \hat{i}_{h+1})$, so that $\hat{i}_h$ becomes the action taken in the previous time step $h - 1$ and hence the observation $o_h$ can depend on such an action $\hat{i}_h$. However, there is still a major issue here. In our process, after taking $a_h$, the agent receives the reward $r_h$, and then takes the query $\hat{i}_{h+1}$ (which depends on $r_h$). In contrast, the above argument of $A_h = (a_h, \hat{i}_{h+1})$ requires taking action $a_h$ and $\hat{i}_{h+1}$ simultaneously, which is not consistent with our setting.

Moreover, we conjecture that a lower bound depending on the revealing condition $\alpha$ could be developed. However, note that the development of the regret lower bound (which depends on $\alpha$) and the development of the regret lower bound in our paper require two different state transitions. In particular, a very special sub-state transition is needed in our case for making partial OSI not useful for the learning agent. We leave this as an interesting future work.

### B.2 THE COMPARISON BETWEEN CLASS 1 AND CLASS 2

We note that each of two classes, i.e., Class 1 and Class 2, becomes learnable due to different natures in their transition model structures, and hence requires very different algorithm designs to handle these specialities. Below we highlight the two key differences between Class 1 and Class 2 that make it difficult to unify the approaches for them. The first difference is the structure of the state transition kernel $\mathbb{P}$. In Class 1, the state transition probability is assumed to be $\mathbb{P}_h(\cdot|s_h^k, a_h^k) = \prod_{i=1}^d \mathbb{P}_{h,i}(\phi_i(\cdot)|\phi_i(s_h^k), a_h^k)$. That is, it is the product of independent transition kernels of sub-states. In contrast, in Class 2, we do not need such a requirement. The second difference is the additional noisy observation $\tilde{o}$. In Class 1, in addition to the partial OSI, the agent receives the partial noisy observation $\tilde{o}_h^k$ for the $d - \tilde{d}$ sub-states that are not queried, where $\tilde{o}_h^k$ is generated according to the partial emission probability $\tilde{\mathbb{O}}_h^{\hat{i}_h^k}\left(\cdot|\{\phi_i(s_h^k)\}_{\{i \notin \hat{i}_h^k\}}\right)$. Moreover, the partial emission matrix $\tilde{\mathbb{O}}_h^{\hat{i}} \in \mathbb{R}^{O \times |\tilde{\mathbb{S}}|^{d-\tilde{d}}}$ is assumed to satisfy the partially revealing condition: there exists a constant $\alpha > 0$, such that $\sigma_{\tilde{S}}(\tilde{\mathbb{O}}_h^{\hat{i}}) \geq \alpha$ for any sub-states $\hat{i}$ and step $h$, where $\tilde{S} = |\tilde{\mathbb{S}}|^{d-\tilde{d}}$ and $\sigma_{\tilde{S}}(\cdot)$ denotes the $\tilde{S}$-th largest singular value of a matrix. Namely, $\min_{\{h,\hat{i}\}} \sigma_{\tilde{S}}(\tilde{\mathbb{O}}_h^{\hat{i}}) \geq \alpha$ holds. In contrast, Class 1 does not require any additional noisy observation $\tilde{o}$ at all.

## C  PROOF OF THEOREM 1

In this section, we provide the complete proof for Theorem 1 with general $d$ and $\tilde{d}$.

*Proof.* As we discussed in Sec. 3, to prove Theorem 1, the most **important** parts are to construct special state representations, state transitions, and reward functions, such that partial online state information (OSI) cannot help the learner to improve her statistical knowledge about the true underlying state. As a result, the learner can only rely on the relation between the action sequence and reward to learn the optimal policy. Therefore, if we carefully construct the reward function for the optimal and sub-optimal action sequences, we should be able to guarantee that only after enough (i.e., exponential number of) episodes, can the learner figure out the $\epsilon$-optimal policy, i.e., the one that determines which probability distribution over the action sequences conditioned on the noisy observation is the best with high probability. Towards this end, we construct a hard instance with $2d$ states, i.e., $s(1), s(2), ..., s(2d)$. For ease of elaboration, we start from the case when $\tilde{d} = 1$, which is ready and easy to be extended to the cases with $\tilde{d} > 1$ as we discuss at the end of the proof.

### C.1  OUR IDEAS FOR CONSTRUCTING THE STATE REPRESENTATION

Our **key idea** is to construct a special state representation, such that by observing only $\tilde{d}$ sub-states, it is still difficult or impossible for the learner to infer the true underlying state. Specifically, the first $d$ states $s(1), s(2), ..., s(d)$ are represented as follows,

$$\vec{\phi}(s(1)) = \begin{bmatrix} x_1 \\ x_2 \\ \vdots \\ x_d \end{bmatrix}, \vec{\phi}(s(2)) = \begin{bmatrix} x_{d+1} \\ x_{d+2} \\ x_3 \\ \vdots \\ x_d \end{bmatrix}, ..., \vec{\phi}(s(\delta)) = \begin{bmatrix} x_1 \\ x_2 \\ \vdots \\ x_{\delta-2} \\ x_{d+\delta-1} \\ x_{d+\delta} \\ x_{\delta+1} \\ \vdots \\ x_d \end{bmatrix}, ..., \vec{\phi}(s(d)) = \begin{bmatrix} x_1 \\ x_2 \\ \vdots \\ x_{d-2} \\ x_{2d-1} \\ x_{2d} \end{bmatrix}. \quad (8)$$

The last $d$ states $s(d+1)$, $s(d+2)$, ..., $s(2d)$ are represented as follows

$$\vec{\phi}(s(d+1)) = \begin{bmatrix} x_{d+1} \\ x_2 \\ \vdots \\ x_d \end{bmatrix}, \vec{\phi}(s(d+2)) = \begin{bmatrix} x_1 \\ x_{d+2} \\ x_3 \\ \vdots \\ x_d \end{bmatrix}, ..., \vec{\phi}(s(d+\delta)) = \begin{bmatrix} x_1 \\ \vdots \\ x_{\delta-1} \\ x_{d+\delta} \\ x_{\delta+1} \\ \vdots \\ x_d \end{bmatrix},$$

$$..., \vec{\phi}(s(2d)) = \begin{bmatrix} x_1 \\ x_2 \\ \vdots \\ x_{d-1} \\ x_{2d} \end{bmatrix}. \quad (9)$$

That is, we first let the state-vector of state $s(1)$ be $\vec{\phi}(s(1)) = [x_1, x_2, ..., x_d]^{\mathrm{T}}$. Then,

- The representation of each of the states $s(2)$, $s(3)$ ..., $s(d)$ differs from the representation of state $s(1)$ by two and only two values. Precisely, the differences between the representation of the state $x(\delta)$ and the representation of state $x(1)$ are the $(\delta-1)$-th element and the $\delta$-th element in the state vector, where $1 < \delta \leq d$. As shown in Eq. (8), we let the new values of the $(\delta-1)$-th element and the $\delta$-th element in the state vector of $s(\delta)$ be $x_{d+\delta-1}$ and $x_{d+\delta}$, respectively.

- The representation of each of the states $s(d+1)$, ..., $s(2d)$ differ from the representation of state $s(1)$ by exactly one value. Precisely, the difference between the representation of the state $x(d+\delta)$ and the representation of state $x(1)$ is the $(d+\delta)$-th element, where $1 \leq \delta \leq d$. As shown in Eq. (9), we let the new value of the $(d+\delta)$-th element in the state vector of $s(d+\delta)$ be $x_{d+\delta}$.

Next, we introduce the high-level reasons for constructing the state representations in this way. For ease of elaboration, let us consider states $s(1)$, $s(2)$, ..., $x(d)$ as a group of states, and we call it group $a$. Similarly, we consider states $s(d+1)$, $s(d+2)$, ..., $x(2d)$ as another group of states, and we call it group $b$.

There are two important properties of our construction of the state representation. The **first property** is that, each state in group $a$ (i.e., $s(1)$, $s(2)$, ..., $s(d)$) must contain at least one same sub-state as that in each state of group $b$ (i.e., $s(d+1)$, $s(d+2)$, ..., $s(2d)$). For example, the first sub-states of state $s(1)$ from group $a$ and states $s(d+2)$, ..., $s(2d)$ from group $b$ are all $x_1$. This means that, by only receiving the partial online state information $\phi_1(s) = x_1$, the learner is not able to know whether she is in a state from group $a$ or group $b$. As another example, the second sub-states of state $s(1)$ from group $a$ and states $s(d+1)$, $s(d+3)$, ..., $s(2d)$ from group $b$ are $x_2$. This means that, by only receiving the partial online state information $\phi_2(s) = x_2$, the learner is also not able to know whether she is in a state from group $a$ or group $b$. As a result, if (i) there is only one specific action sequence that guarantees the learner to be in group $a$, and (ii) group $a$ incurs a larger reward, then intuitively the learner has to constantly keep trying all exponential number of possible action sequences to figure this (only choosing that *unknown* specific action sequence generates a larger expected reward) out with high probability.

The **second property** is that any combination of the sub-states of any state from one group must exist in some states from another group. For example, the sub-state sequence in state $s(1)$ (from group $a$) is $x_1, x_2, ..., x_d$. The same combination of sub-states can be collected by receiving the first sub-state of state $s(d+2)$ (from group $b$), and the second to the last sub-states of state $s(d+1)$ (from group $b$). As another example, the sub-state sequence in state $s(d+1)$ (from group $b$) is $x_{d+1}, x_2, ..., x_d$. The same combination of sub-states can be collected by receiving the first sub-state of state $s(2)$ (from group $a$) and the second to the last sub-states of state $s(1)$ (from group $a$). This property is key to guarantee that the combination of partial OSI from different times still does not help the learner to improve her statistical knowledge about the true underlying state. Please see our more detailed discussions below.

Similar to the simple case when $d = 2$, another question still remains: whether a combination of the partial online state information collected from different times is enough to learn the true underlying states efficiently? To answer this question, we construct a special state transition using our second idea below. Together with the second property of the state representation that we construct above, this special state transition causes learning-difficulty for the learner, even when multiple partial OSI from different times are combined.

### C.2 OUR IDEAS FOR CONSTRUCTING THE SUB-STATE TRANSITION

Our **key idea** here is to construct a special state transition, such that even by combining the partial OSI about different sub-states from different times together, it is still difficult for the learner to infer the true underlying state. Specifically, the state transition probabilities at step $h = 1$ are as follows,

$$\mathbb{P}_1(s_2|s(1), a(1)) = \begin{cases} 1/d, & \text{if } s_2 \in \{s(1), s(2), ..., s(d)\}; \\ 0, & \text{if } s_2 \in \{s(d+1), s(d+2), ..., s(2d)\}; \end{cases} \tag{10}$$

$$\mathbb{P}_1(s_2|s(1), a(2)) = \begin{cases} 0, & \text{if } s_2 \in \{s(1), s(2), ..., s(d)\}; \\ 1/d, & \text{if } s_2 \in \{s(d+1), s(d+2), ..., s(2d)\}. \end{cases} \tag{11}$$

The state transition probabilities at step $h = 2$ are as follows,

$$\mathbb{P}_2(s_3|s_2, a(1)) = \begin{cases} 0, & \text{if } s_3 \in \{s(1), s(2), ..., s(d)\} \text{ and for all } s_2; \\ 1/d, & \text{if } s_3 \in \{s(d+1), s(d+2), ..., s(2d)\} \text{ and for all } s_2; \end{cases} \tag{12}$$

$$\mathbb{P}_2(s_3|s_2, a(2)) = \begin{cases} 1/d, & \text{if } s_3 \in \{s(1), s(2), ..., s(d)\} \text{ and } s_2 \in \{s(1), s(2), ..., s(d)\}; \\ 0, & \text{if } s_3 \in \{s(1), s(2), ..., s(d)\} \text{ and } s_2 \in \{s(d+1), s(d+2), ..., s(2d)\}; \\ 0, & \text{if } s_3 \in \{s(d+1), s(d+2), ..., s(2d)\} \text{ and } s_2 \in \{s(1), s(2), ..., s(d)\}, \\ 1/d, & \text{if } s_3 \in \{s(d+1), s(d+2), ..., s(2d)\} \text{ and } s_2 \in \{s(d+1), ..., s(2d)\}. \end{cases} \tag{13}$$

The state transition probabilities at step $h = 3$ are as follows,

$$\mathbb{P}_3(s_4|s_3, a(1)) = \begin{cases} 1/d, & \text{if } s_4 \in \{s(1), s(2), ..., s(d)\} \text{ and } s_3 \in \{s(1), s(2), ..., s(d)\}; \\ 0, & \text{if } s_4 \in \{s(1), s(2), ..., s(d)\} \text{ and } s_3 \in \{s(d+1), s(d+2), ..., s(2d)\}; \\ 0, & \text{if } s_4 \in \{s(d+1), s(d+2), ..., s(2d)\} \text{ and } s_3 \in \{s(1), s(2), ..., s(d)\}, \\ 1/d, & \text{if } s_4 \in \{s(d+1), s(d+2), ..., s(2d)\} \text{ and } s_3 \in \{s(d+1), ..., s(2d)\}. \end{cases} \tag{14}$$

$$\mathbb{P}_3(s_4|s_3, a(2)) = \begin{cases} 0, & \text{if } s_3 \in \{s(1), s(2), ..., s(d)\} \text{ and for all } s_3; \\ 1/d, & \text{if } s_4 \in \{s(d+1), s(d+2), ..., s(2d)\} \text{ and for all } s_3. \end{cases} \tag{15}$$

That is, in each episode, the learner starts from state $s_1 = s(1)$, i.e.,

$$\Delta_1(s_1) = \begin{cases} 1, & \text{if } s_1 = s(1); \\ 0, & \text{otherwise}. \end{cases} \tag{16}$$

Then, for step $h = 1$ to step $h = 3$, we let

- At step $h = 1$, (i) if action $a(1)$ is chosen, the state $s_1$ will transition to $s(1), s(2), ..., s(d)$ from group $a$ with the same probability; (ii) if action $a(2)$ is chosen, the state will transition to $s(d + 1), s(d + 2), ..., s(2d)$ with the same probability.

- At step $h = 2$, (i) if action $a(1)$ is chosen, all states $s(1), s(2), ..., s(d)$ from group $a$ will transition to states $s(d + 1), s(d + 2), ..., s(2d)$ from group $b$ with the same probability; all states $s(d + 1), s(d + 2), ..., s(2d)$ from group $b$ will transition to states $s(d + 1), s(d + 2), ..., s(2d)$ from group $b$ with the same probability; (ii) if action $a(2)$ is chosen, all states $s(1), s(2), ..., s(d)$ from group $a$ will transition to states $s(1), s(2), ..., s(d)$ from group $a$ with the same probability; all states $s(d+1), s(d+2), ..., s(2d)$ from group $b$ will transition to states $s(d + 1), s(d + 2), ..., s(2d)$ from group $b$ with the same probability.

- At step $h = 3$, (i) if action $a(1)$ is chosen, all states $s(1)$, $s(2)$, ..., $s(d)$ from group $a$ will transition to states $s(1)$, $s(2)$, ..., $s(d)$ from group $a$ with the same probability; all states $s(d+1)$, $s(d+2)$, ..., $s(2d)$ from group $b$ will transition to states $s(d+1)$, $s(d+2)$, ..., $s(2d)$ from group $b$ with the same probability; (ii) if action $a(2)$ is chosen, all states $s(1)$, $s(2)$, ..., $s(d)$ from group $a$ will transition to states $s(d+1)$, $s(d+2)$, ..., $s(2d)$ from group $b$ with the same probability; all states $s(d+1)$, $s(d+2)$, ..., $s(2d)$ from group $b$ will transition to states $s(d+1)$, $s(d+2)$, ..., $s(2d)$ from group $b$ with the same probability.

- Note that for all states $s(d+1)$, $s(d+2)$, ..., $s(2d)$ from group $b$, at step $h = 2$ and $h = 3$, no matter which action is chosen, they will transition to $s(d+1)$, $s(d+2)$, ..., $s(2d)$ from group $b$ with the same probability.

Then, together with the state representations that we construct before, even when the partial OSI about different sub-states from different times are combined together, such a construction for the state transition still prevents the learner from knowing which group of states she is in. For example, at step $h = 1$ of $d$ consecutive episodes, the learner can keep taking action $a(1)$ and query the first sub-state to the last sub-state one-by-one. Then, the partial OSI at step $h = 2$ could be $\phi_1(s_2^k) = x_{d+1}$ (i.e., the first sub-state of $s(2)$), $\phi_2(s_2^{k+1}) = x_2$ (i.e., the second sub-state of $s(1)$), $\phi_2(s_2^{k+2}) = x_3$ (i.e., the third sub-state of $s(2)$, ..., $\phi_2(s_2^{k+d}) = x_d$ (i.e., the last sub-state of $s(2)$. However, note that the sub-states of $s(d+1)$ are also $x_{d+1}$, $x_2$, ..., $x_d$. Thus, such a combination of partial OSI, i.e.,

$$\phi_1(s_2^k) = x_{d+1}, \phi_2(s_2^{k+1}) = x_2, ..., \phi_2(s_2^{k+d}) = x_d, \tag{17}$$

is not powerful enough for the learner to distinguish whether she is visiting states $s(1)$ and $s(2)$ from group $a$ or she is simply visiting state $s(d+1)$ from group $b$. It is not difficult to see that similar issues occur at other steps.

## C.3 Our Ideas for Constructing the Reward Function

Up to here, we can see that only with partial OSI, the learner cannot improve her statistical knowledge about the true underlying states. Thus, she can only rely on the statistical relation between the sequence of actions that is chosen and the reward that is received. Hence, finally we construct the reward as follows.

- The rewards $r_h$ at steps $h = 1$, $h = 2$ and $h = 3$ are all 0;

- If the final state at step $h = 4$ is in group $b$, i.e., $s(d+1)$, $s(d+2)$, ..., $s(2d)$, the reward at step $h = 4$ follows Bernoulli distribution with mean $\frac{1}{2}$;

- If the final state at step $h = 4$ is in group $a$, i.e., $s(1)$, $s(2)$, ..., $s(d)$, the reward at step $h = 4$ follows Bernoulli distribution with a slightly higher mean equal to $\frac{1}{2} + \epsilon$.

In this way, the optimal policy will take action sequence

$$a_1^* = a(1), a_2^* = a(2), a_3^* = a(1), \tag{18}$$

for all episodes, so that she can stay in group $a$ and enjoy a larger expected total reward in every episode equal to $\frac{1}{2} + \epsilon$. Note that the optimal action sequence $(a(1), a(2), a(1))$ in Eq. (18) is simply because of the specific constructions that we introduce above. This optimal action sequence could easily be changed to any other action sequence $(a_1^*, a_2^*, a_3^*)$, e.g., $(a(1), a(1), a(1))$ or $(a(2), a(2), a(2))$. The **key idea** is that there exists one and only one action sequence that generates larger reward at the end of each episode.

Note that the online learner has no idea about which state or group she is in, and partial OSI cannot provide any help for distinguishing the true underlying states and groups. Thus, in sharp contrast to the optimal policy, the online learner has to try every possible sequence of actions to figure out which sequence of actions provides a larger reward with high probability.

C.4  OUR IDEAS FOR LOWER-BOUNDING THE FINAL REGRET

Since there are $A^{H-1}$ number of possible action sequences, according to the Hoeffding's inequality, we can show that the sample complexity for achieving an $\epsilon$-optimal policy is $\Omega(A^H/\epsilon^2)$. Precisely, based on the reasons from Appendix C.1, Appendix C.2 and Appendix C.3, this hard instance is equivalent to finding the best arm in a multi-armed bandit problem by using the random reward feedback, where each arm corresponds to an action sequence. For this part, similar to the KL-divergence analysis in the bandit learning, we can consider an equivalent instance of the multi-armed bandit problem with $A^{H-1}$ number of arms as follows:

- The reward of one arm $i^*$, i.e., the optimal arm, is generated according to the Bernoulli distribution with mean $\frac{1}{2} + \epsilon$, while the reward of all other $A^{H-1} - 1$ arms, i.e., the sub-optimal arms, are generated according to the Bernoulli distribution with slightly smaller mean $\frac{1}{2}$.

- In addition, the optimal arm $i^*$ is chosen uniformly randomly by the environment.

- Note that at each time $k$, the algorithm chooses one arm based on the past reward feedback $\Gamma^{k-1} = \{0,1\}^{k-1}$.

We use $P_{i^*}(K)$ to denote the probability that the online algorithm $\pi$ chooses the optimal arm $i^*$ in this instance, i.e.,

$$P_{i^*}(K) = Pr\{i^\pi(K) = i^*\}. \tag{19}$$

Thus, in the following, we focus on upper-bounding the expected probability of choosing the optimal arm $i^*$ by any online algorithm $\pi$, where the expectation is taken with respect to the randomness of the feedback and the randomness of the optimal arm $i^*$. To prove this, we can use Pinsker's inequality and Hoeffding's inequality. Specifically, first, we use $P_0(K)$ to denote the probability that the online algorithm $\pi$ chooses the optimal arm $i^*$ in a fictitious case, where the reward of all arms are generated according to the same Bernoulli distribution with mean $\frac{1}{2}$. In such a fictitious case, each arm (including arm $i^*$) performs equally, and thus could be chosen arbitrarily. Next, we prove that the difference between the probability $P_{i^*}(K)$ of choosing the optimal arm $i^*$ in the instance that we construct above and the probability $P_0(K)$ in the fictitious instance can be upper-bounded by $\frac{1}{2}\sqrt{\mathbb{E}_0[\mathcal{N}]\log\frac{1}{1-4\epsilon^2}}$, where $\mathbb{E}_0[\mathcal{N}]$ is the expected number of times choosing the arm $i^*$ when the reward of all arms follows the same distribution.

First, according to the total variation distance and Pinsker's inequality, we have

$$|P_{i^*}(K) - P_0(K)| \leq \|P_{i^*} - P_0\|_{\text{TV}} \leq \sqrt{\frac{1}{2}KL(P_{i^*}\|P_0)}. \tag{20}$$

Next, based on the definition of the KL-divergence and the chain rule, we have

$$KL(P_{i^*}\|P_0) = \sum_{k=1}^{K}\sum_{\{0,1\}^{k-1}:a^k=i^*} P_0(\{0,1\}^{k-1})\left(\frac{1}{2}\log(\frac{1/2}{1/2-\epsilon}) + \frac{1}{2}\log(\frac{1/2}{1/2+\epsilon})\right)$$

$$= \frac{1}{2}\log\left(\frac{1}{1-4\epsilon^2}\right)\sum_{k=1}^{K}P_0(a^k = i^*), \tag{21}$$

where the sum of the probabilities $\sum_{k=1}^{K}P_0(a^k = i^*)$ is equal to the expected number of times choosing the optimal arm $i^*$. Moreover, since the optimal arm $i^*$ is chosen uniformly randomly among all arms, by combining Eq. (20) and Eq. (21), we have that the expected probability of choosing the optimal arm $i^*$ can be upper-bounded as follows,

$$\mathbb{E}_{i^*}[P_{i^*}(K)] \leq \mathbb{E}_{i^*}[P_0(K)] + \frac{1}{2}\mathbb{E}_{i^*}\left[\sqrt{\mathbb{E}_0[\mathcal{N}]\log\frac{1}{1-4\epsilon^2}}\right]. \tag{22}$$

Then, according to Jensen's inequality, from Eq. (22), we have

$$\mathbb{E}_{i^*}[P_{i^*}(K)] \leq \frac{1}{A^{H-1}} + \frac{1}{2}\sqrt{\mathbb{E}_{i^*}[\mathbb{E}_0[\mathcal{N}]]\log\frac{1}{1-4\epsilon^2}}. \tag{23}$$

Finally, since $\mathbb{E}_{i^*}\left[\mathbb{E}_0\left[\mathcal{N}\right]\right] = \frac{K}{A^{H-1}}$ and $\log\frac{1}{1-x} \le 2x$ for all $x \le \frac{1}{2}$, from Eq. (23), we have that whenever $K \le O(\frac{A^H}{\epsilon^2})$,

$$\mathbb{E}_{i^*}\left[P_{i^*}(K)\right] \le \frac{2}{3}, \tag{24}$$

for all $\epsilon \le \sqrt{1/8}$.

Furthermore, when $\tilde{d} > 1$, the same line of proof can be easily generalized to prove the corresponding dependency in the sample complexity lower bound, i.e., we can still use the similar construction of state representations and state transitions, just by guaranteeing that any $\tilde{d}$ combination of sub-states in group $a$ must exist in group $b$, and vice versa. To avoid too much repetition, we take the case with $d = 3, \tilde{d} = 2$ as an example. In this case, we consider 8 states, i.e., $s(1), ..., s(8)$. Then, we let

$$\vec{\phi}(s(1)) = \begin{bmatrix} x_1 \\ x_2 \\ x_3 \end{bmatrix}, \vec{\phi}(s(2)) = \begin{bmatrix} x_1 \\ x_6 \\ x_4 \end{bmatrix}, \vec{\phi}(s(3)) = \begin{bmatrix} x_5 \\ x_2 \\ x_4 \end{bmatrix}, \vec{\phi}(s(4)) = \begin{bmatrix} x_5 \\ x_6 \\ x_3 \end{bmatrix},$$

$$\vec{\phi}(s(5)) = \begin{bmatrix} x_1 \\ x_2 \\ x_4 \end{bmatrix}, \vec{\phi}(s(6)) = \begin{bmatrix} x_5 \\ x_2 \\ x_3 \end{bmatrix}, \vec{\phi}(s(7)) = \begin{bmatrix} x_1 \\ x_6 \\ x_3 \end{bmatrix}, \vec{\phi}(s(8)) = \begin{bmatrix} x_5 \\ x_6 \\ x_4 \end{bmatrix}.$$

In this way, the first property that we mentioned in Appendix C.1 still holds. That is, each state in group $a$ (i.e., $s(1)$, $s(2)$, $s(3)$, and $s(4)$) must contain at least one same combination of $\tilde{d} = 2$ sub-states as that in each state of group $b$ (i.e., $s(5)$, $s(6)$, $x(7)$, and $s(8)$), and vice versa. For example, the first two sub-states of state $s(1)$ and state $s(5)$ are all $(x_1, x_2)$. This means that, by only receiving partial OSI $\phi_{\{1,2\}}(s) = (x_1, x_2)$, the learner is not able to know whether she is in a state from group $a$ or group $b$. As another example, the last two sub-states of state $s(1)$ and states $s(6)$ are $(x_2, x_3)$. This means that, by only receiving partial OSI $\phi_{\{2,3\}}(s) = (x_2, x_3)$, the learner is also not able to know whether she is in a state from group $a$ or group $b$.

Moreover, the second property that we describe in Appendix C.1 still holds. That is, any full combination of the sub-states of any state from one group must exist in some states from another group. For example, the sub-state sequence in state $s(1)$ (from group $a$) is $x_1, x_2, x_3$. The same combination of sub-states can be collected by receiving the first two sub-states of state $s(5)$ (from group $b$), and the second two sub-states of state $s(6)$ (from group $b$). As another example, the sub-state sequence in state $s(5)$ (from group $b$) is $x_1, x_2, x_4$. The same combination of sub-states can be collected by receiving the first two sub-states of state $s(1)$ (from group $a$) and the second two sub-states of state $s(3)$ (from group $a$). Therefore, finally, by constructing the same state-transition and reward function, all the previous proof steps still hold. This concludes the proof. $\square$

## D  PROOF OF PROPOSITION 1

In this section, we provide and prove a more general version of Proposition 1. Please see Proposition 2 below.

**Proposition 2.** *(Intractability) There exist POMDPs with independent sub-states and even with noisy observations, such that with a probability $p \ge 1/3$, learning an $\epsilon$-optimal policy necessarily requires $\tilde{\Omega}(A^H/\epsilon^2)$ samples.*

Proposition 2 indicates that without the partial OSI in item (i) of Class 1 that is defined in Sec. 4, learning in POMDPs with independent sub-states is still intractable, i.e., with exponentially scaling sample complexity.

*Proof.* To prove Proposition 2, we construct a **new** hard instance, where the emission probabilities of all states are exactly the same. Thus, the noisy observation cannot help the learner to improve her statistical knowledge about the true underlying state. As a result, the learner can only rely on the relation between the action sequence and reward to learn the optimal policy. Therefore, if we carefully construct the reward function for the optimal and sub-optimal action sequences, we should

be able to guarantee that only after enough (i.e., exponential number of) episodes, can the learner figure out the $\epsilon$-optimal policy, i.e., the one that determines which probability distribution over the action sequences conditioned on the noisy observation is the best with high probability. The hard instance that we construct for proving Proposition 2 is as follows. We still focus on showing a simple instance that is easy to understand, which is easy to be extended to more general case.

## D.1 Our Ideas for Constructing the Sub-states and the State Representation

We consider a hard instance with the value set of elements $\tilde{\mathbb{S}} = \{x_1, x_2\}$. That is, there are $d = 2$ sub-states and the caldinality is $|\tilde{\mathbb{S}}| = 2$. Thus, the total number of states is $S = |\tilde{\mathbb{S}}|^d = 2^2 = 4$. The representations of these four states are

$$\vec{\phi}(s(1)) = \begin{bmatrix} x_1 \\ x_1 \end{bmatrix}, \vec{\phi}(s(2)) = \begin{bmatrix} x_1 \\ x_2 \end{bmatrix}, \vec{\phi}(s(3)) = \begin{bmatrix} x_2 \\ x_1 \end{bmatrix}, \vec{\phi}(s(4)) = \begin{bmatrix} x_2 \\ x_2 \end{bmatrix}. \tag{25}$$

The representations are not necessarily exactly the same as that in Eq. (25). Our key **idea** is to guarantee that each sub-state takes at least two different values, such that by constructing a special emission model and transition kernel (as follows), it is difficult for the learner to improve her statistical knowledge about the true underlying sub-states.

## D.2 Our Ideas for Constructing the Emission Model

Our key **idea** for constructing the emission model is to guarantee that, at each step $h$, the emission probabilities for all states are exactly the same. As a result, even with noisy observations, the learner cannot improve her statistical knowledge about the true underlying state at all. In other words, we let the emission probability be

$$\mathbb{O}_h(\cdot|s(1)) = \mathbb{O}_h(\cdot|s(2)) = \mathbb{O}_h(\cdot|s(3)) = \mathbb{O}_h(\cdot|s(4)), \text{ for all steps } h. \tag{26}$$

In this way, by receiving any noisy observation, the probability of the true underlying state is the same. Thus, the noisy observation does not provide any useful information for the learner to infer the true underlying states at any step.

## D.3 Our Ideas for Constructing the Sub-state Transition

Recall that in our proof for Theorem 1 in Appendix C, the idea is to construct a special state transition, such that even by combining the partial online state information about different sub-states from different times together, it is still difficult for the learner to infer the true underlying state. In contrast, there is no partial online state information here. Thus, **differently from** the idea in Appendix C, our **idea** here is to guarantee that there exists only one specific sequence of actions, such that for all sub-states, the values that generate a larger reward can be attained. In order to achieve this, we construct the special sub-state transition below. We consider $A = 2$ actions. Specifically, for each sub-state $\phi_i(s)$, at step $h = 1$,

$$\mathbb{P}_1(\phi_i(s_2)|\phi_i(s(1)), a(1)) = \begin{cases} 1, & \text{if } \phi_i(s_2) = x_1; \\ 0, & \text{if } \phi_i(s_2) = x_2; \end{cases} \tag{27}$$

$$\mathbb{P}_1(\phi_i(s_2)|\phi_i(s(1)), a(2)) = \begin{cases} 0, & \text{if } \phi_i(s_2) = x_1; \\ 1, & \text{if } \phi_i(s_2) = x_2; \end{cases} \tag{28}$$

Then, we construct the sub-state transition probabilities of each sub-state $\phi_i(s)$ at step $h = 2$ as follows,

$$\mathbb{P}_2(\phi_i(s_3)|\phi_i(s_2), a(1)) = \begin{cases} 0, & \text{if } \phi_i(s_3) = x_1 \text{ and for all } \phi_i(s_2); \\ 1, & \text{if } \phi_i(s_3) = x_2 \text{ and for all } \phi_i(s_2); \end{cases} \tag{29}$$

$$\mathbb{P}_2(\phi_i(s_3)|\phi_i(s_2), a(2)) = \begin{cases} 1, & \text{if } \phi_i(s_3) = x_1 \text{ and } \phi_i(s_2) = x_1; \\ 0, & \text{if } \phi_i(s_3) = x_1 \text{ and } \phi_i(s_2) = x_2; \\ 0, & \text{if } \phi_i(s_3) = x_2 \text{ and } \phi_i(s_2) = x_1; \\ 1, & \text{if } \phi_i(s_3) = x_2 \text{ and } \phi_i(s_2) = x_2; \end{cases} \tag{30}$$

Finally, we construct the sub-state transition probabilities of each sub-state $\phi_i(s)$ at step $h = 3$ as follows,

$$\mathbb{P}_3(\phi_i(s_4)|\phi_i(s_3), a(1)) = \begin{cases} 1, & \text{if } \phi_i(s_4) = x_1 \text{ and } \phi_i(s_3) = x_1; \\ 0, & \text{if } \phi_i(s_4) = x_1 \text{ and } \phi_i(s_3) = x_2; \\ 0, & \text{if } \phi_i(s_4) = x_2 \text{ and } \phi_i(s_3) = x_1; \\ 1, & \text{if } \phi_i(s_4) = x_2 \text{ and } \phi_i(s_3) = x_2; \end{cases} \tag{31}$$

$$\mathbb{P}_3(\phi_i(s_4)|\phi_i(s_3), a(2)) = \begin{cases} 0, & \text{if } \phi_i(s_4) = x_1 \text{ and for all } \phi_i(s_3); \\ 1, & \text{if } \phi_i(s_4) = x_2 \text{ and for all } \phi_i(s_3); \end{cases} \tag{32}$$

That is, in each episode, the learner starts from state $s_1 = s(1) = [x_1, x_1]^\mathsf{T}$, i.e.,

$$\Delta_1(s_1) = \begin{cases} 1, & \text{if } s_1 = s(1); \\ 0, & \text{otherwise.} \end{cases} \tag{33}$$

Then, for step $h = 1$ to step $h = 3$, we let

- At step $h = 1$, (i) if action $a(1)$ is chosen, each sub-state will transition to $x_1$; (ii) if action $a(2)$ is chosen, each sub-state will transition to $x_2$.

- At step $h = 2$, (i) if action $a(1)$ is chosen, sub-state $x_1$ will transition to states $x_2$ and sub-state $x_2$ will transition to $x_2$; (ii) if action $a(2)$ is chosen, sub-state $x_1$ will transition to states $x_1$ and sub-state $x_2$ will transition to $x_2$.

- At step $h = 3$, (i) if action $a(1)$ is chosen, sub-state $x_1$ will transition to sub-state $x_1$ and sub-state $x_2$ will transition to $x_2$; (ii) if action $a(2)$ is chosen, it will transition to states $x_2$.

- Note that for sub-state $x_2$ at step $h = 2$ and $h = 3$, no matter which action is chosen, the states will transition to $x_2$.

By constructing so, only taking action sequence $a(1), a(2)$ and $a(1)$ will guarantee that both sub-states are in $x_1$.

### D.4 OUR IDEAS FOR CONSTRUCTING THE REWARD FUNCTIONS

Since (i) the emission model that we construct in Appendix D.2 guarantees that the noisy observation cannot help for inferring the true underlying state, and (ii) the sub-state transitions that we construct in Appendix D.3 guarantees that there exists only one specific sequence of actions that guarantees that the sub-state $x_1$ is attained, then our **idea** for constructing the reward functions is to guarantee that only sub-state $x_1$ provides larger reward. Hence, finally we construct the reward functions as follows:

- The rewards $r_h$ at steps $h = 1$, $h = 2$ and $h = 3$ are all 0;

- If the final sub-state is $x_2$, the reward at step $h = 4$ follows Bernoulli distribution with mean $\frac{1}{2}$;

- If the final sub-state is $x_1$, the reward at step $h = 4$ follows Bernoulli distribution with a slightly higher mean equal to $\frac{1}{2} + \epsilon$.

In this way, the optimal policy will take action sequence $(a(1), a(2), a(1))$ for all episodes, so that she can enjoy a larger expected total reward in every episode equal to $\frac{1}{2} + \epsilon$. In contrast, the online learner has to try every $A^{H-1}$ possible sequence of actions to figure out which sequence provides larger reward with high probability. Following the same line of the final part of our proof for Theorem 1, we obtain the final conclusion in Proposition 2. □

# E  OUR NEW ALGORITHM AND REGRET ANALYSIS FOR THE CASE WITH $\tilde{d} = 1$

In this section, we first introduce our new algorithm for the simpler case with $\tilde{d} = 1$ for completeness, i.e., a special case of Class 1 of POMDPs with independent sub-states. Then, we provide our complete proof for the regret of our algorithm, i.e., Theorem 5 below.

**Theorem 5.** *(Regret) For POMDPs with partial OSI ($\tilde{d} = 1$) and independent sub-states, with probability $1 - \delta$ for any $\delta \in (0, 1)$, the regret of our OP-TLL algorithm with parameter $\eta_1 = O(\sqrt{\frac{d \ln d}{H^2 K}})$ can be upper-bounded as follows,*

$$Reg^{OP\text{-}TLL}(K) \leq \tilde{O}\left(AH^3|\tilde{\mathbb{S}}|^2 d\sqrt{K}\left(\ln(AH^2|\tilde{\mathbb{S}}|K/\delta)\right)^2\right). \tag{34}$$

Theorem 5 shows that OP-TLL achieves a regret that depends *polynomially* in $A$, $H$, $|\tilde{\mathbb{S}}|$, $d$ and $K$. We note that the key difference between the tractable Class 1 and factored MDPs is that in the tractable Class 1, the agent needs to actively query only one sub-state, and then observe the information of only this queried sub-state. Hence the problem is still *POMDP*. In contrast, in factored MDPs, the full state is observed, and hence the problem is *MDP*. This is also the main reason that the same regret in factored MDPs cannot be obtained in the tractable Class 1.

## E.1  OPTIMISTIC-PESSIMISTIC TWO-LAYER LEARNING (OP-TLL)

There are three steps in Algorithm 3. In "Step 1", OP-TLL updates the weights $w^k(i)$ and probabilities $p^k(i)$ according to Eq. (35). After updating the query policy $\pi_q$ according to the updated probability distribution, in "Step-2", OP-TLL chooses a sub-state $i_h^k$ according to probability $p^k(i)$ and query partial OSI $\phi_{i_h^k}(s_h^k)$. In "Step-3", OP-TLL takes an action $a_h^k$ that maximizes the updated $Q$-value function in Eq. (37). Thus, OP-TLL contains two critical learning layers that involve our two new ideas.

**Layer-I (Update the query policy pessimistically):** This pessimism is because the query policy updated in Step-1 of Algorithm 1 affects the choice of action $a_h^k$ in Step-3, which requires complete state information for $V$-value estimation. As a result of this, the relation between the regret and model misspecification error Jin et al. (2020) indicates a *linear-in-$K$* regret if the estimation error due to query is not sufficiently considered. Thus, although the state-transition and reward are stochastic, the query needs to be made sufficiently conservatively. Specifically, at the beginning of each episode $k$, OP-TLL updates the weights and probabilities for each sub-state $\phi_i(s)$ according to Eq. (35). We note that this is a new variant of the standard exponential weight method, where the new change in estimating the reward is due to the partial OSI. For example, with a smaller $\eta_1$, the term $e^{\frac{\eta_1}{dp^{k-1}(i)}\sum_{h=1}^{H} r_h^{k-1}}$ increases more slowly. As a result, the weight $w^k(i)$ increases more slowly, and thus the algorithm behaves more pessimistically. In Step-2, OP-TLL chooses the query according to probability $p^k(i)$, where the first term $\frac{w^k(i)}{\sum_{i'=1}^{d} w^k(i')}$ captures the query importance of sub-state $i$ among all sub-states.

**Layer-II (Update the action policy optimistically):** The intuition for this optimism is to minimize the bias in reward estimates, which is critical because the query policy updated in Step-1 relies on the estimated reward. Specifically, in Step-3, OP-TLL takes an action that maximizes the $Q$-value function following the optimism-in-face-of-uncertainty principle. Note that the new challenge here is how to design the bonus term $\beta_h^k = O(\sqrt{H^2/\mathcal{N}_h^k(\phi_{i_h^k}(s), a)})$ to address the impact of partial OSI.

## E.2  PROOF OF THEOREM 5

We first provide a complete statement for the regret upper-bound of our OP-TLL algorithm. For simplicity, we will drop $(\phi_{\hat{i}_h^k}(s_h^k), a_h^k)$ from $r_h(\phi_{\hat{i}_h^k}(s_h^k), a_h^k)$ when it is clear from the context.

---

**Algorithm 3** Optimistic-Pessimistic Two-Layer Learning (OP-TLL)

---

Initialization: $w^1(i) = 1$ and $p^1(i) = \frac{1}{d}$ for all $i = 1, ..., d$.
**for** $k = 1 : K$ **do**

    *Step 1:* update the weights $w^k(i)$ and probabilities $p^k(i)$ according to Eq. (35) as follows,

$$w^k(i) = w^{k-1}(i) \cdot e^{\frac{\eta_1}{dp^{k-1}(i)} \sum_{h=1}^{H} \hat{r}_h^{k-1}(\phi_i(s_h^{k-1}), a_h^{k-1})},$$

$$p^k(i) = \frac{(1 - \eta_1)w^k(i)}{\sum_{i'=1}^{d} w^k(i')} + \frac{\eta_1}{d}, \tag{35}$$

    where

$$\hat{r}_h^{k-1}(\phi_i(s_h^{k-1}), a_h^{k-1}) = r_h^{k-1}(\phi_i(s_h^{k-1}), a_h^{k-1}), \text{ if } i = i_h^k,$$
$$\text{and } \hat{r}_h^{k-1}(\phi_i(s_h^{k-1}), a_h^{k-1}) = 0, \text{ otherwise}, \tag{36}$$

$\eta_1 = O(\sqrt{\frac{d \ln d}{H^2 K}})$ is a key parameter that determines how pessimistic the algorithm is.
**for** $h = 1 : H$ **do**

    *Step-2:* choose a sub-state $i_h^k$ according to probability $p^k(i)$ and query partial OSI $\phi_{i_h^k}(s_h^k)$.

    *Step-3:* take an action $a_h^k$ that maximizes the updated $Q$-value function in Eq. (37) as follows,

$$Q_h^k(\phi_i(s), a) = \min\{r_h(\phi_i(s), a) + [\mathbb{P}_h^k V_{h+1}^k](\phi_i(s), a) + O(\sqrt{H^2/\mathcal{N}_h^k(\phi_i(s), a)}), H\}, \tag{37}$$

    where $\mathbb{P}_h^k(\phi_i(s')|\phi_i(s), a) = \frac{\mathcal{N}_h^k(\phi_i(s), a, \phi_i(s'))}{\mathcal{N}_h^k(\phi_i(s), a)}$ is the estimated transition kernel, $\mathcal{N}_h^k(\phi_i(s), a)$ and $\mathcal{N}_h^k(\phi_i(s), a, \phi_i(s'))$ are the number of times $(\phi_i(s), a)$ and $(\phi_i(s), a, \phi_{\hat{i}}(s'))$ have been visited at step $h$ up to episode $k$, respectively, and $V_h^k(\phi_{i_h^k}(s)) = \max_a Q_h^k(\phi_{i_h^k}(s), a)$ is the estimated $V$-value function.
    **end for**
**end for**

---

*Proof.* The **main challenges** in the proof of Theorem 2 result from the mutual effects between the first learning layer and the second learning layer. Specifically, **first,** note that in Algorithm 1, the first learning layer is affected by the chosen action and $V$-value function estimates in the second learning layer. For example, in the first learning layer, the weight $w^{k+1}(i)$ at each episode is updated based on the reward $\sum_{h=1}^{H} r_h(\phi_{\hat{i}_h^k}(s_h^k), a_h^k)$ received in the last episode, which further depends on the chosen action and $V$-value function estimates in the second learning layer. Thus, a larger gap in the observed reward will make the quality of the weights and probabilities worse. This will then affect the choice of the reward action. **Second,** note that the chosen action and $V$-value estimates in the second learning layer depend on the feedback collected from the determined reward sub-state, which further depend on the first learning layer. Thus, a larger gap in the weights and probabilities will make the choice of the rewarding action worse. This will make the $V$-value estimate even worse than that in classic MDPs.

To address these new challenges and capture the effects between two learning layers, our **idea** is to first analyze each layer separately conditioned on a fixed error from the other layer. Then, based on the resulting bound, which could be a random value, we further characterize the expected total gap due to the bias from each layer. Finally, by combining these gap together and taking the expectation over all possible past realizations, we obtain the final regret upper-bound.

### E.2.1 STEP-1: CONNECT THE STOCHASTIC OBSERVED REWARD TO THE SUB-REGRET IN THE FIRST LEARNING LAYER

In this step, we upper-bound the sub-regret due to the sub-optimiality in the first learning layer. First, according to the update of the weights in Eq. (2), the property in Lemma 1 below holds. Please see Appendix E.3 for the proof of Lemma 1.

**Lemma 1.** *At each step $h$ of episode $k$, we must have*

$$\frac{1}{\eta_1} \ln \left( \frac{\sum_{i=1}^{d} w^{k+1}(i)}{\sum_{i=1}^{d} w^k(i)} \right) \leq \frac{1}{d(1-\eta_1)} \sum_{i=1}^{d} \sum_{h=1}^{H} r_{h,i}^k + \frac{H(e-2)\eta_1}{d^2(1-\eta_1)} \sum_{i=1}^{d} \sum_{h=1}^{H} r_{h,i}^k / p^k(i), \quad (38)$$

*where $r_{h,i}^k \triangleq r_h(\phi_i(s_h^k), a_h^k)$.*

Note that the left-hand side of Eq. (38) captures the evolution of the weights in the logarithmic scale. The first term on the right-hand side of Eq. (38) represents the parameterized stochastic observed total reward in episode $k$. The second term on the right-hand side of Eq. (38) represents a reduced version of the variance of the observed reward.

Next, for the sum of both sides of Eq. (38) over all episodes $k$, we take the expectation with respect to the randomness of the algorithm, including the randomness of the query policy and the action policy, and the state transition. Then, by rearranging the terms, the expected cumulative reward of OP-TLL over all episodes $k$ can be lower-bounded as follows,

$$\mathbb{E} \left[ \frac{1}{d(1-\eta_1)} \sum_{k=1}^{K} \sum_{i=1}^{d} \sum_{h=1}^{H} r_{h,i}^k \right]$$

$$\geq \mathbb{E} \left[ -\frac{H(e-2)\eta_1}{d^2(1-\eta_1)} \sum_{k=1}^{K} \sum_{i=1}^{d} \sum_{h=1}^{H} r_{h,i}^k / p^k(i) \right] + \mathbb{E} \left[ \frac{1}{\eta_1} \sum_{k=1}^{K} \ln \left( \frac{\sum_{i=1}^{d} w^{k+1}(i)}{\sum_{i=1}^{d} w^k(i)} \right) \right]. \quad (39)$$

Since $0 \leq r_{h,i}^k \leq 1$ for all $h$ and $k$, according to the telescoping sum, we have

$$\mathbb{E} \left[ \frac{1}{d(1-\eta_1)} \sum_{k=1}^{K} \sum_{i=1}^{d} \sum_{h=1}^{H} r_{h,i}^k \right] \geq -\frac{H(e-2)\eta_1}{d^2(1-\eta_1)} KHd + \mathbb{E} \left[ \frac{1}{\eta_1} \cdot \left( \frac{\eta_1}{d} \sum_{k=1}^{K} \sum_{h=1}^{H} \hat{r}_{h,i_h^k}^k - \ln d \right) \right]. \quad (40)$$

where $\hat{r}_{h,i_h^k}^k = \hat{r}_h^k(\phi_{i_h^k}(s_h^k), a_h^k)$, and the expectation is taken with respect to the randomness of the algorithm, including the query policy and the action policy, and the state transition. Thus, we have

$$\mathbb{E} \left[ \sum_{k=1}^{K} \sum_{i=1}^{d} \sum_{h=1}^{H} r_{h,i}^k \right] \geq -\frac{H\eta_1}{d} KHd + \mathbb{E} \left[ \frac{d(1-\eta_1)}{\eta_1} \cdot \left( \frac{\eta_1}{d} \sum_{k=1}^{K} \sum_{h=1}^{H} \hat{r}_{h,i_h^k}^k - \ln d \right) \right], \quad (41)$$

Finally, we can upper-bound the conditional difference between the stochastic observed reward of the our OP-TLL and that of the optimal policy as follows,

$$\mathbb{E} \left[ \sum_{k=1}^{K} \sum_{h=1}^{H} \hat{r}_{h,i_h^k}^k \right] - \mathbb{E} \left[ \sum_{k=1}^{K} \sum_{i=1}^{d} \sum_{h=1}^{H} r_{h,i}^k \right] \leq 2H^2 \eta_1 K + \frac{d \ln d}{\eta_1}, \quad (42)$$

where the expectation is taken with respect to the randomness of the algorithm, including the query policy and the action policy, and the state transition. Note that we now successfully connect the stochastic observed reward to the sub-regret in the first learning layer. However, the **new difficulty** here is that the reward is not the observed one with noise, i.e., not the true optimal reward that we can obtain in each episode. This is mainly because of the sub-optimality in the second learning layer for collecting the partial OSI and estimating the $V$-value function.

### E.2.2    STEP-2: CONNECT THE SUB-REGRET IN THE SECOND LEANING LAYER TO THE WEIGHT ESTIMATION IN THE FIRST LEARNING LAYER

In this step, we **address** the problem that we mentioned at the end of Appendix E.2.1. We first focus on the first term on the left-hand side of Eq. (42). Note that to upper-bound the final regret, we need to capture the gap due to the imperfect reward used for updating weights in Eq. (2). To achieve this,

we note that the expected difference between the best achievable reward and the observed reward can be connected as follows,

$$\mathbb{E}\left[\sum_{k=1}^{K}\sum_{h=1}^{H}\hat{r}^*_{h,i^k_h}\right] - \mathbb{E}\left[\sum_{k=1}^{K}\sum_{h=1}^{H}\hat{r}^k_{h,i^k_h}\right] = \mathbb{E}\left[\sum_{k=1}^{K}(\sum_{h=1}^{H}\hat{r}^*_{h,i^k_h} - V^k_h)\right] + \mathbb{E}\left[\sum_{k=1}^{K}(V^k_h - \sum_{h=1}^{H}\hat{r}^k_{h,i^k_h})\right].$$
(43)

where $\hat{r}^*_{h,i^*} = \hat{r}^k_{i^*_h}(\phi_{i^*_h}(s^k_h), a^k_h)$, and the expectation is taken with respect to the randomness of the algorithm, including the query policy and the action policy, and the state transition. To upper-bound the first term on the right-hand side of Eq. (43), below we provide an important lemma, i.e., Lemma 2, that is proved to be useful later.

**Lemma 2.** *In each episode $k$, for any $\delta \in (0,1)$, with probability at least $1-\delta$, we have*

$$\mathbb{E}\left[\sum_{h'=h}^{H}\hat{r}^*_{h',i^*_{h'}}\right] \leq V^k_h,$$
(44)

*where $V^k_h$ is the $V$-value estimated by OP-TLL in Algorithm 1.*

Please see the proof of Lemma 2 in Appendix E.4. Note that in Lemma 2, the term on the left-hand side of Eq. (44) is the best possible expected observed reward, which is equivalent to the expected residual reward obtained by the optimal policy. Moreover, the term on the right-hand side of Eq. (44) is the estimated $V$-value at step $h$ of episode $k$. According to Eq. (44), we have

$$\mathbb{E}\left[\sum_{k=1}^{K}\sum_{h=1}^{H}\hat{r}^*_{h,i^*_h}\right] - \mathbb{E}\left[\sum_{k=1}^{K}\sum_{h=1}^{H}\hat{r}^k_{h,i^k_h}\right] \leq \mathbb{E}\left[\sum_{k=1}^{K}(V^k_h - \sum_{h=1}^{H}\hat{r}^k_{h,i^k_h})\right].$$
(45)

Thus, we can focus on upper-bounding the difference between the estimated expected reward, i.e., $V^k_h$, and the expected reward after taking the action $a^k_h$. Note that in Lemma 2, we have related the expected reward used in the first learning layer to the $V$-value estimate in the second learning layer of Algorithm 1. After relating the expected reward $\mathbb{E}[\sum_{h'=h}^{H}r^*_{h',i^*_{h'}}]$ used in the first learning layer to the V-value estimate $V^k_h$ in the second learning layer, we can focus on the difference between the estimated V-value $V^k_h$ and the true reward $r^k_{h,i^k_h}$ under the randomness of the queries on the sub-states $i^k_h$. Then, conditioned on the $\sigma$-algebra generated by the observation history, such a difference in each episode $k$ can be decomposed into the sub-differences $V^k_{h,i} - \mathbb{E}[\sum_{h'=h}^{H}r^k_{h',i}]$ resulting from each sub-state $i$ under the randomness of observing sub-state $i$. The corresponding sub-regret $V^k_{h,i} - \mathbb{E}[\sum_{h'=h}^{H}r^k_{h',i}]$ due to each sub-state $i$ is equivalent to the sub-regret $V^k_h(s^k_h) - V^{\pi^k}_h(s^k_h)$ (i.e., the term inside the summation on the left-hand-side of Proposition 3) in the tabular MDP case studied in Azar et al. (2017).

**Proposition 3.** *(Lemma 14 in Azar et al. (2017)): With probability $1-\delta$, we have*

$$\sum_{k=1}^{K}\left[V^k_1(s^k_1) - V^{\pi^k}_1(s^k_1)\right] \leq O\left(H^{3/2}\sqrt{SAK}\ln\left(\frac{H^2SAK}{\delta}\right) + H^2S^2A\left(\ln\left(\frac{H^2SAK}{\delta}\right)\right)^2\right).$$
(46)

Thus, the sum of the sub-regret over all episodes $k$, i.e., $\sum_{k=1}^{K}\left\{V^k_{h,i} - \mathbb{E}[\sum_{h'=h}^{H}r^k_{h',i}]\right\}$, can be upper-bounded by applying Proposition 3. Thus, we have that

$$\mathbb{E}\left[\sum_{k=1}^{K}(V^k_h - \sum_{h=1}^{H}r^k_h)\right]$$

$$\leq O\left(H^{3/2}\sqrt{d|\tilde{\mathbb{S}}|AK}\ln\left(\frac{H^2|\tilde{\mathbb{S}}|AK}{\delta}\right) + H^2\sqrt{d}S^2A\left(\ln\left(\frac{H^2|\tilde{\mathbb{S}}|AK}{\delta}\right)\right)^2\right).$$
(47)

### E.2.3 STEP-3: UPPER-BOUND THE FINAL REGRET

By connecting the stochastic observed reward to the sub-regret in the first learning layer in step-1 and connecting the sub-regret in the second learning layer to the weight estimation in the first learning layer in step-2, we have finally upper-bounded the sub-regrets in both layers due to the errors from the other layer. Thus, by combining Eq. (42), Eq. (43), Eq. (45) and Eq. (47), we have that, with probability $1 - \delta$, the regret of OP-TLL is upper-bounded as follows,

$$Reg^{\text{OP-TLL}}(K)/H$$

$$\leq O\left(H^2\eta_1 K + \frac{d\ln d}{\eta_1} + H^{3/2}\sqrt{d|\tilde{\mathbb{S}}|AK}\ln\left(\frac{H^2|\tilde{\mathbb{S}}|AK}{\delta}\right) + H^2\sqrt{d}S^2 A\left(\ln\left(\frac{H^2|\tilde{\mathbb{S}}|AK}{\delta}\right)\right)^2\right)$$

$$= O\left(H\sqrt{dK\ln d} + H^{\frac{3}{2}}\sqrt{d|\tilde{\mathbb{S}}|AK}\ln\frac{H^2|\tilde{\mathbb{S}}|AK}{\delta} + H^2\sqrt{d}|\tilde{\mathbb{S}}|^2 A\left(\ln\frac{H^2|\tilde{\mathbb{S}}|AK}{\delta}\right)^2\right)$$

$$= O\left(H\sqrt{dK\ln d} + AH^2\sqrt{d}|\tilde{\mathbb{S}}|^2\sqrt{K}\left(\ln\frac{AH^2|\tilde{\mathbb{S}}|K}{\delta}\right)^2\right), \tag{48}$$

where the first equality is by taking $\eta_1 = O\left(\sqrt{\frac{d\ln d}{H^2 K}}\right)$.  $\qquad\square$

### E.3 PROOF OF LEMMA 1

*Proof.* First, according to the weight updates in Eq. (2), we have

$$\frac{1}{\eta_1}\ln\left(\frac{\sum_{i=1}^{d} w^{k+1}(i)}{\sum_{i=1}^{d} w^k(i)}\right)$$

$$= \frac{1}{\eta_1}\ln\left(\frac{\sum_{i=1}^{d} w^k(i)\cdot e^{\frac{\eta_1}{d}\sum_{h=1}^{H}\hat{r}_h^k(\phi_{i_h^k}(s_h^k),a_h^k)}}{\sum_{i=1}^{d} w^k(i)}\right). \tag{49}$$

Next, since $e^x \leq 1 + x + (e-2)x^2$ for all $x \leq 1$, according to the update of the probability $p^k(i)$ in Eq. (2), we have

$$\frac{1}{\eta_1}\ln\left(\frac{\sum_{i=1}^{d} w^{k+1}(i)}{\sum_{i=1}^{d} w^k(i)}\right)$$

$$\leq \frac{1}{\eta_1}\ln\left(1 + \frac{\eta_1}{d(1-\eta_1)}\sum_{i=1}^{d} p^k(i)\sum_{h=1}^{H} r_{h,i}^k/p^k(i) + \frac{(e-2)\eta_1^2}{d^2(1-\eta_1)}\sum_{i=1}^{d} p^k(i)\left(\sum_{h=1}^{H} r_{h,i}^k/p^k(i)\right)^2\right)$$

$$\leq \frac{1}{\eta_1}\ln\left(1 + \frac{\eta_1}{d(1-\eta_1)}\sum_{i=1}^{d}\sum_{h=1}^{H} r_{h,i}^k + \frac{(e-2)\eta_1^2}{d^2(1-\eta_1)}\sum_{i=1}^{d}\left(\sum_{h=1}^{H} r_{h,i}^k\right)^2/p^k(i)\right)$$

$$\leq \frac{1}{\eta_1}\ln\left(1 + \frac{\eta_1}{d(1-\eta_1)}\sum_{i=1}^{d}\sum_{h=1}^{H} r_{h,i}^k + \frac{H(e-2)\eta_1^2}{d^2(1-\eta_1)}\sum_{i=1}^{d}\sum_{h=1}^{H} r_{h,i}^k/p^k(i)\right). \tag{50}$$

Then, since $\ln(1+x) \leq x$ for all $x$, we have

$$\frac{1}{\eta_1} \ln\left(\frac{\sum_{i=1}^{d} w^{k+1}(i)}{\sum_{i=1}^{d} w^k(i)}\right) \leq \frac{1}{d(1-\eta_1)} \sum_{i=1}^{d} \sum_{h=1}^{H} r_{h,i}^k + \frac{H(e-2)\eta_1}{d^2(1-\eta_1)} \sum_{i=1}^{d} \sum_{h=1}^{H} r_{h,i}^k / p^k(i). \qquad (51)$$

□

### E.4 PROOF OF LEMMA 2

*Proof.* The proof of Lemma 2 mainly utilizes a set of concentration inequalities Azar et al. (2017). Specifically, we prove Lemma 2 by mathematical induction.

Base case: when $h = H$, Eq. (44) trivially holds, since according to the definition of $V_H^k$, we have $\mathbb{E}[r_H^*] \leq V_H^k$. Note that the reward $r_{h'}^* = r_{h'}(\hat{\phi}_{i_{h'}^*}(s_{h'}), a_{h'})$ is a function of the query $\hat{i}_{h'}^*$.

Induction: we hypothesize that $\mathbb{E}\left[\sum_{h'=h+1}^{H} r_{h'}^*\right] \leq V_{h+1}^k$. Then, we focus on proving that $\mathbb{E}\left[\sum_{h'=h}^{H} r_{h'}^*\right] \leq V_h^k$. Note that, we have

$$V_h^k - \mathbb{E}\left[\sum_{h'=h}^{H} r_{h'}^*\right] \geq \mathbb{P}_h^{k,*} V_{h+1}^k(s) - \mathbb{P}_h^* V_{h+1}^k(s) + b_{h+1}^k(s,a), \qquad (52)$$

where $b_{h+1}^k(s,a)$ is the bonus term in Eq. (62). By adding and subtracting some middle terms, we have that

$$V_h^k - \mathbb{E}\left[\sum_{h'=h}^{H} r_{h'}^*\right] \geq \mathbb{P}_h^{k,*}\left[V_{h+1}^k(s) - V_{h+1}^*(s)\right] + \left[\mathbb{P}_h^{k,*} - \mathbb{P}_h^*\right] V_{h+1}^*(s) + b_{h+1}^k(s,a). \qquad (53)$$

According to induction hypothesis that $\mathbb{E}\left[\sum_{h'=h+1}^{H} r_{h'}^*\right] \leq V_{h+1}^k$, we have

$$V_h^k - \mathbb{E}\left[\sum_{h'=h}^{H} r_{h'}^*\right] \geq \left[\mathbb{P}_h^{k,*} - \mathbb{P}_h^*\right] V_{h+1}^*(s) + b_{h+1}^k(s,a). \qquad (54)$$

Finally, according to the empirical Bernstein's inequality, we have that with probability $1 - \delta$,

$$\left[\mathbb{P}_h^* - \mathbb{P}_h^{k,*}\right] V_{h+1}^*(s) \leq b_{h+1}^k(s,a). \qquad (55)$$

Combining Eq. (54) and Eq. (55), we have $\mathbb{E}\left[\sum_{h'=h}^{H} r_{h'}^*\right] \leq V_h^k$. □

## F PROOF OF THEOREM 6

In this section, we first provide the complete version of our new algorithm, OP-MLL, for the case with $\tilde{d} > 1$. Please see Algorithm 4. Then, we present our complete proof for the regret of our algorithm, i.e., Theorem 6. We use $mod(k, x)$ to denote the remainder when $k$ is divided by $x$ and let $\kappa = \lceil (d-1)/(\tilde{d}-1) \rceil$. In this case, after choosing the queried sub-states, since the number of queried sub-states is larger than 1, then agent needs to pick one of the sub-states as the rewarding sub-state. That is, at each step, (step-i) the agent actively selects sub-states $\hat{i}_h^k$ to query, and receives the partial OSI $\{\phi_i(s_h^k)\}_{\{i \in \hat{i}_h^k\}}$; (step-ii) the agent choose one of the queried sub-states as the rewarding state $\phi_{i_h^k}(s_h^k)$; (step-iii) the agent takes the action $a_h^k$ and receives the reward $r_h(\phi_{i_h^k}(s_h^k), a_h^k)$; (step-iv) the next state $s_{h+1}^k$ is drawn according to probability $\mathbb{P}_h(\cdot|s_h^k, a_h^k) = \prod_{i=1}^{d} \mathbb{P}_{h,i}(\phi_i(\cdot)|\phi_i(s_h^k), a_h^k)$, where the product form indicates that the sub-states have independent transition kernels.

---

**Algorithm 4** Optimistic-Pessimistic Multi-Layer Learning (OP-MLL)

---

Initialization: $w^1(i) = 1$ and $p^1(i) = \frac{1}{d}$ for all $i = 1, ..., d$; $\kappa = \left\lceil \frac{d-1}{\tilde{d}-1} \right\rceil$.

**for** $k = 1 : K$ **do**

  **if** $mod(k, \kappa) = 1$ **then**

    *Step-1:* update the global weights $w^k(i)$ and probabilities $p^k(i)$ as follows,

$$w^k(i) = w^{k-\kappa}(i) \cdot e^{\frac{(d-1)\eta_1}{d(\tilde{d}-1)} \sum_{\tau=k-\kappa}^{k-1} \sum_{h=1}^{H} \hat{r}_h^\tau(\phi_i(s_h^\tau), a_h^\tau)},$$

$$p^k(i) = (1 - \eta_1) \frac{w^k(i)}{\sum_{i'=1}^{d} w^k(i')} + \frac{\eta_1}{d}, \tag{56}$$

    where

$$\hat{r}_h^\tau(\phi_i(s_h^{k-1}), a_h^{k-1}) = r_h^\tau(\phi_i(s_h^{k-1}), a_h^{k-1}) - r_h^\tau(\phi_{i_h^\tau}(s_h^{k-1}), a_h^{k-1}), \text{ if } i \neq i_h^\tau,$$

$$\text{and } \hat{r}_h^\tau(\phi_i(s_h^{k-1}), a_h^{k-1}) = 0, \text{ otherwise}, \tag{57}$$

    *Step-2:* choose the leading sub-state, i.e., the leader, according to the updated global probability $p^k(i)$.

    *Step-3:* initialize the local weight $\tilde{w}^k(i)$ according to the global weight $w^k(i)$, i.e., $\tilde{w}^k(i) = w^k(i)$.

  **end if**

  *Step-4:* choose $\tilde{d} - 1$ supporting sub-states, i.e., the follower, uniformly randomly from the sub-states that have not yet been chosen in most-recent $\kappa$ episodes, i.e., from $\left\lfloor \frac{k-1}{\kappa} \right\rfloor \cdot \kappa + 1$ to $(\lfloor \frac{k-1}{\kappa} \rfloor + 1) \cdot \kappa$.

  *Step 5:* update the local weights $\tilde{w}^k(i)$ and probabilities $\tilde{p}^k(i)$ for sub-state $i$ queried as follows,

$$\tilde{w}^k(i) = \tilde{w}^{k-1}(i) \cdot e^{\frac{\eta_2}{\tilde{d}} \sum_{h=1}^{H} r_h(\phi_i(s_h^{k-1}), a_h^{k-1})},$$

$$\tilde{p}^k(i) = (1 - \eta_2) \frac{\tilde{w}^k(i)}{\sum_{i' \in \hat{i}^k} \tilde{w}^k(i')} + \frac{\eta_2}{\tilde{d}}. \tag{58}$$

  *Step-6:* choose the rewarding sub-state according to the updated local probability $\tilde{p}^k(i)$.

  **for** $h = H : 1$ **do**

    *Step-7:* update the $Q$-value function as follows,

$$Q_h^k(\phi_i(s), a) = \min \left\{ r_h(\phi_i(s), a) + [\mathbb{P}_h^k V_{h+1}^k](\phi_i(s), a) + O\left(\sqrt{\frac{H^2}{\mathcal{N}_h^k(\phi_i(s), a)}}\right), H \right\}, \tag{59}$$

    where $\mathbb{P}_h^k(\phi_i(s')|\phi_i(s), a) = \frac{\mathcal{N}_h^k(\phi_i(s), a, \phi_i(s'))}{\mathcal{N}_h^k(\phi_i(s), a)}$ is the estimated transition kernel, $\mathcal{N}_h^k(\phi_i(s), a)$ and $\mathcal{N}_h^k(\phi_i(s), a, \phi_i(s'))$ are the number of times $(\phi_i(s), a)$ and $(\phi_i(s), a, \phi_{\hat{i}}(s'))$ have been visited at step $h$ up to episode $k$, respectively, and $V_h^k(\phi_i(s)) = \max_a Q_h^k(\phi_i(s), a)$ is the estimated $V$-value function.

  **end for**

  **for** $h = 1 : H$ **do**

    *Step-8:* take an action $a_h^k$ that maximizes the updated $Q$-value function, and collect the partial OSI.

  **end for**

**end for**

---

## F.1 OPTIMISTIC-PESSIMISTIC MULTI-LAYER LEARNING (OP-MLL)

**Idea-I (Determine a leading sub-state pessimistically):** The key insight in our idea-I is that, although the state-transition and reward are stochastic, an existing stochastic learning method (such as upper confidence interval) does not apply in our case. This is because the choice of the sub-state set $\hat{i}$

will finally affect the $V$-value estimation, and thus needs to be handled in a more conservative manner. Thus, in the first learning layer (i.e., Step-1 and Step-2 of Algorithm 4), OP-MLL determines a leading sub-state pessimistically. Specifically, at the beginning of every $\kappa$ episodes, OP-MLL updates the *global* weights and probabilities for each sub-state $\phi_i(s)$ according to our new exponential weighting method as follows (and chooses the leading sub-state according to the updated $p^k(i)$),

$$w^k(i) = w^{k-\kappa}(i) \cdot e^{\frac{(d-1)\eta_1}{d(\tilde{d}-1)} \sum_{\tau=k-\kappa}^{k-1} \sum_{h=1}^{H} \hat{r}_h^\tau(\phi_i(s_h^\tau), a_h^\tau)}, \text{ and } p^k(i) = (1-\eta_1)\frac{w^k(i)}{\sum_{i'=1}^{d} w^k(i')} + \frac{\eta_1}{d}, \quad (60)$$

where $\eta_1$ is a key parameter that determines how pessimistic the algorithm is, e.g., with smaller $\eta_1$, the global weight increases more slowly and thus the algorithm behaves more pessimistically. Note that the first term in $p^k(i)$ captures how important the sub-state $\phi_i(s)$ is among all sub-states, and the second term is a uniform distribution (parameterized by $\eta_1$) that allows the learner to keep exploiting different sub-states. An important step in our idea-I is how to get the correct factor (i.e., $\frac{(d-1)\eta_1}{d(\tilde{d}-1)}$) for $w^k(i)$. We explain it in idea-II below, since it is tuned based on both our idea-I and idea-II.

**Idea-II (Choose the supporting sub-states based on the selections in most-recent episodes):** The key insight in our idea-II is that, due to the bias resulting from partial OSI, the standard importance sampling method (e.g., selecting the supporting sub-states $\hat{i}$ with highest weight $\sum_{i\in\hat{i}} w^k(i)$) does not apply in our case. Thus, instead, OP-MLL chooses $\tilde{d}-1$ supporting sub-states uniformly randomly from the sub-states that have not yet been chosen in most-recent episodes from $\lfloor \frac{k-1}{\kappa} \rfloor \cdot \kappa + 1$ to $(\lfloor \frac{k-1}{\kappa} \rfloor + 1) \cdot \kappa$ (i.e., Step-3 in Algorithm 4). As a result, conditioned on the leading sub-state (and independent of the episode), each sub-state is chosen with probability $\frac{\tilde{d}-1}{d-1}$, which results in the factor $\frac{\eta_1}{d\frac{\tilde{d}-1}{d-1}} = \frac{(d-1)\eta_1}{d(\tilde{d}-1)}$ in the global weight. Interestingly, in the regret analysis, we prove that in this way, the unnecessary bias (due to partial OSI) that could result in a linear-in-$K$ regret can be avoided.

**Idea-III (Choose the rewarding sub-state and action optimistically):** In the third layer, OP-MLL chooses the rewarding sub-state and the final action in an optimistic manner. Specifically, in Step-4, OP-MLL updates the *local* weights $\tilde{w}^k(i)$ and probabilities $\tilde{p}^k(i)$ for the sub-states $\phi_i(s)$ that was chosen in the current set $\hat{i}^k$ as follows,

$$\tilde{w}^k(i) = \tilde{w}^{k-1}(i) \cdot e^{\frac{\eta_2}{d} \sum_{h=1}^{H} r_h(\phi_i(s_h^{k-1}), a_h^{k-1})}, \text{ and } \tilde{p}^k(i) = (1-\eta_2)\frac{\tilde{w}^k(i)}{\sum_{i'\in\hat{i}^k} \tilde{w}^k(i')} + \frac{\eta_2}{\tilde{d}}. \quad (61)$$

There are three differences between the global update in Eq. (60) and the local update in Eq. (61). First, the global weight $w^k(i)$ is updated based on the weight $w^{k-\kappa}(i)$ that is $\kappa$ episodes earlier, while the local weight $\tilde{w}^k(i)$ is updated based on the weight $w^{k-1}(i)$ that is 1 episodes earlier and parameterized by an important different $\eta_2$. To make the algorithm more optimistic *locally* to achieve larger reward, the value of $\eta_2$ should be larger than the value of $\eta_1$. Indeed, Theorem 6 below provides a sufficient condition on how $\eta_2$ should be larger than $\eta_1$. Second, the factor $\frac{d-1}{\tilde{d}-1}$ does not appear in Eq. (61), because the local weight is updated for only the sub-states in the chosen set $\hat{i}^k$. Third, since the local probability $\tilde{p}^k(i)$ is also only for the sub-states in $\hat{i}^k$, the denominator in the first term on the right-hand side of the second equation in Eq. (61) is summing only over $i \in \hat{i}^k$. Finally, in Step-5 and Step-6, OP-MLL takes an action that maximizes the $Q$-value function following the optimism-in-face-of-uncertainty principle (please see Appendix F for details),

$$Q_h^k(\phi_i(s), a) = \min\{r_h(\phi_i(s), a) + [\mathbb{P}_h^k V_{h+1}^k](\phi_i(s), a) + O(\sqrt{H^2/\mathcal{N}_h^k(\phi_i(s), a)}), H\}. \quad (62)$$

**Theorem 6.** *(Regret) For POMDPs with partial online state information and independent sub-states, by choosing $\eta_1 = \tilde{O}(1/\sqrt{K})$ and $\eta_2 = \frac{16(d-1)}{\tilde{d}-1}\eta_1$, with probability $1-\delta$, the regret $Reg^{OP\text{-}MLL}(K)$ of OP-MLL can be upper-bounded by*

$$\tilde{O}\left(H^{\frac{5}{2}}\sqrt{\frac{d|\tilde{\mathbb{S}}|AK}{\tilde{d}-1}} \ln\frac{H^3|\tilde{\mathbb{S}}|AK}{\delta} + H^2\sqrt{\frac{d}{\tilde{d}-1}}|\tilde{\mathbb{S}}|^2 A \left(\ln\frac{H^2|\tilde{\mathbb{S}}|AK}{\delta}\right)^2 + H^2\sqrt{\frac{dK\ln d}{\tilde{d}-1}}\right).$$
$$(63)$$

Theorem 6 shows that (i) the regret of OP-MLL depends *polynomially* on all problem parameters $A$, $H$, $|\tilde{\mathbb{S}}|$ and $d$; (ii) the regret of OP-MLL *decreases further* as $\tilde{d}$ increases; (iii) the dependency on $K$ is $\tilde{O}(\sqrt{K})$, which *cannot* be further improved. *To the best of our knowledge, this is the first such near-optimal result for POMDPs without full OSI.* The main difficulties in the proof of Theorem 6 mainly result from the effects between every two of the three learning layers. For example, in the first and second layers, two different weight parameters $\eta_1$ and $\eta_2$ are used for adapting to the change of the cumulative reward resulting from the third learning layer. Thus, in order to connect the local regret in an episode to the global regret across episodes, we leverage the second-order and third-order Taylor expansions of the function $e^x$. Please see the other difficulties and more details in Appendix F.

### F.2 Proof of Theorem 6

*Proof.* The **main challenges** in the proof of Theorem 6 result from the mutual effects among the three learning layers. Specifically, **first,** note that in Algorithm 4, the first learning layer is affected by the supporting sub-states chosen in the second learning layer, and the chosen action and $V$-value function estimates in the third learning layer. For example, in the first learning layer, the weight $w^k(i)$ at each episode is updated based on the reward received in the last episode, which further depends on the supporting sub-states chosen in the second learning layer, and the chosen action and $V$-value function estimates in the third learning layer. **Second,** note that the supporting sub-states chosen in the second learning layer is affected by the leading sub-state chosen in the first learning layer and the reward received from the third learning layer. **Third,** note that the chosen action and $V$-value estimates in the third learning layer depends on the feedback collected from the determined leading sub-state and the chosen supporting sub-states, which further depends on the first learning layer and second learning layer.

To address these new challenges and capture such effects between every two learning layers, our **ideas** are to first analyze each layer separately by assuming a fixed error from the other layer. Then, based on the resulting bound, we characterize the further gap due to the bias from each layer. Finally, by carefully combining these gap together using the first and second order properties of the reward, we get the final regret upper-bound.

#### F.2.1 Step-1: Connect the Sub-Regret in the Second Learning Layer to the Choice in the First Learning Layer

In this step, we focus on capturing the effect from the first learning layer on the sub-regret in the second learning layer. Specifically, **differently** from our proof for the simpler case with $\tilde{d} = 1$ in Appendix E, our OP-MLL algorithm for the case with $\tilde{d} > 1$ selects the supporting sub-states for $\hat{i}^k$ based on the choice of the leading sub-state determined in the first learning layer. Thus, we need to first upper-bound the sub-regret here. First, for every $\kappa$ episodes, we can lower-bound the expected observed reward as in Lemma 3 below.

**Lemma 3.** *For each episode $k$, we have that the expected observed reward can be lower-bounded as follows,*

$$\mathbb{E}\left[\sum_{i\in\hat{i}^k}\sum_{h=1}^{H} r_{h,i}^k\right] \geq \frac{\tilde{d}(1-\eta_2)}{\eta_2}\mathbb{E}\left[\ln\left(\frac{\sum\limits_{i\in\hat{i}^k}\tilde{w}^{k+1}(i)}{\sum\limits_{i\in\hat{i}^k}\tilde{w}^k(i)}\right)\right] - \frac{H^2\eta_2}{\tilde{d}}. \tag{64}$$

*where $r_{h,i}^k \triangleq r_h(\phi_i(s_h^k), a_h^k)$, and the expectation is taken with respect to the randomness of the algorithm, including the query policy and the action policy, and the state transition..* Note that in Lemma 3, the term on the left-hand side of Eq. (64) is the expected reward observed by the agent. Moreover, the first term on the right-hand side of Eq. (64) captures the evolution of the weights and the second term is the gap. Please see Appendix F.3 for the proof of Lemma 3.

Note that the next **difficulty** to upper-bound the sub-regret from the second learning layer is that the weight $\tilde{w}^k(i)$ is the local weight updated according the acceleration parameter $\eta_2$ that we constructed to let the algorithm learn the reward and sub-state transitions faster. However, to upper-bound the sub-regret, we still need to convert it back to the global weight $w^k(i)$ that is directly related to the

whole episode horizon and the true reward. To resolve this difficulty, we connect the logarithmic term on the right-hand side of Eq. (64), which contains the local weight $\tilde{w}^k(i)$, to the first and second order moments of the reward as in Lemma 4 below.

**Lemma 4.** *For each step $h$ of each episode $k$, we have*

$$\mathbb{E}\left[\ln\left(\frac{\sum_{i\in\hat{i}^k}\tilde{w}^{k+1}(i)}{\sum_{i\in\hat{i}^k}\tilde{w}^k(i)}\right)\right] \geq \mathbb{E}\left[\sum_{i\in\hat{i}^k}\frac{\eta_2^2}{4\tilde{d}^2}\left(\left(\sum_{h=1}^H r_{h,i}^k - \mathbb{E}[\sum_{h=1}^H r_{h,i}^k]\right)^2\right)\right] + \frac{\eta_2}{\tilde{d}}\mathbb{E}\left[\sum_{i\in\hat{i}^k}\sum_{h=1}^H r_{h,i}^k\right],$$
(65)

*where the expectation is taken with respect to the randomness of the algorithm, including the query policy and the action policy, and the state transition.*

Note that Lemma 4 provides a lower bound for the weight-evolution term on the right-hand side of Eq. (64). Specifically, the first term on the right-hand side of Eq. (65) is related to the second order moment of the observed reward. It is essentially the variance of the total observed reward in episode $k$ conditioned on the realization before episode $k$. This term is critical for further connecting the reward to the global weights $w^k(i)$ updated using the parameter $\eta_1$ for every $\kappa$ episodes. The second term on the right-hand side of Eq. (65) is related to the expectation of the observed reward. This is another critical part that guarantees that the sub-regret can be related to the global weights, since the expected observed reward with respect to the randomness of the sub-state set $\hat{i}^k$ is equal to the expected true reward. Please see Appendix F.4 for the proof of Lemma 4.

### F.2.2   STEP-2: CONNECT THE SUB-REGRET IN THE FIRST TWO LEARNING LAYERS TO THE EXPECTED REWARD IN THE THIRD LEARNING LAYER

Note that another **challenge** to upper-bound the regret is that the the set $\hat{i}^k$ depends on the episode $k$ and changes randomly. To resolve this problem, we connect the local set $\hat{i}^k$ to the global set $\{1, ..., d\}$ of sub-states by using the conclusion in Lemma 5 below.

**Lemma 5.** *For each episode, we have*

$$\mathbb{E}\left[\sum_{i\in\hat{i}^k}\left(\left(\sum_{h=1}^H r_{h,i}^k - \mathbb{E}[\sum_{h=1}^H r_{h,i}^k]\right)^2\right)\right] \geq \frac{\tilde{d}-1}{d-1}\mathbb{E}\left[\sum_{i=1}^d\left(\left(\sum_{h=1}^H r_{h,i}^k - \mathbb{E}[\sum_{h=1}^H r_{h,i}^k]\right)^2\right)\right],$$
(66)

*where the expectation is taken with respect to the randomness of the algorithm, including the query policy and the action policy, and the state transition.*

Note that Lemma 5 connects the expected variance of the observed reward (i.e., the left-hand side of Eq. (66)) with respect to the sub-state set $\hat{i}^k$ to the variance of the true reward (i.e., the right-hand side of Eq. (66)). Please see Appendix F.5 for the proof of Lemma 5. Moreover, similar to the proof of Lemma 3 in Appendix F.3, we can show Lemma 6 below.

**Lemma 6.** *For each episodes, we have that*

$$\mathbb{E}\left[\sum_{k=(u-1)\kappa+1}^{u\kappa}\sum_{h=1}^H r_h^k\right] \geq \frac{d(1-\eta_1)}{\eta_1}\mathbb{E}\left[\ln\left(\frac{\sum_{i=1}^d w^{u\kappa}(i)}{\sum_{i=1}^d w^{(u-1)\kappa+1}(i)}\right)\right] - \frac{H^2\eta_1}{d},$$
(67)

*where the expectation is taken with respect to the randomness of the algorithm, including the query policy and the action policy, and the state transition.*

Next, by connecting the variance of the true reward on the right-hand side of Eq. (66) to the global weights $w^k(i)$, we connect the sub-regret in the first two learning layers to the expected reward in

the third learning layer. To this end, we prove that the evolution of the global weights satisfies the following inequality,

$$
\mathbb{E}\left[\ln\left(\frac{\sum\limits_{i=1}^{d} w^{k+1}(i)}{\sum\limits_{i=1}^{d} w^k(i)}\right)\right]
$$
$$
\leq \mathbb{E}\left[\sum_{i=1}^{d} \frac{(\eta_1(d-1))^2}{(d(\tilde{d}-1))^2}\left(\left(\sum_{h=1}^{H} r_{h,i}^k - \mathbb{E}[\sum_{h=1}^{H} r_{h,i}^k]\right)^2\right)\right] + \frac{\eta_1(d-1)}{d(\tilde{d}-1)}\mathbb{E}\left[\sum_{i=1}^{d}\sum_{h=1}^{H} r_{h,i}^k\right], \quad (68)
$$

This is because according to the update for the global weights $w_i(k)$ in Eq. (56), we have

$$
\mathbb{E}\left[\ln\left(\frac{\sum\limits_{i=1}^{d} w^{k+1}(i)}{\sum\limits_{i=1}^{d} w^k(i)}\right)\right] = \mathbb{E}\left[\ln\left(\sum_{i=1}^{d} \frac{w^k(i)}{\sum\limits_{i=1}^{d} w^k(i)} e^{\frac{\eta_1(d-1)}{d(\tilde{d}-1)}\sum\limits_{h=1}^{H} r_{h,i}^k}\right)\right]
$$
$$
= \mathbb{E}\left[\ln\left(\sum_{i=1}^{d} \frac{w^k(i)}{\sum\limits_{i=1}^{d} w^k(i)} e^{\frac{\eta_1(d-1)}{d(\tilde{d}-1)}\sum\limits_{h=1}^{H}(r_{h,i}^k - \mathbb{E}[r_{h,i}^k])}\right)\right] + \frac{\eta_1(d-1)}{d(\tilde{d}-1)}\mathbb{E}\left[\sum_{i=1}^{d}\sum_{h=1}^{H} r_{h,i}^k\right]. \quad (69)
$$

Then, since $e^x \leq 1 + x + x^2$ for $x \leq \ln 2$, the first term on the right-hand side of Eq. (69) can be upper-bounded as follows,

$$
\mathbb{E}\left[\ln\left(\sum_{i=1}^{d} \frac{w^k(i)}{\sum\limits_{i=1}^{d} w^k(i)} e^{\frac{\eta_1(d-1)}{d(\tilde{d}-1)}\sum\limits_{h=1}^{H}(r_{h,i}^k - \mathbb{E}[r_{h,i}^k])}\right)\right]
$$
$$
\leq \mathbb{E}\left[\ln\left(1 + \mathbb{E}\left[\sum_{i=1}^{d} \frac{\eta_1(d-1)}{d(\tilde{d}-1)}\left(\sum_{h=1}^{H} r_{h,i}^k - \mathbb{E}[\sum_{h=1}^{H} r_{h,i}^k]\right)\right]\right.\right.
$$
$$
\left.\left. + \mathbb{E}\left[\sum_{i=1}^{d} \frac{(\eta_1(d-1))^2}{(d(\tilde{d}-1))^2}\left(\left(\sum_{h=1}^{H} r_{h,i}^k - \mathbb{E}[\sum_{h=1}^{H} r_{h,i}^k]\right)^2\right)\right]\right)\right]
$$
$$
\leq \mathbb{E}\left[\ln\left(1 + \mathbb{E}\left[\sum_{i=1}^{d} \frac{(\eta_1(d-1))^2}{(d(\tilde{d}-1))^2}\left(\left(\sum_{h=1}^{H} r_{h,i}^k - \mathbb{E}[\sum_{h=1}^{H} r_{h,i}^k]\right)^2\right)\right]\right)\right]. \quad (70)
$$

Since $\ln(1+x) \leq x$, we have

$$
\mathbb{E}\left[\ln\left(\sum_{i=1}^{d} \frac{w^k(i)}{\sum\limits_{i=1}^{d} w^k(i)} e^{\frac{\eta_1(d-1)}{d(\tilde{d}-1)}\sum\limits_{h=1}^{H}(r_{h,i}^k - \mathbb{E}[r_{h,i}^k])}\right)\right]
$$
$$
\leq \mathbb{E}\left[\sum_{i=1}^{d} \frac{(\eta_1(d-1))^2}{(d(\tilde{d}-1))^2}\left(\left(\sum_{h=1}^{H} r_{h,i}^k - \mathbb{E}[\sum_{h=1}^{H} r_{h,i}^k]\right)^2\right)\right]. \quad (71)
$$

Thus, Eq. (68) is true because of Eq. (69) and Eq. (71). Finally, by combining Eq. (64), Lemma 4, Eq. (66) and Eq. (68), we have

$$\mathbb{E}\left[\sum_{i\in\hat{i}^k}\sum_{h=1}^{H}r_{h,i}^k\right] \geq \mathbb{E}\left[\ln\left(\frac{\sum_{i=1}^{d}w^{k+1}(i)}{\sum_{i=1}^{d}w^k(i)}\right)\right] - \frac{H^2\eta_2}{\tilde{d}}. \tag{72}$$

Taking the expectation and the sum over all episodes for both sides of Eq. (72), we have

$$\mathbb{E}\left[\sum_{k=1}^{K}\sum_{i\in\hat{i}^k}\sum_{h=1}^{H}r_{h,i}^k\right] \geq \mathbb{E}\left[\sum_{k=1}^{K}\ln\left(\frac{\sum_{i=1}^{d}w^{k+1}(i)}{\sum_{i=1}^{d}w^k(i)}\right)\right] - \frac{H^2\eta_2 K}{\tilde{d}}$$

$$= \mathbb{E}\left[\ln\left(\frac{\sum_{i=1}^{d}w^{K}(i)}{\sum_{i=1}^{d}w^1(i)}\right)\right] - \frac{H^2\eta_2 K}{\tilde{d}}, \tag{73}$$

where the last step is because of the telescoping sum. Finally, combining Eq. (73) with the updates of the global weights in Eq. (56), we have

$$\mathbb{E}\left[\sum_{k=1}^{K}\sum_{h=1}^{H}r_h^k\right] - \mathbb{E}\left[\sum_{k=1}^{K}\sum_{i=1}^{d}\sum_{h=1}^{H}r_{h,d}^k\right] \leq \frac{H^2\eta_2 K}{\tilde{d}} + \frac{d\ln d}{\eta_1}. \tag{74}$$

### F.2.3 STEP-3: CONNECT THE EXPECTED REWARD IN THE THIRD LEARNING LAYER TO THE TRUE OPTIMAL REWARD

Note that we now connect the expected observed reward to the sub-regret in the first two learning layers. However, the **new difficulty** here is that the reward is not the true optimal reward that we can obtain in each episode. This is because of the sub-optimality in the third learning layer for estimating the $V$-value function.

Hence, in this step, we address the problem that we mentioned above. Let us focus on the first term on the left-hand side of Eq. (74). We have

$$\mathbb{E}\left[\sum_{k=1}^{K}\sum_{h=1}^{H}r_h^*\right] - \mathbb{E}\left[\sum_{k=1}^{K}\sum_{h=1}^{H}r_h^k\right] = \mathbb{E}\left[\sum_{k=1}^{K}(\sum_{h=1}^{H}r_h^* - V_h^k)\right] + \mathbb{E}\left[\sum_{k=1}^{K}(V_h^k - \sum_{h=1}^{H}r_h^k)\right]. \tag{75}$$

Below, we provide an important lemma, i.e., Lemma 7, that is proved to be useful soon later.

**Lemma 7.** *For any $\delta \in (0,1)$, with probability at least $1-\delta$, we have*

$$\mathbb{E}\left[\sum_{h'=h}^{H}r_{h',i_{h'}^*}^*\right] \leq V_h^k. \tag{76}$$

Please see Appendix F.6 for the proof of Lemma 7. Then, according to Lemma 7, we have

$$\mathbb{E}\left[\sum_{k=1}^{K}\sum_{h=1}^{H}r_{h,i_h^*}^*\right] - \mathbb{E}\left[\sum_{k=1}^{K}\sum_{h=1}^{H}r_h^k\right] \leq \mathbb{E}\left[\sum_{k=1}^{K}(V_h^k - \sum_{h=1}^{H}r_h^k)\right]. \tag{77}$$

Thus, we can focus on upper-bounding the difference between the estimated expected reward, i.e., $V_h^k$, and the expected reward after taking the action $a_h^k$. Note that in Lemma 7, we have related the expected reward used in the first two learning layers to the $V$-value estimate in the third learning layer of Algorithm 4. After relating the expected reward $\mathbb{E}[\sum_{h'=h}^{H}r_{h',i_{h'}^*}^*]$ used in the first *two* learning layers to the V-value estimate $V_h^k$ in the *third* learning layer, we can focus on the difference between the estimated V-value $V_h^k$ and the true reward $r_{h,i_h^k}^k$ under the randomness of the queries

on the sub-states $i_h^k$. Then, conditioned on the $\sigma$-algebra generated by the observation history, such a difference in each episode $k$ can be decomposed into the sub-differences $V_{h,i}^k - \mathbb{E}[\sum_{h'=h}^H r_{h',i}^k]$ resulting from each sub-state $i$ under the randomness of observing sub-state $i$. The corresponding sub-regret $V_{h,i}^k - \mathbb{E}[\sum_{h'=h}^H r_{h',i}^k]$ due to each sub-state $i$ is equivalent to the sub-regret $V_h^k(s_h^k) - V_h^{\pi^k}(s_h^k)$ in the tabular MDP case studied in Azar et al. (2017).

**Proposition 4.** *(Lemma 14 in Azar et al. (2017)): With probability $1 - \delta$, we have*

$$\sum_{k=1}^K \left[ V_1^k(s_1^k) - V_1^{\pi^k}(s_1^k) \right] \le O \left( H^{3/2}\sqrt{SAK} \ln \left( \frac{H^2 SAK}{\delta} \right) + H^2 S^2 A \left( \ln \left( \frac{H^2 SAK}{\delta} \right) \right)^2 \right). \tag{78}$$

Thus, the sum of the sub-regret over all episodes $k$, i.e., $\sum_{k=1}^K \{V_{h,i}^k - \mathbb{E}[\sum_{h'=h}^H r_{h',i}^k]\}$, can be upper-bounded by applying Proposition 3. Thus, we have that,

$$\mathbb{E}\left[ \sum_{k=1}^K (V_h^k - \sum_{h=1}^H r_h^k) \right]$$
$$\le O \left( H^{3/2}\sqrt{\frac{d}{\tilde{d}-1}|\tilde{\mathbb{S}}|AK} \ln \left( \frac{H^2|\tilde{\mathbb{S}}|AK}{\delta} \right) + H^2 \sqrt{\frac{d}{\tilde{d}-1}}|\tilde{\mathbb{S}}|^2 A \left( \ln \left( \frac{H^2|\tilde{\mathbb{S}}|AK}{\delta} \right) \right)^2 \right). \tag{79}$$

### F.2.4 STEP-4: UPPER-BOUND THE FINAL REGRET

By connecting the sub-regret in the second learning layer to the choice in the first learning layer in step-1, connecting the sub-regret in the first two learning layers to the expected reward in the third learning layer in step-2, and connecting the expected reward in the third learning layer to the true optimal reward in step-3, we have finally upper-bounded the sub-regrets in all three layers due to the errors from the other layers. Hence, we have that, with probability $1 - \delta$, the regret of OP-MLL is upper-bounded as follows,

$Reg^{\text{OP-MLL}}(K)/H$

$$\le O \left( \frac{H^2 \eta_2 K}{\tilde{d}} + \frac{d \ln d}{\eta_1} + dH^{3/2}\sqrt{|\tilde{\mathbb{S}}|AK} \ln \left( \frac{H^2|\tilde{\mathbb{S}}|AK}{\delta} \right) + dH^2|\tilde{\mathbb{S}}|^2 A \left( \ln \left( \frac{H^2|\tilde{\mathbb{S}}|AK}{\delta} \right) \right)^2 \right)$$
$$= O \left( H\sqrt{\frac{dK \ln d}{\tilde{d}-1}} + H^2 \sqrt{\frac{d}{\tilde{d}-1}}|\tilde{\mathbb{S}}|^2 A \left( \ln \frac{H^2|\tilde{\mathbb{S}}|AK}{\delta} \right)^2 + H^{\frac{3}{2}}\sqrt{\frac{d|\tilde{\mathbb{S}}|AK}{\tilde{d}-1}} \ln \frac{H^2|\tilde{\mathbb{S}}|AK}{\delta} \right). \tag{80}$$

$\square$

### F.3 PROOF OF LEMMA 3

*Proof.* First, similar to the proof of Lemma 1 in Appendix E.3, we can show that

$$\frac{1}{\eta_2} \ln \left( \frac{\sum_{i \in \hat{i}^k} \tilde{w}^{k+1}(i)}{\sum_{i \in \hat{i}^k} \tilde{w}^k(i)} \right) = \frac{1}{\eta_2} \ln \left( \frac{\sum_{i \in \hat{i}^k} \tilde{w}^k(i) \cdot e^{\frac{\eta_2}{d}\sum_{h=1}^H r_{h,i}^k}}{\sum_{i \in \hat{i}^k} \tilde{w}^k(i)} \right)$$
$$\le \frac{1}{\eta_2} \ln \left( \frac{\sum_{i \in \hat{i}^k} \tilde{w}^k(i) \cdot e^{\frac{\eta_2}{d}\sum_{h=1}^H r_{h,i}^k}}{\sum_{i \in \hat{i}^k} \tilde{w}^k(i)} \right). \tag{81}$$

Note that **differently from** that in the proof of Lemma 1 in Appendix E.3, in Eq. (81) we focus on $i \in \hat{i}^k$. Next, since $e^x \leq 1 + x + (e-2)x^2$ for all $x \leq 1$, according to the update of the probability $p_i(k)$ in Eq. (2), we have

$$
\frac{1}{\eta_2} \ln \left( \frac{\sum\limits_{i \in \hat{i}^k} \tilde{w}^{k+1}(i)}{\sum\limits_{i \in \hat{i}^k} \tilde{w}^k(i)} \right)
$$

$$
\leq \frac{1}{\eta_2} \ln \left( 1 + \frac{\eta_2}{\tilde{d}(1 - \eta_2)} \sum_{i \in \hat{i}^k} p^k(i) \sum_{h=1}^{H} r_{h,i}^k + \frac{(e-2)\eta_2^2}{\tilde{d}^2(1-\eta_2)} \sum_{i \in \hat{i}^k} p^k(i) \left( \sum_{h=1}^{H} r_{h,i}^k \right)^2 \right)
$$

$$
\leq \frac{1}{\eta_2} \ln \left( 1 + \frac{\eta_2}{\tilde{d}(1 - \eta_2)} \sum_{i \in \hat{i}^k} \sum_{h=1}^{H} r_{h,i}^k + \frac{(e-2)\eta_2^2}{\tilde{d}^2(1-\eta_2)} \sum_{i \in \hat{i}^k} \left( \sum_{h=1}^{H} r_{h,i}^k \right)^2 \right)
$$

$$
\leq \frac{1}{\eta_2} \ln \left( 1 + \frac{\eta_2}{\tilde{d}(1 - \eta_2)} \sum_{i \in \hat{i}^k} \sum_{h=1}^{H} r_{h,i}^k + \frac{H(e-2)\eta_1^2}{\tilde{d}^2(1-\eta_2)} \sum_{i \in \hat{i}^k} \sum_{h=1}^{H} r_{h,i}^k \right). \tag{82}
$$

Then, since $\ln(1 + x) \leq x$ for all $x$, we have

$$
\frac{1}{\eta_2} \ln \left( \frac{\sum\limits_{i \in \hat{i}^k} \tilde{w}^{k+1}(i)}{\sum\limits_{i \in \hat{i}^k} \tilde{w}^k(i)} \right) \leq \frac{1}{\tilde{d}(1 - \eta_2)} \sum_{i \in \hat{i}^k} \sum_{h=1}^{H} r_{h,i}^k + \frac{H(e-2)\eta_2}{\tilde{d}^2(1-\eta_2)} \sum_{i \in \hat{i}^k} \sum_{h=1}^{H} r_{h,i}^k. \tag{83}
$$

Thus, we have

$$
\mathbb{E} \left[ \sum_{i \in \hat{i}^k} \sum_{h=1}^{H} r_h^k \right] \geq \frac{d(1 - \eta_2)}{\eta_2} \mathbb{E} \left[ \ln \left( \frac{\sum\limits_{i \in \hat{i}^k} \tilde{w}^{k+1}(i)}{\sum\limits_{i \in \hat{i}^k} \tilde{w}^k(i)} \right) \right] - \frac{H(e-2)\eta_2}{\tilde{d}} \mathbb{E} \left[ \sum_{i \in \hat{i}^k} \sum_{h=1}^{H} r_{h,i}^k \right]
$$

$$
\geq \frac{d(1 - \eta_2)}{\eta_2} \mathbb{E} \left[ \ln \left( \frac{\sum\limits_{i \in \hat{i}^k} \tilde{w}^{k+1}(i)}{\sum\limits_{i \in \hat{i}^k} \tilde{w}^k(i)} \right) \right] - \frac{H^2(e-2)\eta_2}{\tilde{d}}. \tag{84}
$$

$\square$

### F.4 Proof of Lemma 4

*Proof.* First, according to the update for the local weight in Eq. (58), we have

$$
\mathbb{E} \left[ \ln \left( \frac{\sum\limits_{i \in \hat{i}^k} \tilde{w}^{k+1}(i)}{\sum\limits_{i \in \hat{i}^k} \tilde{w}^k(i)} \right) \right]
$$

$$
= \mathbb{E} \left[ \ln \left( \sum_{i \in \hat{i}^k} \frac{\tilde{w}^k(i)}{\sum\limits_{i \in \hat{i}^k} \tilde{w}^k(i)} e^{\frac{\eta_2}{\tilde{d}} \sum\limits_{h=1}^{H} r_{h,i}^k} \right) \right]
$$

$$
= \mathbb{E} \left[ \ln \left( \sum_{i \in \hat{i}^k} \frac{\tilde{w}^k(i)}{\sum\limits_{i \in \hat{i}^k} \tilde{w}^k(i)} e^{\frac{\eta_2}{\tilde{d}} \sum\limits_{h=1}^{H} (r_{h,i}^k - \mathbb{E}[r_{h,i}^k])} \right) \right] + \frac{\eta_2}{\tilde{d}} \mathbb{E} \left[ \sum_{i \in \hat{i}^k} \sum_{h=1}^{H} r_{h,i}^k \right]. \tag{85}
$$

Since $e^x \geq 1 + x + \frac{1}{2}x^2$, the first term on the right-hand side of Eq. (85) can be lower-bounded as follows,

$$
\mathbb{E}\left[\ln\left(\sum_{i\in\hat{i}^k} \frac{\tilde{w}^k(i)}{\sum_{i\in\hat{i}^k}\tilde{w}^k(i)} e^{\frac{\eta_2}{\tilde{d}}\sum_{h=1}^{H}(r_{h,i}^k - \mathbb{E}[r_{h,i}^k])}\right)\right]
$$

$$
\geq \mathbb{E}\left[\ln\left(1 + \mathbb{E}\left[\sum_{i\in\hat{i}^k} \frac{\eta_2}{\tilde{d}}\left(\sum_{h=1}^{H} r_{h,i}^k - \mathbb{E}[\sum_{h=1}^{H} r_{h,i}^k]\right)\right]\right.\right.
$$

$$
\left.\left. + \mathbb{E}\left[\sum_{i=1}^{d} \frac{\eta_2^2}{2\tilde{d}^2}\left(\left(\sum_{h=1}^{H} r_{h,i}^k - \mathbb{E}[\sum_{h=1}^{H} r_{h,i}^k]\right)^2\right)\right]\right)\right]
$$

$$
\geq \mathbb{E}\left[\ln\left(1 + \mathbb{E}\left[\sum_{i\in\hat{i}^k} \frac{\eta_2^2}{2\tilde{d}^2}\left(\left(\sum_{h=1}^{H} r_{h,i}^k - \mathbb{E}[\sum_{h=1}^{H} r_{h,i}^k]\right)^2\right)\right]\right)\right]. \tag{86}
$$

Since $\ln(1 + x) \geq \frac{x}{2}$ for all $x \in [0, 1]$, we have

$$
\mathbb{E}\left[\ln\left(\sum_{i\in\hat{i}^k} \frac{\tilde{w}^{(u-1)\kappa+1}(i)}{\sum_{i\in\hat{i}^k}\tilde{w}^k(i)} e^{\frac{\eta_2}{\tilde{d}}\sum_{h=1}^{H}(r_{h,i}^k - \mathbb{E}[r_{h,i}^k])}\right)\right]
$$

$$
\geq \mathbb{E}\left[\sum_{i\in\hat{i}^k} \frac{\eta_2^2}{4\tilde{d}^2}\left(\left(\sum_{h=1}^{H} r_{h,i}^k - \mathbb{E}[\sum_{h=1}^{H} r_{h,i}^k]\right)^2\right)\right]. \tag{87}
$$

$\square$

## F.5 Proof of Lemma 5

*Proof.* We start from considering the right-hand side of Lemma 5. First, we have

$$
\mathbb{E}\left[\sum_{i=1}^{d}\left(\left(\sum_{h=1}^{H} r_{h,i}^k - \mathbb{E}[\sum_{h=1}^{H} r_{h,i}^k]\right)^2\right)\right]
$$

$$
= \mathbb{E}\left[\left(\sum_{h=1}^{H} r_{h,i}^k\right)^2\right] - \left(\mathbb{E}[\sum_{h=1}^{H} r_{h,i}^k]\right)^2
$$

$$
= \mathbb{E}_{i\in\{1,\dots,d\}}\left[\mathbb{E}_{\{i'\neq i\}}\left[\left(\sum_{h=1}^{H} r_{h,i'}^k\right)^2\right]\right] - \mathbb{E}_{i\in\{1,\dots,d\}}\left[\sum_{h=1}^{H} r_{h,i}^k \cdot \mathbb{E}_{\{i'\neq i\}}[\sum_{h=1}^{H} r_{h,i'}^k]\right]. \tag{88}
$$

By rearranging the terms in Eq. (88), we have

$$
\mathbb{E}\left[\sum_{i=1}^{d}\left(\left(\sum_{h=1}^{H} r_{h,i}^k - \mathbb{E}[\sum_{h=1}^{H} r_{h,i}^k]\right)^2\right)\right]
$$

$$
= \frac{1}{2}\mathbb{E}_{i\in\{1,\dots,d\}}\left[\mathbb{E}_{\{i'\neq i\}}\left[\left(\sum_{h=1}^{H} r_{h,i}^k\right)^2 + \left(\sum_{h=1}^{H} r_{h,i'}^k\right)^2\right]\right]
$$

$$
+ \mathbb{E}_{i\in\{1,\dots,d\}}\left[\mathbb{E}_{\{i'\neq i\}}\left[\left(\sum_{h=1}^{H} r_{h,i}^k\right)\left(\sum_{h=1}^{H} r_{h,i'}^k\right)\right]\right]
$$

$$
= \frac{1}{2}\mathbb{E}_{i\in\{1,\dots,d\}}\left[\mathbb{E}_{\{i'\neq i\}}\left[\left(\sum_{h=1}^{H} r_{h,i}^k - \sum_{h=1}^{H} r_{h,i'}^k\right)^2\right]\right]. \tag{89}
$$

Then, we consider all possible random choice of the sub-states sets $\hat{i}^k$, i.e., taking another expectation with respect to the randomness of $\hat{i}^k$. According to the law of total expectation, we have

$$
\mathbb{E}\left[\sum_{i=1}^{d}\left(\left(\sum_{h=1}^{H} r_{h,i}^{k}-\mathbb{E}[\sum_{h=1}^{H} r_{h,i}^{k}]\right)^{2}\right)\right]
$$
$$
=\frac{1}{2}\mathbb{E}_{\hat{i}^k}\left[\mathbb{E}_{i\in\{1,\ldots,d\}}\left[\mathbb{E}_{\{i'\neq i\}}\left[\left(\sum_{h=1}^{H} r_{h,i}^{k}-\sum_{h=1}^{H} r_{h,i'}^{k}\right)^{2}\right]\right]\Bigg|\hat{i}^k\right]. \tag{90}
$$

According to the updates of weights and probabilities in Eq. (58), we can see that the local probability $\tilde{p}_i(k)$ can be related to the global probabilities. That is, $\tilde{p}_i(k) = p_i(k) \cdot \frac{1}{\binom{d-1}{\tilde{d}-1}}$. Thus, from Eq. (90), we have

$$
\mathbb{E}\left[\sum_{i=1}^{d}\left(\left(\sum_{h=1}^{H} r_{h,i}^{k}-\mathbb{E}[\sum_{h=1}^{H} r_{h,i}^{k}]\right)^{2}\right)\right]
$$
$$
\leq\frac{\binom{d-1}{\tilde{d}-1}}{2\binom{d-2}{\tilde{d}-2}}\mathbb{E}_{\hat{i}^k}\left[\mathbb{E}_{i\in\{1,\ldots,d\}}\left[\mathbb{E}_{\{i'\neq i\}}\left[\left(\sum_{h=1}^{H} r_{h,i}^{k}-\sum_{h=1}^{H} r_{h,i'}^{k}\right)^{2}\right]\right]\Bigg|\hat{i}^k\right]
$$
$$
=\frac{d-1}{\tilde{d}-1}\mathbb{E}\left[\sum_{i\in\hat{i}^k}\left(\left(\sum_{h=1}^{H} r_{h,i}^{k}-\mathbb{E}[\sum_{h=1}^{H} r_{h,i}^{k}]\right)^{2}\right)\right]. \tag{91}
$$

$\square$

### F.6 PROOF OF LEMMA 7

*Proof.* The proof of Lemma 7 follows the same line of the proof for Lemma 2 and mainly utilizes a set of concentration inequalities Azar et al. (2017). Specifically, we prove Lemma 7 by mathematical induction.

Base case: when $h = H$, Eq. (76) trivially holds, since according to the definition of $V_H^k$, we have $\mathbb{E}\left[r_H^*\right] \leq V_H^k$. Note that the reward $r_{h'}^* = r_{h'}(\phi_{i_{h'}^*}(s_{h'}), a_{h'})$ is a function of the query $\hat{i}_{h'}^*$.

Induction: we hypothesize that $\mathbb{E}\left[\sum_{h'=h+1}^{H} r_{h'}^*\right] \leq V_{h+1}^k$. Then, we focus on proving that $\mathbb{E}\left[\sum_{h'=h}^{H} r_{h'}^*\right] \leq V_h^k$. Note that, we have

$$
V_h^k - \mathbb{E}\left[\sum_{h'=h}^{H} r_{h'}^*\right] \geq \mathbb{P}_h^{k,*} V_{h+1}^k(s) - \mathbb{P}_h^* V_{h+1}^k(s) + b_{h+1}^k(s,a), \tag{92}
$$

where $b_{h+1}^k(s,a)$ is the bonus term in Eq. (62). By adding and subtracting some middle terms, we have that

$$
V_h^k - \mathbb{E}\left[\sum_{h'=h}^{H} r_{h'}^*\right] \geq \mathbb{P}_h^{k,*}\left[V_{h+1}^k(s) - V_{h+1}^*(s)\right] + \left[\mathbb{P}_h^{k,*} - \mathbb{P}_h^*\right] V_{h+1}^*(s) + b_{h+1}^k(s,a). \tag{93}
$$

According to induction hypothesis that $\mathbb{E}\left[\sum_{h'=h+1}^{H} r_{h'}^*\right] \leq V_{h+1}^k$, we have

$$
V_h^k - \mathbb{E}\left[\sum_{h'=h}^{H} r_{h'}^*\right] \geq \left[\mathbb{P}_h^{k,*} - \mathbb{P}_h^*\right] V_{h+1}^*(s) + b_{h+1}^k(s,a). \tag{94}
$$

Finally, according to the empirical Bernstein's inequality, we have that with probability $1 - \delta$,

$$\left[ \mathbb{P}_h^* - \mathbb{P}_h^{k,*} \right] V_{h+1}^*(s) \leq b_{h+1}^k(s, a). \tag{95}$$

Combining Eq. (94) and Eq. (95), we have $\mathbb{E} \left[ \sum\limits_{h'=h}^{H} r_{h'}^* \right] \leq V_h^k.$ $\qquad\qquad\square$

## G  PROOF OF THEOREM 3

In this section, we prove the regret upper-bound in Theorem 3 of our OMLE-POSI algorithm, i.e., Algorithm 2. Recall that partial online state information (OSI) $\phi_{\hat{i}_{1:H}^k}$ affects both the maximum likelihood estimation (MLE) and joint query and action policy optimization. Thus, the main difficulty in the proof of Theorem 3 is how to capture such new effects. Indeed, directly applying existing observable operator method (OOM) Jaeger (2000); Liu et al. (2022) will result in a regret that does not decrease with $\tilde{d}$, which is a critical benefit that we should obtain from using partial OSI. To resolve this issue, we develop a new sub-matrix decomposition representation for the cumulative partial OSI and partial noisy observations, which non-trivially generalizes OOM to the case with both partial feedback and noisy observations.

In the following, we first provide a proof sketch in Appendix G.1, where we highlight the new difficulties and our new analytical ideas. Then, we provide a complete proof in Appendix G.4.

### G.1  SKETCH OF THE PROOF OF THEOREM 3

In this subsection, we first provide the proof sketch of Theorem 3, where we highlight the new difficulties and our new analytical ideas. Please see Appendix G.4 for the complete proof.

*Proof.* First, the regret can be represented as follows,

$$Reg^{\text{OMLE-POSI}}(K) = \sum_{k=1}^{K} \left[ V^* - V^{\pi^k} \right] = \sum_{k=1}^{K} \left[ V^* - V^k \right] + \sum_{k=1}^{K} \left[ V^k - V^{\pi^k} \right]. \tag{96}$$

According to Lemma 8 below, from Eq. (96), we have that with probability $1 - \delta$,

$$Reg^{\text{OMLE-POSI}}(K) = \sum_{k=1}^{K} \left[ V^* - V^{\pi^k} \right] \leq \sum_{k=1}^{K} \left[ V^k - V^{\pi^k} \right]. \tag{97}$$

**Lemma 8.** *By choosing* $\beta = O\left( \left( |\tilde{\mathbb{S}}|^{2d} A + |\tilde{\mathbb{S}}|^{d-\tilde{d}} O \right) \ln(|\tilde{\mathbb{S}}|^d AOHK) + \ln \frac{K}{\delta} \right)$*, with probability at least* $1 - \delta$*, we have*

$$\left( \mathbb{P}, \tilde{\mathbb{O}}, \Delta_1 \right) \in \left\{ \left( \mathbb{P}^k, \tilde{\mathbb{O}}^k, \Delta_1^k \right) \right\}, \tag{98}$$

*where the big-O notation hides the constant and logarithmic terms including* $\binom{d}{\tilde{d}}$*.*

Lemma 8 shows an important property of the maximum likelihood estimation used in our OMLE-POSI algorithm. That is, by utilizing partial OSI, with high probability, the true transition kernel and partial emission model must be characterized by the set of the estimated transition kernels and the partial emission models. Please see Appendix G.2 for the proof of Lemma 8.

Since the expected total reward in each episode is at most $H$, from Eq. (97), we have

$$Reg^{\text{OMLE-POSI}}(K) \leq H \sum_{k=1}^{K} \sum_{\Gamma} \left| P_{\mathbb{P}^k, \overline{\mathbb{O}}^{\hat{i},k}, \Delta_1^k}^{\pi^k}(\Gamma) - P_{\mathbb{P}, \overline{\mathbb{O}}^{\hat{i}}, \Delta_1}^{\pi^k}(\Gamma) \right|. \tag{99}$$

Next, based on the observable operator method (OOM) in Jaeger (2000); Liu et al. (2022); Chen et al. (2022a); Foster et al. (2021), we develop a new trajectory representation using sub-matrix decomposition. This new development is for addressing the aforementioned problem caused by

using partial OSI in our case. Specifically, we use $\bar{\mathbb{O}}^{\hat{i}} \in \mathbb{R}^{O \times |\tilde{\mathbb{S}}|^d}$ to denote the *augmented partial emission model* and use $\Delta_1 \in \mathbb{R}^S$ to denote the initialization model. We note that the augmented partial emission matrix $\bar{\mathbb{O}}^{\hat{i}}$ is generated based on the partial emission matrix $\tilde{o}^{\hat{i}}$ and depends on the query $\hat{i}$. That is, $|\tilde{\mathbb{S}}|^{d-\tilde{d}}$ columns of $\bar{\mathbb{O}}^{\hat{i}}$ correspond to the partial noisy observations $\tilde{o}$ for the non-queried sub-states $\{\phi_i(s)\}_{\{i \notin \hat{i}\}}$ and $|\tilde{\mathbb{S}}|^{\tilde{d}}$ columns are set to $0$ because the corresponding sub-states do not generate partial noisy observation. Then, the observable operator corresponding to a specific $\theta$ in our setting can be represented as follows,

$$
\boldsymbol{b}_0^\theta(\hat{i}_1) = \bar{\mathbb{O}}_1^{\hat{i}_1,\theta} \Delta_1^\theta \in \mathbb{R}^O,
$$
$$
\boldsymbol{B}_h^\theta(\tilde{o}, a, \hat{i}_h, \hat{i}_{h+1}) = \bar{\mathbb{O}}_{h+1}^{\hat{i}_{h+1},\theta} \mathbb{P}_{h,a}^\theta \mathrm{diag}(\bar{\mathbb{O}}_h^{\hat{i}_h,\theta}(\tilde{o}|\cdot))(\bar{\mathbb{O}}_h^{\hat{i}_h,\theta})^\dagger \in \mathbb{R}^{O \times O}, \tag{100}
$$

where $\mathbb{P}_{h,a}^\theta \in \mathbb{R}^{S \times S}$ is the transition matrix for action $a$, $\bar{\mathbb{O}}_h^{\hat{i}_h,\theta}(\tilde{o}|\cdot)$ is the $\tilde{o}$-th row of the augmented partial emission matrix $\bar{\mathbb{O}}_h^{\hat{i},\theta}$ and $\mathrm{diag}(\bar{\mathbb{O}}_h^{\hat{i}_h,\theta}(\tilde{o}|\cdot))$ is the diagonal matrix with diagonal entries equal to $\bar{\mathbb{O}}_h^{\hat{i}_h,\theta}(\tilde{o}|\cdot)$. Please note that different from the standard observable operator, due to the partial OSI in our problem, $\boldsymbol{B}_h^\theta(\tilde{o}, a, \hat{i}_h, \hat{i}_{h+1})$ is now a function of two consecutive queries $(\hat{i}_h, \hat{i}_{h+1})$ of the agent. Let us focus on the probability difference on the right-hand side of Eq. (99). Specifically, the probability difference can be represented by the difference between the product of conditional (conditioned on the past feedback) probability of feedback at each step under the observed trajectory and the true case, i.e.,

$$
\sum_{k=1}^K \sum_\Gamma \left| P_{\mathbb{P}^k, \bar{\mathbb{O}}^{\hat{i},k}, \Delta_1^k}^{\pi^k}(\Gamma) - P_{\mathbb{P}, \bar{\mathbb{O}}^{\hat{i}}, \Delta_1}^{\pi^k}(\Gamma) \right|
$$
$$
= \sum_{k=1}^K \sum_\Gamma \left| \left( e_{\tilde{o}_H}^\mathrm{T} \boldsymbol{B}_H^{\theta^k}(\tilde{o}_H, a_H, \hat{i}_H, \hat{i}_{H+1}) \cdots \boldsymbol{B}_1^{\theta^k}(\tilde{o}_1, a_1, \hat{i}_1, \hat{i}_2) \boldsymbol{b}_0^{\theta^k}(\hat{i}_1) \right) \right.
$$
$$
\left. - \left( e_{\tilde{o}_h}^\mathrm{T} \boldsymbol{B}_H^\theta(\tilde{o}_H, a_H, \hat{i}_H, \hat{i}_{H+1}) \cdots \boldsymbol{B}_1^\theta(\tilde{o}_1, a_1, \hat{i}_1, \hat{i}_2) \boldsymbol{b}_0^\theta(\hat{i}_1) \right) \right| \cdot \pi^k(\Gamma)
$$
$$
= \sum_{k=1}^K \sum_\Gamma \left\| \left( \boldsymbol{B}_H^{\theta^k}(\tilde{o}_H, a_H, \hat{i}_H, \hat{i}_{H+1}) \cdots \boldsymbol{B}_1^{\theta^k}(\tilde{o}_1, a_1, \hat{i}_1, \hat{i}_2) \boldsymbol{b}_0^{\theta^k}(\hat{i}_1) \right) \right.
$$
$$
\left. - \left( \boldsymbol{B}_H^\theta(\tilde{o}_H, a_H, \hat{i}_H, \hat{i}_{H+1}) \cdots \boldsymbol{B}_1^\theta(\tilde{o}_1, a_1, \hat{i}_1, \hat{i}_2) \boldsymbol{b}_0^\theta(\hat{i}_1) \right) \right\|_1 \cdot \pi^k(\Gamma). \tag{101}
$$

After decomposing the probability representations as in Eq. (101), according to the triangle inequality and the partial-revealing condition, we have

$$
\sum_{k=1}^K \sum_\Gamma \left| P_{\mathbb{P}^k, \bar{\mathbb{O}}^{\hat{i},k}, \Delta_1^k}^{\pi^k}(\Gamma) - P_{\mathbb{P}, \bar{\mathbb{O}}^{\hat{i}}, \Delta_1}^{\pi^k}(\Gamma) \right|
$$
$$
\leq \frac{|\tilde{\mathbb{S}}|^{(d-\tilde{d})/2}}{\alpha} \left[ \sum_{k=1}^K \sum_{j=1}^H \sum_{\Gamma_j} \left\| \left( \boldsymbol{B}_j^{\theta^k}(\tilde{o}_j, a_j, \hat{i}_j, \hat{i}_{j+1}) - \boldsymbol{B}_j^\theta(\tilde{o}_j, a_j, \hat{i}_j, \hat{i}_{j+1}) \right) \cdot \boldsymbol{b}_{j-1}^\theta(\Gamma_{j-1}) \right\|_1 \cdot \pi^k(\Gamma_j) \right.
$$
$$
\left. + \left\| \boldsymbol{b}_0^{\theta^k}(\hat{i}_1) - \boldsymbol{b}_0^\theta(\hat{i}_1) \right\|_1 \right]. \tag{102}
$$

For the first term on the right-hand side of Eq. (102), we have

$$
\sum_{k=1}^{K-1} \sum_{\Gamma_j} \left\| \left( \boldsymbol{B}_j^{\theta^K}(\tilde{o}_j, a_j, \hat{i}_j, \hat{i}_{j+1}) - \boldsymbol{B}_j^{\theta}(\tilde{o}_j, a_j, \hat{i}_j, \hat{i}_{j+1}) \right) \cdot \boldsymbol{b}_{j-1}^{\theta}(\Gamma_{j-1}) \right\|_1 \cdot \pi^k(\Gamma_j)
$$

$$
\leq \sum_{k=1}^{K-1} \sum_{\Gamma_j} \left\| \boldsymbol{B}_j^{\theta^K}(\tilde{o}_j, a_j, \hat{i}_j, \hat{i}_{j+1}) \cdot \left( \boldsymbol{b}_{j-1}^{\theta^K}(\Gamma_{j-1}) - \boldsymbol{b}_{j-1}^{\theta}(\Gamma_{j-1}) \right) \right\|_1 \cdot \pi^k(\Gamma_j)
$$

$$
+ \sum_{k=1}^{K-1} \sum_{\Gamma_j} \left\| \boldsymbol{B}_j^{\theta^K}(\tilde{o}_j, a_j, \hat{i}_j, \hat{i}_{j+1}) \boldsymbol{b}_{j-1}^{\theta^K}(\Gamma_{j-1}) - \boldsymbol{B}_j^{\theta}(\tilde{o}_j, a_j, \hat{i}_j, \hat{i}_{j+1}) \boldsymbol{b}_{j-1}^{\theta}(\Gamma_{j-1}) \right\|_1 \cdot \pi^k(\Gamma_j).
$$

$$(103)$$

Next, we can upper-bound the partial observation trajectory difference, i.e., the two terms on the right-hand side of Eq. (103), using the conclusion in Lemma 9 below.

**Lemma 9.** *For each episode $k$, we have*

$$
\sum_{\Gamma_{h_1:h_2}^k} \pi^k(\Gamma_{h_1:h_2}^k) \cdot \left\| \left( \prod_{j=h_1}^{h_2} \bar{\mathbb{O}}_{j+1}^{\hat{i}_{h+1}^k} \mathbb{P}_{j,a}^{\theta^k} diag(\bar{\mathbb{O}}_j^{\hat{i}_j^k}(\tilde{o}_j|\cdot)(\bar{\mathbb{O}}_j^{\hat{i}_j^k})^{\dagger}) \right) \bar{\mathbb{O}}_{h_1}^{\hat{i}_{h_1}^k} \Delta_{h_1}^{\theta^k} \right\|_1 \leq \frac{|\tilde{\mathbb{S}}|^{\frac{d-\tilde{d}}{2}}}{\alpha} \left\| \bar{\mathbb{O}}_{h_1}^{\hat{i}_{h_1}^k} \Delta_{h_1}^{\theta^k} \right\|_1.
$$

$$(104)$$

Lemma 9 shows that the conditional probability of receiving a certain trajectory about the partial OSI and noisy observations, i.e., the left-hand side of Lemma 9, depends on the size of the state space $|\tilde{\mathbb{S}}|^{\frac{d-\tilde{d}}{2}}$, partial-revealing parameter $\alpha$ and the initial probability measure $\left\| \bar{\mathbb{O}}_{h_1}^{\hat{i}_{h_1}^k} \Delta_{h_1}^{\theta^k} \right\|_1$, i.e., the terms on the right-hand side of Lemma 9. We note that in Eq. (102) and Eq. (104), the multiplicative factor depends on $|\tilde{\mathbb{S}}|^{d-\tilde{d}}$, which is size of the space of the sub-states that cannot be queried by the agent at each step. Please see Appendix G.3 for the proof of Lemma 9. Applying Lemma 9 to Eq. (103), we have

$$
\sum_{k=1}^{K-1} \sum_{\Gamma_j} \left\| \left( \boldsymbol{B}_j^{\theta^K}(\tilde{o}_j, a_j, \hat{i}_j, \hat{i}_{j+1}) - \boldsymbol{B}_j^{\theta}(\tilde{o}_j, a_j, \hat{i}_j, \hat{i}_{j+1}) \right) \cdot \boldsymbol{b}_{j-1}^{\theta}(\Gamma_{j-1}) \right\|_1 \cdot \pi^k(\Gamma_j)
$$

$$
= O(\frac{|\tilde{\mathbb{S}}|^{(d-\tilde{d})/2}}{\alpha} \sqrt{K\beta}).
$$

$$(105)$$

Finally, by combining Eq. (96)-Eq. (105) and the value of $\beta$, with probability $1 - \delta$, we can upper-bound the regret of OMLE-POSI as follows,

$$
Reg^{OMLE-POSI}(K) \leq O\left( \sqrt{(|\tilde{\mathbb{S}}|^{2d}A + |\tilde{\mathbb{S}}|^{d-\tilde{d}}O)K} \cdot |\tilde{\mathbb{S}}|^{2(d-\tilde{d})} H^4 AO/\alpha^2 \right).
$$

$$(106)$$

$$\square$$

## G.2 PROOF OF LEMMA 8

*Proof.* This lemma mainly follows the standard properties of MLE and the proof of Proposition 13 in Liu et al. (2022). The main difference here is due to the queried partial state information and partial emission model considered in our setting. Specifically, first, according to Eq. (14) there (i.e., $\|\mathbb{P}_\theta^\pi - \mathbb{P}_{\bar{\theta}}^\pi\|_1 \leq \frac{1}{T}$), we have

$$
\mathbb{E}[e^{\sum_{\tau=1}^{K} \ln \frac{P_{\mathbb{P}^\tau, \tilde{\mathbb{O}}^\tau}^{\pi^\tau}(\Gamma^\tau)}{P_{\mathbb{P}^*, \tilde{\mathbb{O}}^*}^{\pi^\tau}(\Gamma^\tau)}}] = \mathbb{E}[e^{\sum_{\tau=1}^{K-1} \ln \frac{P_{\mathbb{P}^\tau, \tilde{\mathbb{O}}^\tau}^{\pi^\tau}(\Gamma^\tau)}{P_{\mathbb{P}^*, \tilde{\mathbb{O}}^*}^{\pi^\tau}(\Gamma^\tau)}} \cdot \|P_{\mathbb{P}^\tau, \tilde{\mathbb{O}}^\tau}^{\pi^\tau}(\Gamma^K)\|_1]
$$

$$
\leq \mathbb{E}[e^{\sum_{\tau=1}^{K-1} \ln \frac{P_{\mathbb{P}^\tau, \tilde{\mathbb{O}}^\tau}^{\pi^\tau}(\Gamma^\tau)}{P_{\mathbb{P}^*, \tilde{\mathbb{O}}^*}^{\pi^\tau}(\Gamma^\tau)}} \cdot \left( 1 + \frac{1}{K} \right)].
$$

$$(107)$$

By applying mathematical induction, we have that the right-hand-side of Eq. (107) is upper-bounded by $(1 + 1/K)^K$, which is less than or equal to $e$. Thus, Eq. (107) further yields

$$\mathbb{E}[e^{\sum_{\tau=1}^{K} \ln \frac{P^{\pi^\tau}_{\mathbb{P}^\tau, \tilde{\mathbb{O}}^\tau}(\Gamma^\tau)}{P^{\pi^\tau}_{\mathbb{P}^*, \tilde{\mathbb{O}}^*}(\Gamma^\tau)}}] \leq e. \tag{108}$$

Then, according to Markov's inequality, we have that

$$Pr\left\{ \sum_{\tau=1}^{K} \ln \frac{P^{\pi^\tau}_{\mathbb{P}^\tau, \tilde{\mathbb{O}}^\tau}(\Gamma^\tau)}{P^{\pi^\tau}_{\mathbb{P}^*, \tilde{\mathbb{O}}^*}(\Gamma^\tau)} > \log(\frac{1}{\delta}) \right\} \leq e\delta. \tag{109}$$

Then, by taking the union bound over the state-transition space $\mathbb{P}$, partial emission space $\tilde{\mathbb{O}}$ and episodes $\tau$, we have that for any $\delta \in (0,1)$, there exists $c > 0$, s.t.,

$$Pr\left\{ \max_{\mathbb{P}, \tilde{\mathbb{O}}, k} \sum_{\tau=1}^{K} \ln \frac{P^{\pi^\tau}_{\mathbb{P}^k, \tilde{\mathbb{O}}^k}(\Gamma^\tau)}{P^{\pi^\tau}_{\mathbb{P}, \tilde{\mathbb{O}}}(\Gamma^\tau)} \geq c(|\tilde{\mathbb{S}}|^{2d}A + |\tilde{\mathbb{S}}|^{d-\tilde{d}}O\binom{d}{\tilde{d}}) \ln(|\tilde{\mathbb{S}}|^d AOHK) + \ln \frac{K}{\delta} \right\} \leq \delta, \tag{110}$$

where the quantity $c(|\tilde{\mathbb{S}}|^{2d}A + |\tilde{\mathbb{S}}|^{d-\tilde{d}}O\binom{d}{\tilde{d}}) \ln(|\tilde{\mathbb{S}}|^d AOHK) + \ln \frac{K}{\delta}$ follows because the dimension of the state-transition space is $|\tilde{\mathbb{S}}|^{2d}A$, the dimension of the partial emission model space (i.e., $\tilde{\mathbb{O}} : \tilde{S} \times \tilde{\mathcal{O}} \to [0,1]$) is $|\tilde{\mathbb{S}}|^{d-\tilde{d}}O$, the structure of the partial emission model space is independent of the queried sub-states, and the total number of episodes is $K$. Finally, Lemma 8 follows because $\beta$ is an upper-bound of this quantity.

$\square$

## G.3 PROOF OF LEMMA 9

Note that this lemma is one of the **key** parts where we can leverage partial OSI to improve the regret to be decreasing exponentially as $\tilde{d}$ increases.

*Proof.* First, we have

$$\sum_{\Gamma^k_{h_1:h_2}} \pi^k(\Gamma^k_{h_1:h_2}) \cdot \left\| \left( \prod_{j=h_1}^{h_2} \bar{\mathbb{O}}^{\hat{i}^k_{h+1}}_{j+1} \mathbb{P}^{\theta^k}_{j,a} \text{diag}(\bar{\mathbb{O}}^{\hat{i}^k_j}_j(\tilde{o}_j|\cdot)(\bar{\mathbb{O}}^{\hat{i}^k_j}_j)^\dagger) \right) \bar{\mathbb{O}}^{\hat{i}^k_{h_1}}_{h_1} \Delta^{\theta^k}_{h_1} \right\|_1$$

$$= \sum_{\Gamma^k_{h_1:h_2}} \pi^k(\Gamma^k_{h_1:h_2}) \cdot \left\| \left( \prod_{j=h_1}^{h_2} \bar{\mathbb{O}}^{\hat{i}^k_{h+1}}_{j+1} \mathbb{P}^{\theta^k}_{j,a} \text{diag}(\bar{\mathbb{O}}^{\hat{i}^k_j}_j(\tilde{o}_j|\cdot)(\bar{\mathbb{O}}^{\hat{i}^k_j}_j)^\dagger) \right) \prod_{j=h_1}^{h_2} \bar{\mathbb{O}}^{\hat{i}^k_j}_j \left( \prod_{j=h_1}^{h_2} \bar{\mathbb{O}}^{\hat{i}^k_j}_j \right)^{-1} \bar{\mathbb{O}}^{\hat{i}^k_{h_1}}_{h_1} \Delta^{\theta^k}_{h_1} \right\|_1. \tag{111}$$

Due to the partial online state information for the chosen $\tilde{d}$ sub-states, the row space of partial-OSI representation product in Eq. (111) belongs to the column space of the partial-revealing matrix. Thus, by canceling out some middle terms on the right-hand side of Eq. (111), we have

$$\sum_{\Gamma^k_{h_1:h_2}} \pi^k(\Gamma^k_{h_1:h_2}) \cdot \left\| \left( \prod_{j=h_1}^{h_2} \bar{\mathbb{O}}^{\hat{i}^k_{h+1}}_{j+1} \mathbb{P}^{\theta^k}_{j,a} \text{diag}(\bar{\mathbb{O}}^{\hat{i}^k_j}_j(\tilde{o}_j|\cdot)(\bar{\mathbb{O}}^{\hat{i}^k_j}_j)^\dagger) \right) \bar{\mathbb{O}}^{\hat{i}^k_{h_1}}_{h_1} \Delta^{\theta^k}_{h_1} \right\|_1$$

$$\leq \sum_{\Gamma^k_{h_1:h_2}} \pi^k(\Gamma^k_{h_1:h_2}) \cdot \left\| \left( \prod_{j=h_1}^{h_2} \bar{\mathbb{O}}^{\hat{i}^k_j}_j \right)^{-1} \bar{\mathbb{O}}^{\hat{i}^k_{h_1}}_{h_1} \Delta^{\theta^k}_{h_1} \right\|_1$$

$$\leq \frac{|\tilde{\mathbb{S}}|^{\frac{d-\tilde{d}}{2}}}{\alpha} \left\| \bar{\mathbb{O}}^{\hat{i}^k_{h_1}}_{h_1} \Delta^{\theta^k}_{h_1} \right\|_1. \tag{112}$$

$\square$

### G.4 THE COMPLETE PROOF OF THEOREM 3

In previous subsections, we have introduced the proof sketch for Theorem 3, where we highlighted the new difficulties and our new analytical ideas in the proof. In this subsection, we provide the complete proof of Theorem 3, where we repeat some details in Liu et al. (2022) for completeness.

To begin with, we state Theorem 3 more formally with more details.

**Theorem 7.** *(Regret) For POMDPs with the partial online state information and partially revealing condition, with probability $1 - \delta$ for any $\delta \in (0, 1]$, the regret of OMLE-POSI can be upper-bounded as follows,*

$$Reg^{\text{OMLE-POSI}}(K) \leq \tilde{O}\left(|\tilde{\mathbb{S}}|^{2(d-\tilde{d})}OAH^4\sqrt{K(|\tilde{\mathbb{S}}|^{2d}A + |\tilde{\mathbb{S}}|^{d-\tilde{d}}O)}/\alpha^2\right) \tag{113}$$

*Proof.* The proof of Theorem 3 can be divided into four important steps as follows.

**Step 1 (Observable operator decomposition):**

Recall that in Class 2, we use $\Gamma_h^k \triangleq \{\hat{i}_1^k, \phi_{\hat{i}_1^k}(s_1^k), \tilde{o}_1^k, a_1^k, ..., \hat{i}_{h-1}^k, \phi_{\hat{i}_{h-1}^k}(s_{h-1}^k), \tilde{o}_{h-1}^k, a_{h-1}^k\}$ to denote the feedback (including both the partial OSI $\Phi_h^k$ and partial noisy observations $\tilde{o}_{1:h-1}^k$). For simplicity, similar to that in Algorithm 2, we use $\theta \triangleq (\mathbb{P}, \bar{\mathbb{O}}^{\hat{i}}, \Delta_1)$ to denote the joint problem model, where we use $\mathbb{P} \in \mathbb{R}^{S \times S \times A}$ to denote the POMDP transition matrix, use $\bar{\mathbb{O}}^{\hat{i}} \in \mathbb{R}^{O \times |\tilde{\mathbb{S}}|^d}$ to denote *augmented partial emission matrix*, and use $\Delta_1 \in \mathbb{R}^S$ to denote the initialization matrix. We note that the augmented partial emission matrix $\bar{\mathbb{O}}^{\hat{i}}$ is generated based on the partial emission matrix $\tilde{o}^{\hat{i}}$ and depends on the query $\hat{i}$. That is, $|\tilde{\mathbb{S}}|^{d-\tilde{d}}$ columns of $\bar{\mathbb{O}}^{\hat{i}}$ correspond to the partial noisy observations $\tilde{o}$ for the non-queried sub-states $\{\phi_i(s)\}_{\{i \notin \hat{i}\}}$ and $|\tilde{\mathbb{S}}|^{\tilde{d}}$ columns are set to 0 because the corresponding sub-states do not generate partial noisy observation. Then, the observable operator corresponding to a specific $\theta$ in our setting can be represented as follows,

$$\boldsymbol{b}_0^\theta(\hat{i}_1) = \bar{\mathbb{O}}_1^{\hat{i}_1, \theta}\Delta_1^\theta \in \mathbb{R}^O,$$

$$\boldsymbol{B}_h^\theta(\tilde{o}, a, \hat{i}_h, \hat{i}_{h+1}) = \bar{\mathbb{O}}_{h+1}^{\hat{i}_{h+1}, \theta}\mathbb{P}_{h,a}^\theta \text{diag}(\bar{\mathbb{O}}_h^{\hat{i}_h, \theta}(\tilde{o}|\cdot))(\bar{\mathbb{O}}_h^{\hat{i}_h, \theta})^\dagger \in \mathbb{R}^{O \times O}, \tag{114}$$

where $\mathbb{P}_{h,a}^\theta \in \mathbb{R}^{S \times S}$ is the transition matrix for action $a$, $\bar{\mathbb{O}}_h^{\hat{i}_h, \theta}(\tilde{o}|\cdot)$ is the $\tilde{o}$-th row of the augmented partial emission matrix $\bar{\mathbb{O}}_h^{\hat{i}, \theta}$, $\text{diag}(\bar{\mathbb{O}}_h^{\hat{i}_h, \theta}(\tilde{o}|\cdot))$ is the diagonal matrix with diagonal entries equal to $\bar{\mathbb{O}}_h^{\hat{i}_h, \theta}(\tilde{o}|\cdot)$, and $(\cdot)^\dagger$ represents the Moore-Penrose inverse of a matrix. Please note that different from the standard observable operator, due to the partial OSI in our problem, $\boldsymbol{B}_h^\theta(\tilde{o}, a, \hat{i}_h, \hat{i}_{h+1})$ is now a function of two consecutive queries $(\hat{i}_h, \hat{i}_{h+1})$ of the agent. Moreover, we use $\theta^k \triangleq (\mathbb{P}^k, \bar{\mathbb{O}}^{\hat{i}, k}, \Delta_1^k)$ to denote the joint POMDP transition model, augmented partial emission model and initialization model of Algorithm 2 in the $k$-th episode. Given the POMDP parameters, we can represent the probability of observing a specific trajectory (of the queries, partial OSI and partial noisy observation) based on the observable operator $\boldsymbol{B}_h^\theta(\tilde{o}, a, \hat{i}_h, \hat{i}_{h+1})$. Specifically, we have that

$$P_\theta^\pi(\Gamma_h) = \pi(\Gamma_h) \times \left(e_{\tilde{o}_h}^{\text{T}}\boldsymbol{B}_{h-1}^\theta(\tilde{o}, a, \hat{i}_{h-1}, \hat{i}_h)\cdots\boldsymbol{B}_1^\theta(\tilde{o}, a, \hat{i}_1, \hat{i}_2)\boldsymbol{b}_0^\theta(\hat{i}_1)\right) \in \mathbb{R}, \tag{115}$$

where $\pi(\Gamma_h) \triangleq \prod_{h'=1}^{h-1} \pi_{q,h'}(\hat{i}_{h'}|\Gamma_{h'-1})\pi_{a,h'}(a_{h'}|\Gamma_{h'-1}, \phi_{\hat{i}_{h'}}, \tilde{o}_{h'})$, $\pi_q$ and $\pi_a$ denotes the query policy and action policy, respectively, and $e_{\tilde{o}_h}$ is an identity vector.

**Step 2 (Bound the total variation distance by the operator estimation error):**

The regret of OMLE-POSI can be represented as follows,

$$Reg^{\text{OMLE-POSI}}(K) = \sum_{k=1}^K \left[V^* - V^{\pi^k}\right] = \sum_{k=1}^K \left[V^* - V^k\right] + \sum_{k=1}^K \left[V^k - V^{\pi^k}\right]. \tag{116}$$

According to Lemma 8, from Eq. (116), we have that with probability $1 - \delta$,

$$Reg^{\text{OMLE-POSI}}(K) = \sum_{k=1}^K \left[V^* - V^{\pi^k}\right] \leq \sum_{k=1}^K \left[V^k - V^{\pi^k}\right]. \tag{117}$$

Since the expected total reward in each episode is at most $H$, from Eq. (117), we have

$$Reg^{\text{OMLE-POSI}}(K) \leq H \sum_{k=1}^{K} \sum_{\Gamma} \left| P_{\mathbb{P}^k, \bar{\mathbb{O}}^{\hat{i},k}, \Delta_1^k}^{\pi^k}(\Gamma) - P_{\mathbb{P}, \bar{\mathbb{O}}^{\hat{i}}, \Delta_1}^{\pi^k}(\Gamma) \right|, \tag{118}$$

where $\Gamma = \{\hat{i}_1, \phi_{\hat{i}_1}(s_1), \tilde{o}_1, a_1, ..., \hat{i}_H, \phi_{\hat{i}_H}(s_H), \tilde{o}_H, a_H\}$ represents any possible feedback trajectory in a whole episode, $\mathbb{P}^k, \bar{\mathbb{O}}^{\hat{i},k}, \Delta_1^k$ is the POMDP parameters estimated at episode $k$, and $\mathbb{P}, \bar{\mathbb{O}}^{\hat{i}}, \Delta_1$ are the true POMDP parameters of the environment. Hence, to bound the regret of OMLE-POSI, we need to bound the total variation (TV) distance on the right-hand-side of Eq. (118).

Specifically, for a matrix $\mathbb{A}$ and $p \geq 1$, we denote the matrix norm $\|\mathbb{A}\|_p = \sup_{\vec{x}:\|\vec{x}\|_p \leq 1} \|\mathbb{A}\vec{x}\|_p$. Then, the TV difference can be represented by the difference between the product of conditional (conditioned on the past feedback) probability of feedback at each step under the observed feedback trajectory and the true model, i.e.,

$$\sum_{k=1}^{K} \sum_{\Gamma} \left| P_{\mathbb{P}^k, \bar{\mathbb{O}}^{\hat{i},k}, \Delta_1^k}^{\pi^k}(\Gamma) - P_{\mathbb{P}, \bar{\mathbb{O}}^{\hat{i}}, \Delta_1}^{\pi^k}(\Gamma) \right|$$

$$= \sum_{k=1}^{K} \sum_{\Gamma} \left| \left( e_{\tilde{o}_H}^{\mathrm{T}} \boldsymbol{B}_H^{\theta^k}(\tilde{o}_H, a_H, \hat{i}_H, \hat{i}_{H+1}) \cdots \boldsymbol{B}_1^{\theta^k}(\tilde{o}_1, a_1, \hat{i}_1, \hat{i}_2) \boldsymbol{b}_0^{\theta^k}(\hat{i}_1) \right) \right.$$

$$\left. - \left( e_{\tilde{o}_h}^{\mathrm{T}} \boldsymbol{B}_H^{\theta}(\tilde{o}_H, a_H, \hat{i}_H, \hat{i}_{H+1}) \cdots \boldsymbol{B}_1^{\theta}(\tilde{o}_1, a_1, \hat{i}_1, \hat{i}_2) \boldsymbol{b}_0^{\theta}(\hat{i}_1) \right) \right| \cdot \pi^k(\Gamma)$$

$$= \sum_{k=1}^{K} \sum_{\Gamma} \left\| \left( \boldsymbol{B}_H^{\theta^k}(\tilde{o}_H, a_H, \hat{i}_H, \hat{i}_{H+1}) \cdots \boldsymbol{B}_1^{\theta^k}(\tilde{o}_1, a_1, \hat{i}_1, \hat{i}_2) \boldsymbol{b}_0^{\theta^k}(\hat{i}_1) \right) \right.$$

$$\left. - \left( \boldsymbol{B}_H^{\theta}(\tilde{o}_H, a_H, \hat{i}_H, \hat{i}_{H+1}) \cdots \boldsymbol{B}_1^{\theta}(\tilde{o}_1, a_1, \hat{i}_1, \hat{i}_2) \boldsymbol{b}_0^{\theta}(\hat{i}_1) \right) \right\|_1 \cdot \pi^k(\Gamma), \tag{119}$$

where we keep the additional fictitious query $\hat{i}_{H+1}$ for clarity.

Thus, to bound the TV distance, we need to bound the difference between the above two products of operators in $\|\cdot\|_1$. Next, we bound such a difference by the difference between observable operator pairs at each step one-by-one. Specifically, after decomposing the probability representations as in Eq. (119), according to the triangle inequality and the partially revealing condition in Class 2, we have that

$$\sum_{\Gamma} \left\| \left( \boldsymbol{B}_H^{\theta^k}(\tilde{o}_H, a_H, \hat{i}_H, \hat{i}_{H+1}) \cdots \boldsymbol{B}_1^{\theta^k}(\tilde{o}_1, a_1, \hat{i}_1, \hat{i}_2) \boldsymbol{b}_0^{\theta^k}(\hat{i}_1) \right) \right.$$

$$\left. - \left( \boldsymbol{B}_H^{\theta}(\tilde{o}_H, a_H, \hat{i}_H, \hat{i}_{H+1}) \cdots \boldsymbol{B}_1^{\theta}(\tilde{o}_1, a_1, \hat{i}_1, \hat{i}_2) \boldsymbol{b}_0^{\theta}(\hat{i}_1) \right) \right\|_1 \cdot \pi^k(\Gamma)$$

$$\leq \sum_{j=1}^{H} \sum_{\Gamma} \left\| \left( \prod_{j'=j+1}^{H} \boldsymbol{B}_{j'}^{\theta^k}(\tilde{o}_{j'}, a_{j'}, \hat{i}_{j'}, \hat{i}_{j'+1}) \left( \boldsymbol{B}_j^{\theta^k}(\tilde{o}_j, a_j, \hat{i}_j, \hat{i}_{j+1}) - \boldsymbol{B}_j^{\theta}(\tilde{o}_j, a_j, \hat{i}_j, \hat{i}_{j+1}) \right) \right. \right.$$

$$\left. \left. \cdot \boldsymbol{b}_{j-1}^{\theta}(\Gamma_{j-1}) \right) \right\|_1 \cdot \pi^k(\Gamma) + \sum_{\Gamma} \left\| \left( \prod_{j'=1}^{H} \boldsymbol{B}_{j'}^{\theta^k}(\tilde{o}_{j'}, a_{j'}, \hat{i}_{j'}, \hat{i}_{j'+1}) \left( \boldsymbol{b}_0^{\theta^k}(\hat{i}_1) - \boldsymbol{b}_0^{\theta}(\hat{i}_1) \right) \right) \right\|_1 \cdot \pi^k(\Gamma), \tag{120}$$

where $\boldsymbol{b}_{j-1}^{\theta}(\Gamma_{j-1}) = \prod_{j'=1}^{j-1} \boldsymbol{B}_{j'}^{\theta}(\tilde{o}_{j'}, a_{j'}, \hat{i}_{j'}, \hat{i}_{j'+1}) \boldsymbol{b}_0^{\theta}(\hat{i}_1)$. Next, we use Lemma 32 in Liu et al. (2022) to bound the two terms on the right-hand side of Eq. (120). The tricky parts when apply their result are (i) how to construct the observable operator term and the controlled variable; (ii) how to

address the impact of the partial noisy observation. Specifically, we have that for all $j$,

$$\sum_{\Gamma} \left\| \left( \prod_{j'=j+1}^{H} \boldsymbol{B}_{j'}^{\theta^k}(\tilde{o}_{j'}, a_{j'}, \hat{i}_{j'}, \hat{i}_{j'+1}) \left( \boldsymbol{B}_{j}^{\theta^k}(\tilde{o}_j, a_j, \hat{i}_j, \hat{i}_{j+1}) - \boldsymbol{B}_{j}^{\theta}(\tilde{o}_j, a_j, \hat{i}_j, \hat{i}_{j+1}) \right) \cdot \boldsymbol{b}_{j-1}^{\theta}(\Gamma_{j-1}) \right) \right\|_1$$

$$\cdot \pi^k(\Gamma) \leq \frac{|\tilde{\mathbb{S}}|^{(d-\tilde{d})/2}}{\alpha} \sum_{\Gamma_j} \left\| \left( \boldsymbol{B}_{j}^{\theta^k}(\tilde{o}_j, a_j, \hat{i}_j, \hat{i}_{j+1}) - \boldsymbol{B}_{j}^{\theta}(\tilde{o}_j, a_j, \hat{i}_j, \hat{i}_{j+1}) \right) \cdot \boldsymbol{b}_{j-1}^{\theta}(\Gamma_{j-1}) \right\|_1 \cdot \pi^k(\Gamma_j), \tag{121}$$

where we consider $\left( \boldsymbol{B}_{j}^{\theta^k}(\tilde{o}_j, a_j, \hat{i}_j, \hat{i}_{j+1}) - \boldsymbol{B}_{j}^{\theta}(\tilde{o}_j, a_j, \hat{i}_j, \hat{i}_{j+1}) \right) \cdot \boldsymbol{b}_{j-1}^{\theta}(\Gamma_{j-1})$ as the variable $x$ in Lemma 32, and we have that

$$\sum_{\Gamma} \left\| \left( \prod_{j'=1}^{H} \boldsymbol{B}_{j'}^{\theta^k}(\tilde{o}_{j'}, a_{j'}, \hat{i}_{j'}, \hat{i}_{j'+1}) \left( \boldsymbol{b}_{0}^{\theta^k}(\hat{i}_1) - \boldsymbol{b}_{0}^{\theta}(\hat{i}_1) \right) \right) \right\|_1 \cdot \pi^k(\Gamma)$$

$$\leq \frac{|\tilde{\mathbb{S}}|^{(d-\tilde{d})/2}}{\alpha} \left\| \boldsymbol{b}_{0}^{\theta^k}(\hat{i}_1) - \boldsymbol{b}_{0}^{\theta}(\hat{i}_1) \right\|_1, \tag{122}$$

where we consider $\boldsymbol{b}_{0}^{\theta^k}(\hat{i}_1) - \boldsymbol{b}_{0}^{\theta}(\hat{i}_1)$ as the variable $x$ in Lemma 32. We note that in Eq. (121) and Eq. (122), the multiplicative factor depends on $|\tilde{\mathbb{S}}|^{d-\tilde{d}}$, which is size of the space of the sub-states that cannot be queried by the agent at each step.

By combining Eq. (120), Eq. (121) and Eq. (122), we have that

$$\sum_{\Gamma} \left\| \left( \boldsymbol{B}_{H}^{\theta^k}(\tilde{o}_H, a_H, \hat{i}_H, \hat{i}_{H+1}) \cdots \boldsymbol{B}_{1}^{\theta^k}(\tilde{o}_1, a_1, \hat{i}_1, \hat{i}_2) \boldsymbol{b}_{0}^{\theta^k}(\hat{i}_1) \right) \right.$$

$$\left. - \left( \boldsymbol{B}_{H}^{\theta}(\tilde{o}_H, a_H, \hat{i}_H, \hat{i}_{H+1}) \cdots \boldsymbol{B}_{1}^{\theta}(\tilde{o}_1, a_1, \hat{i}_1, \hat{i}_2) \boldsymbol{b}_{0}^{\theta}(\hat{i}_1) \right) \right\|_1 \cdot \pi^k(\Gamma)$$

$$\leq \frac{|\tilde{\mathbb{S}}|^{(d-\tilde{d})/2}}{\alpha} \left[ \sum_{j=1}^{H} \sum_{\Gamma_j} \left\| \left( \boldsymbol{B}_{j}^{\theta^k}(\tilde{o}_j, a_j, \hat{i}_j, \hat{i}_{j+1}) - \boldsymbol{B}_{j}^{\theta}(\tilde{o}_j, a_j, \hat{i}_j, \hat{i}_{j+1}) \right) \cdot \boldsymbol{b}_{j-1}^{\theta}(\Gamma_{j-1}) \right\|_1 \cdot \pi^k(\Gamma_j) \right.$$

$$\left. + \left\| \boldsymbol{b}_{0}^{\theta^k}(\hat{i}_1) - \boldsymbol{b}_{0}^{\theta}(\hat{i}_1) \right\|_1 \right]. \tag{123}$$

Then, by combining Eq. (119) and Eq. (123), we have that the TV distance can be upper bounded as follows,

$$\sum_{k=1}^{K} \sum_{\Gamma} \left| P_{\mathbb{P}^k, \bar{\mathbb{O}}^{\hat{i}, k}, \Delta_1^k}^{\pi^k}(\Gamma) - P_{\mathbb{P}, \bar{\mathbb{O}}^{\hat{i}}, \Delta_1}^{\pi^k}(\Gamma) \right|$$

$$\leq \frac{|\tilde{\mathbb{S}}|^{(d-\tilde{d})/2}}{\alpha} \left[ \sum_{k=1}^{K} \sum_{j=1}^{H} \sum_{\Gamma_j} \left\| \left( \boldsymbol{B}_{j}^{\theta^k}(\tilde{o}_j, a_j, \hat{i}_j, \hat{i}_{j+1}) - \boldsymbol{B}_{j}^{\theta}(\tilde{o}_j, a_j, \hat{i}_j, \hat{i}_{j+1}) \right) \cdot \boldsymbol{b}_{j-1}^{\theta}(\Gamma_{j-1}) \right\|_1 \cdot \pi^k(\Gamma_j) \right.$$

$$\left. + \left\| \boldsymbol{b}_{0}^{\theta^k}(\hat{i}_1) - \boldsymbol{b}_{0}^{\theta}(\hat{i}_1) \right\|_1 \right]. \tag{124}$$

**Step 3 (Bound the operator estimation error by the properties of MLE):**

According to Eq. (117) and Eq. (118), to upper bound the regret, we can focus on bounding the difference between each observable operator pair at each step on the right-hand side of Eq. (124). Note that such a difference mainly captures the operator estimation error under the estimated parameter $\theta^k$ and the true parameter $\theta$, which further depends on the property of MLE. Thus, we next upper bound such a difference by applying the properties of MLE. First, according to Lemma 8, we have that with high probability,

$$\sum_{\tau=1}^{K-1} \ln \frac{P_{\mathbb{P}, \bar{\mathbb{O}}^{\hat{i}}, \Delta_1}^{\pi^\tau}(\Gamma^\tau)}{P_{\mathbb{P}^K, \bar{\mathbb{O}}^{\hat{i}, K}, \Delta_1^K}^{\pi^\tau}(\Gamma^\tau)} \leq \beta. \tag{125}$$

Next, according to Proposition 14 in Liu et al. (2022), we have that with probability $1 - \delta$,

$$\sum_{\tau=1}^{K-1} \left( \sum_{\Gamma} \left| P^{\pi^\tau}_{\mathbb{P}^K, \bar{\mathbb{O}}^{\hat{i}, K}, \Delta_1^K}(\Gamma) - P^{\pi^\tau}_{\mathbb{P}, \bar{\mathbb{O}}^{\hat{i}}, \Delta_1}(\Gamma) \right| \right)^2 = O(\beta K). \tag{126}$$

By applying the Cauchy-Schwarz inequality to Eq. (126), we have

$$\sum_{\tau=1}^{K-1} \sum_{\Gamma} \left| P^{\pi^\tau}_{\mathbb{P}^K, \bar{\mathbb{O}}^{\hat{i}, K}, \Delta_1^K}(\Gamma) - P^{\pi^\tau}_{\mathbb{P}, \bar{\mathbb{O}}^{\hat{i}}, \Delta_1}(\Gamma) \right| = O(\sqrt{\beta K}). \tag{127}$$

This means that

$$\sum_{\tau=1}^{K-1} \sum_{\Gamma_h, \tilde{o}_{h+1}, \hat{i}_{h+1}} \left| P^{\pi^\tau}_{\mathbb{P}^K, \bar{\mathbb{O}}^{\hat{i}, K}, \Delta_1^K}(\Gamma_h, \tilde{o}_{h+1}, \hat{i}_{h+1}) - P^{\pi^\tau}_{\mathbb{P}, \bar{\mathbb{O}}^{\hat{i}}, \Delta_1}(\Gamma_h, \tilde{o}_{h+1}, \hat{i}_{h+1}) \right| = O(\sqrt{\beta K})$$

$$\sum_{\tau=1}^{K-1} \sum_{\tilde{o}_1, \hat{i}_1} \left| P^{\pi^\tau}_{\mathbb{P}^K, \bar{\mathbb{O}}^{\hat{i}, K}, \Delta_1^K}(\tilde{o}_1, \hat{i}_1) - P^{\pi^\tau}_{\mathbb{P}, \bar{\mathbb{O}}^{\hat{i}}, \Delta_1}(\tilde{o}_1, \hat{i}_1) \right| = O(\sqrt{\beta / K}). \tag{128}$$

We note that in the above marginalization of the distribution, we have taken the new impact of partial OSI and query $\hat{i}_h$ in our problem into consideration, because the query is part of the feedback trajectory and appears in the sum of Eq. (128). According to Eq. (115) and Eq. (128), we have that

$$\sum_{\tau=1}^{K-1} \sum_{\Gamma_h} \left\| \boldsymbol{b}_h^{\theta^K}(\Gamma_h) - \boldsymbol{b}_h^{\theta}(\Gamma_h) \right\|_1 \cdot \pi^\tau(\Gamma_h) = O(\sqrt{\beta K})$$

$$\left\| \boldsymbol{b}_0^{\theta^K}(\Gamma_h) - \boldsymbol{b}_0^{\theta}(\Gamma_h) \right\|_1 = O(\sqrt{\beta / K}), \tag{129}$$

where $\boldsymbol{b}_h^{\theta^K}(\Gamma_h) = \prod_{h'=1}^{h} \boldsymbol{B}_{h'}^{\theta}(\tilde{o}_{h'}, a_{h'}, \hat{i}_{h'}, \hat{i}_{h'+1}) \boldsymbol{b}_0^{\theta^K}(\hat{i}_1)$ and $\boldsymbol{b}_h^{\theta}(\Gamma_h) = \prod_{h'=1}^{h} \boldsymbol{B}_{h'}^{\theta}(\tilde{o}_{h'}, a_{h'}, \hat{i}_{h'}, \hat{i}_{h'+1}) \boldsymbol{b}_0^{\theta}(\hat{i}_1)$.

Recall from Eq. (124) that our goal is to bound the following two terms:

$$\sum_{k=1}^{K-1} \sum_{\Gamma_j} \left\| \left( \boldsymbol{B}_j^{\theta^K}(\tilde{o}_j, a_j, \hat{i}_j, \hat{i}_{j+1}) - \boldsymbol{B}_j^{\theta}(\tilde{o}_j, a_j, \hat{i}_j, \hat{i}_{j+1}) \right) \cdot \boldsymbol{b}_{j-1}^{\theta}(\Gamma_{j-1}) \right\|_1 \cdot \pi^k(\Gamma_j),$$

and

$$\left\| \boldsymbol{b}_0^{\theta^K}(\hat{i}_1) - \boldsymbol{b}_0^{\theta}(\hat{i}_1) \right\|_1 .$$

The second equation in Eq. (129) already gives the upper bound for the second term $\left\| \boldsymbol{b}_0^{\theta^K}(\hat{i}_1) - \boldsymbol{b}_0^{\theta}(\hat{i}_1) \right\|_1$. Thus, we next focus on bounding the first term using the first equation in Eq. (129). To prove this, we first apply the triangle inequality to this term as follows,

$$\sum_{k=1}^{K-1} \sum_{\Gamma_j} \left\| \left( \boldsymbol{B}_j^{\theta^K}(\tilde{o}_j, a_j, \hat{i}_j, \hat{i}_{j+1}) - \boldsymbol{B}_j^{\theta}(\tilde{o}_j, a_j, \hat{i}_j, \hat{i}_{j+1}) \right) \cdot \boldsymbol{b}_{j-1}^{\theta}(\Gamma_{j-1}) \right\|_1 \cdot \pi^k(\Gamma_j)$$

$$\leq \sum_{k=1}^{K-1} \sum_{\Gamma_j} \left\| \boldsymbol{B}_j^{\theta^K}(\tilde{o}_j, a_j, \hat{i}_j, \hat{i}_{j+1}) \cdot \left( \boldsymbol{b}_{j-1}^{\theta^K}(\Gamma_{j-1}) - \boldsymbol{b}_{j-1}^{\theta}(\Gamma_{j-1}) \right) \right\|_1 \cdot \pi^k(\Gamma_j)$$

$$+ \sum_{k=1}^{K-1} \sum_{\Gamma_j} \left\| \boldsymbol{B}_j^{\theta^K}(\tilde{o}_j, a_j, \hat{i}_j, \hat{i}_{j+1}) \boldsymbol{b}_{j-1}^{\theta^K}(\Gamma_{j-1}) - \boldsymbol{B}_j^{\theta}(\tilde{o}_j, a_j, \hat{i}_j, \hat{i}_{j+1}) \boldsymbol{b}_{j-1}^{\theta}(\Gamma_{j-1}) \right\|_1 \cdot \pi^k(\Gamma_j). \tag{130}$$

By applying the first equation in Eq. (129) and Lemma 32 to Eq. (130), we have that

$$\sum_{k=1}^{K-1} \sum_{\Gamma_j} \left\| \left( \boldsymbol{B}_j^{\theta^K}(\tilde{o}_j, a_j, \hat{i}_j, \hat{i}_{j+1}) - \boldsymbol{B}_j^{\theta}(\tilde{o}_j, a_j, \hat{i}_j, \hat{i}_{j+1}) \right) \cdot \boldsymbol{b}_{j-1}^{\theta}(\Gamma_{j-1}) \right\|_1 \cdot \pi^k(\Gamma_j)$$

$$= O\left( \frac{|\tilde{\mathbb{S}}|^{(d-\bar{d})/2}}{\alpha} \sqrt{K\beta} \right). \tag{131}$$

**Step 4 (Deriving final regret using $l_1$ eluder argument):**

In this final step, we prove the final regret, which depends on the performance of OMLE-POSI in a new episode $K + 1$, based on the upper bound that we have proved for the previous $K$ episodes in Eq. (131). To prove this, we use the $l_1$ eluder argument. Specifically, we first use $[\mathbb{A}]_r$ to denote the $r$-th row of matrix $\mathbb{A}$. Next, Eq. (131) implies that

$$\sum_{k=1}^{K-1} \sum_{\Gamma_j:(\tilde{o}_j, a_j, \hat{i}_j, \hat{i}_{j+1})=(\tilde{o},a,\hat{i},\hat{i}')} \sum_{l=1}^{O} \left| \left[ \left( \boldsymbol{B}_j^{\theta^K}(\tilde{o}, a, \hat{i}, \hat{i}') - \boldsymbol{B}_j^{\theta}(\tilde{o}, a, \hat{i}, \hat{i}') \right) \bar{\mathbb{O}}_j \right]_r \cdot \bar{\mathbb{O}}_j^{\dagger} \boldsymbol{b}_{j-1}^{\theta}(\Gamma_{j-1}) \right|$$
$$\cdot \pi^k(\Gamma_j) = O(\frac{|\tilde{\mathbb{S}}|^{(d-\tilde{d})/2}}{\alpha} \sqrt{K\beta}). \tag{132}$$

Then, to apply the $l_1$ eluder argument in Proposition 22 of Liu et al. (2022), we can consider $\left[ \left( \boldsymbol{B}_j^{\theta^K}(\tilde{o}_j, a_j, \hat{i}_j, \hat{i}_{j+1}) - \boldsymbol{B}_j^{\theta}(\tilde{o}_j, a_j, \hat{i}_j, \hat{i}_{j+1}) \right) \bar{\mathbb{O}}_j \right]_r$ as the $w_{t,l}$ variable and consider $\bar{\mathbb{O}}_j^{\dagger} \boldsymbol{b}_{j-1}^{\theta}(\Gamma_{j-1}) \cdot \pi^k(\Gamma_j)$ as the $x_{t,i}$ variable. Then, we have that,

$$\sum_{k=1}^{K} \sum_{\Gamma_j:(\tilde{o}_j, a_j, \hat{i}_j, \hat{i}_{j+1})=(\tilde{o},a,\hat{i},\hat{i}')} \sum_{l=1}^{O} \left| \left[ \left( \boldsymbol{B}_j^{\theta^k}(\tilde{o}, a, \hat{i}, \hat{i}') - \boldsymbol{B}_j^{\theta}(\tilde{o}, a, \hat{i}, \hat{i}') \right) \bar{\mathbb{O}}_j \right]_r \cdot \bar{\mathbb{O}}_j^{\dagger} \boldsymbol{b}_{j-1}^{\theta}(\Gamma_{j-1}) \right|$$
$$\cdot \pi^k(\Gamma_j) = O(\frac{|\tilde{\mathbb{S}}|^{3(d-\tilde{d})/2} H^2}{\alpha} \sqrt{K\beta}), \tag{133}$$

where we slightly abuse the notation $K$ and use it to denote any episode. By taking the sum of both sides of Eq. (133) over all steps $j$ and all possible feedback trajectory $\Gamma_j$, we have that

$$\sum_{k=1}^{K} \left( \sum_{j=1}^{H} \sum_{\Gamma_j} \left\| \left[ \left( \boldsymbol{B}_j^{\theta^k}(\tilde{o}_j, a_j, \hat{i}_j, \hat{i}_{j+1}) - \boldsymbol{B}_j^{\theta}(\tilde{o}_j, a_j, \hat{i}_j, \hat{i}_{j+1}) \right) \right] \cdot \boldsymbol{b}_{j-1}^{\theta}(\Gamma_{j-1}) \right\|_1 \cdot \pi^k(\Gamma_j) \right.$$
$$\left. + \left\| \boldsymbol{b}_0^{\theta^k}(\hat{i}_1) - \boldsymbol{b}_0^{\theta}(\hat{i}_1) \right\|_1 \right) = O(\frac{|\tilde{\mathbb{S}}|^{3(d-\tilde{d})/2} OAH^3}{\alpha} \sqrt{K\beta}), \tag{134}$$

where the big-$O$ notation hides the constant and logarithmic terms including[2] $\binom{d}{\tilde{d}}$.

Finally, by combining Eq. (118), Eq. (124) and Eq. (134), we have that the regret of OMLE-POSI can be upper bounded as follows,

$$Reg^{OMLE-POSI}(K) = \tilde{O} \left( \frac{|\tilde{\mathbb{S}}|^{2(d-\tilde{d})} OAH^4 \left( \binom{d}{\tilde{d}} \right)^2}{\alpha^2} \sqrt{K(|\tilde{\mathbb{S}}|^{2d} A + |\tilde{\mathbb{S}}|^{d-\tilde{d}} O \binom{d}{\tilde{d}}))} \right),$$
$$= \tilde{O} \left( \frac{|\tilde{\mathbb{S}}|^{2d-\tilde{d}} OAH^4}{\alpha^2} \sqrt{K(|\tilde{\mathbb{S}}|^{2d} A + |\tilde{\mathbb{S}}|^{d-\tilde{d}/2} O))} \right), \tag{135}$$

where the last inequality is because $|\tilde{\mathbb{S}}| \geq (d/\tilde{d})^2$, $\binom{d}{\tilde{d}} = O((d/\tilde{d})^{\tilde{d}})$, and the big-$O$ notation hides the constant and logarithmic terms.

$\square$

## H  PROOF OF THEOREM 4

*Proof.* To prove the regret lower-bound in Theorem 4, we construct a new special state transition, such that even with partial online state information (OSI), all combinations of sub-states $\phi_i(s)$ must be explored to achieve a regret that is sub-linear in $K$. Formally, to achieve this, we construct a new special hard instance as follows.

---

[2]Since in our case, the observable operator is not only for the observation-action pair $(\tilde{o}, a)$, but also for the two consecutive query pair $(\hat{i}_j, \hat{i}_{j+1})$, there will an new factor $\left( \binom{d}{\tilde{d}} \right)^2$ after taking the sum.

**A new hard instance:** We consider $S = |\tilde{\mathbb{S}}|^d$ states, i.e., $s(1), s(2), ..., s(S)$. Each of them is one specific permutation of the sub-state values. Let us consider $\tilde{\mathbb{S}} = \{0, 1\}$ and $d = 3$ as an example. Then, there are 8 states, i.e.,

$$\vec{\phi}(s(1)) = \begin{bmatrix} 0 \\ 0 \\ 0 \end{bmatrix}, \vec{\phi}(s(2)) = \begin{bmatrix} 1 \\ 0 \\ 0 \end{bmatrix}, \vec{\phi}(s(3)) = \begin{bmatrix} 0 \\ 1 \\ 0 \end{bmatrix}, \vec{\phi}(s(4)) = \begin{bmatrix} 0 \\ 0 \\ 1 \end{bmatrix},$$

$$\vec{\phi}(s(5)) = \begin{bmatrix} 1 \\ 1 \\ 0 \end{bmatrix}, \vec{\phi}(s(6)) = \begin{bmatrix} 1 \\ 0 \\ 1 \end{bmatrix}, \vec{\phi}(s(7)) = \begin{bmatrix} 0 \\ 1 \\ 1 \end{bmatrix}, \vec{\phi}(s(8)) = \begin{bmatrix} 1 \\ 1 \\ 1 \end{bmatrix}. \tag{136}$$

Moreover, there are $A$ actions, i.e., $a(1), a(2), ..., a(A)$, where $A \leq S$. In each episode $k$, the learner starts from a fixed initial state $s_1 = [x_1, x_2, ..., x_d]^{\mathrm{T}}$, e.g., $s_1 = s(1) = [0, 0, 0]^{\mathrm{T}}$, i.e.,

$$\Delta_1(s_1) = \begin{cases} 1, & \text{if } s_1 = s(1); \\ 0, & \text{otherwise.} \end{cases} \tag{137}$$

Based on these, we construct a new special state transition as follows.

*State transitions:* In each episode, after taking an action $a(j)$ at the first step $h = 1$, state $s_1$ transitions to one specific new state at step $h = 2$ deterministically. Specifically, there are $A$ states at step $h = 2$. After taking action $a(j)$, state $s_1 = s(1)$ transitions and only transitions to state $s_2 = s(j)$ with probability 1, i.e.,

$$\mathbb{P}_1(s_2|s_1, a(j)) = \begin{cases} 1, & \text{if } s_2 = s(j); \\ 0, & \text{otherwise.} \end{cases} \tag{138}$$

Let us consider $A = 2$ as an example. Then,

$$\mathbb{P}_1(s_2|s_1, a(1)) = \begin{cases} 1, & \text{if } s_2 = s(1); \\ 0, & \text{if } s_2 \neq s(1). \end{cases}$$

$$\mathbb{P}_1(s_2|s_1, a(2)) = \begin{cases} 1, & \text{if } s_2 = s(2); \\ 0, & \text{if } s_2 \neq s(2). \end{cases} \tag{139}$$

For the next $\log_A S - 1$ steps, the state transitions are as follows. (In the above example with $|\tilde{\mathbb{S}}| = 2$, $d = 3$ and $A = 2$, we have $\log_A S = 3$.) After taking action $a(j)$ at step $h$, the $m$-th state transitions to one specific new state at step $h + 1$ deterministically. Specifically, there are $A^{h-1}$ states at step $h$. by taking action $a(j)$, state $s_{h,m}$ transitions to state $s_{h+1} = s(A(m-1) + j)$ with probability 1, i.e.,

$$\mathbb{P}_2(s_3|s(m), a(j)) = \begin{cases} 1, & \text{if } s_3 = s(A(m-1) + j); \\ 0, & \text{otherwise.} \end{cases} \tag{140}$$

Let us still consider the aforementioned example. Then, at step $h = 2$, we have

$$\mathbb{P}_2(s_3|s_2, a(1)) = \begin{cases} 1, & \text{if } s_2 = s(1) \text{ and } s_3 = s(1); \\ 1, & \text{if if } s_2 = s(2) \text{ and } s_3 = s(3); \\ 0, & \text{otherwise.} \end{cases}$$

$$\mathbb{P}_2(s_3|s_2, a(2)) = \begin{cases} 1, & \text{if } s_2 = s(1) \text{ and } s_3 = s(2); \\ 1, & \text{if if } s_2 = s(2) \text{ and } s_3 = s(4); \\ 0, & \text{otherwise.} \end{cases} \tag{141}$$

Finally, for the rest of steps, the number of states keeps to be equal to $S$. For the $j$-th state $s(m)$ at each episode, only by taking action $a(\max\{mod(m, A), 1\})$, the next state is still $s(m)$ with probability 1. Otherwise, the next state is $s(1)$. Formally, we let

$$\mathbb{P}_{h(s(m))}(s_{h(s(m)+1)}|s(m), a(j)) = \begin{cases} 1, & \text{if } j = \max\{mod(m, A), 1\} \text{ and } s_{h(s(m)+1)} = s(m); \\ 1, & \text{if } j \neq \max\{mod(m, A), 1\} \text{ and } s_{h(s(m)+1)} = s(1); \\ 0, & \text{otherwise,} \end{cases} \tag{142}$$

where $h(s(m))$ denotes the step of state $s(m)$ that is under consideration and $mod(m, A)$ denotes the remainder of dividing $m$ by $A$.

*Reward function:* We let the reward be $r_h = 0$ for all the first $\log_A S$ steps. For the later steps, only when taking action $a(\max\{mod(m, A), 1\})$ at state $s(m)$, the reward is $r_h = Ber(\frac{1}{2})$ for some $h$, while uniformly randomly pick one of them to be $j^*$ and $r_h(s(m^*), a(\max\{mod(m^*, A), 1\})) = Ber(\frac{1}{2} + \epsilon)$. Formally, there exists a step $h_0 > \log_A S$, such that,

$$r_h(s(m), a(j)) = \begin{cases} Ber(\frac{1}{2}), & \text{if } j = \max\{mod(m, A), 1\}, h = h_0, \text{ and } m \neq m^*; \\ Ber(\frac{1}{2} + \epsilon), & \text{if } j = \max\{mod(m, A), 1\}, h = h_0, \text{ and } m = m^*; \\ 0, & \text{otherwise.} \end{cases} \quad (143)$$

Note that in such a instance, even with partial OSI, the online learner has to try all possible action sequence to figure out which one provide the slightly higher reward. Therefore, learning in this instance with partial online state information is equivalent to learning which action sequence is the best one. Since there are at least $\Omega(|\tilde{\mathbb{S}}|^d AH)$ of action sequences with expected reward $r_h \geq \frac{1}{2}$. Thus, the regret lower bound $\tilde{\Omega}(\sqrt{MK})$ (where $M$ denotes the total number of arms) in bandit learning implies that the regret lower bound in our case is $Reg^\pi(K) \geq \tilde{\Omega}\left(\sqrt{AH} \cdot |\tilde{\mathbb{S}}|^{d/2} \cdot \sqrt{K}\right)$ for any algorithm $\pi$.

$\square$

