# OpenReview forum: "Theoretical Hardness and Tractability of POMDPs in RL with Partial Online State Information"
_ICLR.cc/2024/Conference — Submitted to ICLR 2024_

### Official Review · Reviewer_VorK · 2023-10-31

**Soundness:** 3 good
**Presentation:** 3 good
**Contribution:** 3 good
**Rating:** 6
**Confidence:** 4

**Summary:**

This paper studies the partially observable reinforcement learning with certain feedback structure (partial online state information, POSI). Given that POSI is not sufficient for statistical learnability, the authors identify two learnable problem classes of interest with POSI and extra structural assumptions (independent sub-states / revealing condition) by establishing regret upper bounds and lower bounds.

**Strengths:**

- The hardness result for POSI (Theorem 1) uses a novel and interesting construction based on combinatorics lock. The clever construction demonstrates that partially observable RL is still hard even the agent can actively query (part of) the latent state information, which is itself a meaningful message.

- The motivation of considering class 1 & 2 is very clear, as the authors provide various empirical works where the structure of class 1 / 2 naturally arises.

- Algorithm 1 adopts a novel strategy for determining the substates to be queried based on exponential weight update, which seems promising.

**Weaknesses:**

The two problem classes are studied in a case-by-case manner (both algorithms and analysis are very different). It would be a natural question whether there is a connection between these two. Such a connection is important because it will provide a more unified understanding of partially observable RL with POSI. In this paper, there lacks an investigation (or at least a discussion) on this connection (either positive or negative).

**Questions:**

1. For problem class 1, the structure of the optimal policy is not very clear. If my understanding is correct, the guarantee of algorithm 1 implicitly implies that there is an optimal policy that proceed as follows: for each step h, sample a substate i from a fixed distribution p, query i and take an action according to \phi_i(s_h). In my opinion, this result is non-trivial, and stating this result can help the reader better understand the motivation of Algorithm 1.

2. For theorem 3, the authors highlight that the upper bound is "decreases exponentially as $\tilde{d}$ increases". I am a little bit confused of this point, because there is a possibility that such a decreasing upper bound is an artifact of the algorithm design or the analysis. More concretely, it is possible that the minimax optimal regret actually scales with $|\tilde{\mathbb{S}}|^d$ (as the lower bound), which does not decrease as $\tilde{d}$ increases. There also lacks a discussion of the revealing condition $\alpha$ which only appears in the upper bound side, yet it is known that for revealing POMDP the minimax optimal regret actually scales with $\alpha^{-1}$.

---

> ### Author Response · Authors · 2023-11-20
> **Response to Reviewer VorK (part 1)**
>
> We thank the reviewer for providing the helpful review! We have addressed the reviewer’s helpful comments and modified the paper accordingly. Please note that in the revised paper, we highlighted our changes by blue-colored texts.
>
> **Q1:** The two problem classes are studied in a case-by-case manner (both algorithms and analysis are very different). It would be a natural question whether there is a connection between these two. Such a connection is important because it will provide a more unified understanding of partially observable RL with POSI. In this paper, there lacks an investigation (or at least a discussion) on this connection (either positive or negative).
>
> **A1:** Many thanks for the suggestion! In fact, each of two classes here becomes learnable due to different natures in their transition model structures, and hence requires very different algorithm designs to handle these specialities. So it seems to be nontrivial to unify the algorithm design for them. Below we highlight the two key differences between Class 1 and Class 2 that make it difficult to unify the approaches for them. The first difference is the structure of the state transition kernel $\mathbb{P}$. In Class1, the state transition probability is assumed to be $\mathbb P_h(\cdot \vert s_h^k,a_h^k) = \prod_{i=1}^{d} \mathbb P_{h,i}(\phi_i(\cdot) \vert \phi_i(s_h^k),a_h^k)$. That is, it is the product of independent transition kernels of sub-states. In contrast, in Class 2, we do not need such a requirement. The second difference is the additional noisy observation $\tilde{o}$. In Class 2, in addition to the partial OSI, the agent receives the partial noisy observation $\tilde o_h^k$ for the $d-\tilde{d}$ sub-states that are not queried, where $\tilde o_h^k$ is generated according to the partial emission probability $\tilde{\mathbb O_h^{\hat i_h^k}}\left(\cdot \big\vert \{\phi_i(s_h^k)\}_{\{i\notin \hat i_h^k\}}\right)$. In contrast, Class 1 does not require any additional noisy observation $\tilde{o}$ at all. In Appendix B.2 of the revision, we provide a comparison between the two problem classes.
>
> **Q2:** For problem class 1, the structure of the optimal policy is not very clear. If my understanding is correct, the guarantee of algorithm 1 implicitly implies that there is an optimal policy that proceed as follows: for each step $h$, sample a substate $i$ from a fixed distribution $p$, query $i$ and take an action according to $\phi_i(s_h)$. In my opinion, this result is non-trivial, and stating this result can help the reader better understand the motivation of Algorithm 1.
>
> **A2:** Many thanks for the suggestion. Yes, at each step $h$, the optimal policy queries a sub-state $i$ according to a fixed distribution $p$, and receives the partial OSI for this queried sub-state. Then, the agent takes an action according to $\phi_i(s_h)$. In the revision, we have added this explanation before introducing Algorithm 1.
>
> **Q3:** For theorem 3, the authors highlight that the upper bound "decreases exponentially as $\tilde{d}$ increases”. I am a little bit confused of this point, because there is a possibility that such a decreasing upper bound is an artifact of the algorithm design or the analysis. More concretely, it is possible that the minimax optimal regret actually scales with $\tilde{\mathbb{S}}^d$ (as the lower bound), which does not decrease as $\tilde{d}$ increases.
>
> **A3:** Thanks. Yes, our achieved regret upper bound decreases exponentially as $\tilde{d}$ increases. We agree with the reviewer that finding a lower bound that depends on $\tilde{\mathbb{S}}^d$ would be interesting. However, this is an extremely challenging problem, and we are not certain that such a lower bound even exists.To elaborate further, to develop a lower bound that increases when $\tilde{d}$ decreases, one needs to construct special (transitions of) states, such that the sub-states that are not queried keep adding uncertainty to any algorithm. From an analytical point of view, such uncertainty increases the KL divergence between the probability measures under the case with such a lower-bound instance and the case with fictitiously same sub-states. However, in our problem setting, the agent can query any sub-states freely at each time, which makes it highly non-trivial to construct such a lower-bound instance.
>
> We conjecture that a stronger lower bound depending on the query capability would be $\tilde{\Omega}\left( \sqrt{AH} \cdot \vert \tilde{\mathbb{S}} \vert^{(d-\tilde{d})/2} \cdot \sqrt{K} / \alpha \right)$, and leave this as a future open question. Right after Theorem 4 in the revision, we have added this explanation.

---

> ### Author Response · Authors · 2023-11-20
> **Response to Reviewer VorK (part 2)**
>
> **Q4:** There also lacks a discussion of the revealing condition $\alpha$ which only appears in the upper bound side, yet it is known that for revealing POMDP the minimax optimal regret actually scales with $\alpha^{-1}$.
>
> **A4:** In Appendix B of the revision, we have added a discussion for the dependency of the regret lower bound on the revealing condition $\alpha$. It may be possible to develop such a lower bound, which of course will need further rigorous technical development. In Particular, a very special sub-state transition is needed in our case to make partial OSI not useful for the learning agent. We leave this as an interesting future work.
>
> Finally, we thank the reviewer again for the helpful comments and suggestions for our work. If our response resolves your concerns to a satisfactory level, we kindly ask the reviewer to consider raising the rating of our work. Certainly, we are more than happy to address any further questions that you may have during the discussion period.

---

### Official Review · Reviewer_oftj · 2023-10-31

**Soundness:** 2 fair
**Presentation:** 2 fair
**Contribution:** 2 fair
**Rating:** 5
**Confidence:** 4

**Summary:**

This paper delves into the realm of partially observable reinforcement learning. The authors initially establish that achieving a near-optimal policy solution for POMDPs necessitates an exponentially increasing sample complexity unless full online state information (OSI) is readily available. Moreover, the authors demonstrate that POMDPs with only partial OSI become tractable when certain additional assumptions are made, such as independent sub-states or the revealing assumption.

**Strengths:**

The domain of partially observable reinforcement learning holds significant interest for the RL theory community, with the identification of new tractable POMDPs being of paramount importance. This paper aims to present a comprehensive set of results for POMDPs with online state information.

**Weaknesses:**

One of the principal issues with this paper is its writing quality, marred by numerous typographical errors that impede comprehension. Some concerns and questions are outlined in the questions part.


While I can grasp the main storyline and contributions of this work to some extent, its subpar writing hinders my ability to follow and validate its accuracy. Therefore, I firmly believe that this paper requires a comprehensive revision to enhance its clarity and coherence.

**Questions:**

- The reward function $r_h$ remains independent of the episode index $k$," but the reward $r_h^k$ is frequently referenced (e.g., Figure 1, Section 2.1, Section 2.2...).

- In Section 2.3, it is unclear whether the authors intend to use $\hat{\Phi}_h$ or $\hat{\Phi}_h^k$."

- Regarding the definition of $V^{\pi^k}$, the expectation is seemingly taken with respect to $\pi_q^k$ rather than $\pi_p^k$. The authors should also provide clarity that $\pi^k = (\pi_q^k, \pi_a^k)$.

- The definition of regret raises questions. Why do the authors employ $Reg^{\pi}(K)$ instead of $Reg(K)?$ If the authors aim to emphasize the dependency on the executed policies, it should be expressed as $Reg^{\pi^{1:K}}(K).$

- The statement of Theorem 1 appears non-rigorous as it lacks a precise definition of "with only partial online state information."

- The definition of $\mathbb{P}\_h^k$ in (3) is unclear. The counter $\mathcal{N}^k$ also seems to depend on $h.$ The definition of $\mathcal{N}^k(\phi_{\hat{i}}(s), a, \phi_{\hat{i}}(s'))$ should be provided;  and the $\mathbb{P}\_h^k$ should be replaced by $\mathbb{P}\_h^k(\phi_{\hat{i}}(s') | \phi_{\hat{i}}(s), a)$?



- The proposed algorithms appear to be a fusion of the strategic querying mechanism and existing methods, which include value iteration and OMLE. It's evident that the strategic querying mechanism stands out as the primary novelty in this work. However, the paper lacks a detailed and thorough elaboration of this mechanism. For instance, deferring the definition of $\hat{r}_h^k$ to the appendix without accompanying explanation seems to be a suboptimal choice, as it hinders a clear understanding of this critical component. Providing more clarity and detail on this novel approach would greatly enhance the paper's readability.


- Regarding Theorem 6: Why can't the cases of $\tilde{d} = 1$ and $\tilde{d} \ge 2$ be unified? Why is Theorem 6 unable to encompass the results for the case of $\tilde{d}=1$"? Perhaps presenting the general case for $\tilde{d}$ would better illustrate the contributions of this paper.

- Another minor comment: it would be much more interesting if the lower bound in Theorem 4 depends on $\tilde{d}$.

There are a few issues I identified in the main paper. I hope the authors will thoroughly review the paper (including the Appendix) and make comprehensive revisions, aiming to make the paper easy to follow.

---

> ### Author Response · Authors · 2023-11-20
> **Response to Reviewer oftj (part 1)**
>
> We thank the reviewer for providing the helpful review! We have addressed the reviewer’s helpful comments and modified the paper accordingly. Please note that in the revised paper, we highlighted our changes by blue-colored texts.
>
> **Q1:** One of the principal issues with this paper is its writing quality, marred by numerous typographical errors that impede comprehension.
>
> **A1:** We have carefully addressed the reviewer’s comments one-by-one below and polished the paper again. Should the reviewer have any further questions, we are more than happy to provide detailed responses.
>
> **Q2:** The reward function $r_h$ remains independent of the episode index $k$, but the reward $r_h^k$ is frequently referenced (e.g., Figure 1, Section 2.1, Section 2.2...).
>
> **A2:** Different from the reward **function** $r_h$ that is independent of the episode index $k$, the reward **value** $r_h^k$ received in each episode $k$ could be different, e.g., depending on the observation-action pair $(o_h^k,a_h^k)$ visited in episode $k$. Thus, we had used the superscript $k$ to emphasize this difference. To address the reviewer’s concern, in Figure 1 and Section 2 of the revision, we have removed the superscript $k$ in the reward-value notation, i.e., changed $r_h^k$ to $r_h$.
>
> **Q3:** In Section 2.3, it is unclear whether the authors intend to use $\hat{\Phi}_h$ or $\hat{\Phi}_h^k$.
>
> **A3:** We did intend to use $\hat{\Phi}_h$, not $\hat{\Phi}_h^k$ in our paper. This is because we use $\hat{\Phi}_h$ to denote the **feedback space**, which should be independent of the episode index $k$.
>
> **Q4:** Regarding the definition of $V^{\pi^k}$, the expectation is seemingly taken with respect to $\pi_q^k$ rather than $\pi_p^k$. The authors should also provide clarity that $\pi^k = (\pi_q^k,\pi_a^k)$.
>
> **A4:** Thanks. In Section 2.3 of the revision, we have changed $\pi_p^k$ to $\pi_q^k$, and clarified that $\pi^k = (\pi_q^k,\pi_a^k)$.
>
> **Q5:** Why do the authors employ $Reg^{\pi}(K)$ instead of $Reg(K)$? If the authors aim to emphasize the dependency on the executed policies, it should be expressed as $Reg^{\pi^{1:K}}(K)$.
>
> **A5:** Thanks. In Equation (1) of the revision, we have changed $Reg^{\pi}(K)$ to $Reg^{\pi^{1:K}}(K)$.
>
> **Q6:** The statement of Theorem 1 appears non-rigorous as it lacks a precise definition of "with only partial online state information."
>
> **A6:** Please note that the definition of “with only partial online state information” is in Section 2.2 and is illustrated in Figure 1(b). Specifically, “with only partial online state information” means that at each step, the agent can query a subset of $\tilde{d}$ sub-states, where $1 \leq \tilde{d} < d$. Then, the partial online state information, i.e., the precise information of the queried sub-states is revealed to the agent. Further, note that if $\tilde{d} = d$, then *partial* online state information becomes *full* online state information.
>
> **Q7:** The definition of $\mathbb{P_h^k}$ in (3) is unclear. The counter $\mathcal{N}^k$ also seems to depend on $h$. The definition of $\mathcal{N}^k(\phi_{\hat{i}}(s),a,\phi_{\hat{i}}(s'))$ should be provided; and the $\mathbb{P_h^k}$ should be replaced by $\mathbb{P_h^k}(\phi_{\hat{i}}(s') | \phi_{\hat{i}}(s),a)$?
>
> **A7:** Many thanks. In Equation (3) of the revision, we have (i) changed $\mathbb{P_h^k}$ to $\mathbb{P_h^k}(\phi_{\hat{i}}(s') | \phi_{\hat{i}}(s),a)$; (ii) changed $\mathcal{N}^k$ to $\mathcal{N_h^k}$; and (iii) clarified that $\mathcal{N_h^k}(\phi_{\hat{i}}(s),a,\phi_{\hat{i}}(s'))$ is the number of times the state-action-state triplet $(\phi_{\hat{i}}(s),a,\phi_{\hat{i}}(s'))$ has been visited at step $h$ up to episode $k$.

---

> ### Author Response · Authors · 2023-11-20
> **Response to Reviewer oftj (part 2)**
>
> **Q8:** The proposed algorithms appear to be a fusion of the strategic querying mechanism and existing methods, which include value iteration and OMLE. It's evident that the strategic querying mechanism stands out as the primary novelty in this work. However, the paper lacks a detailed and thorough elaboration of this mechanism. For instance, deferring the definition of $\hat{r}_h^k$ to the appendix without accompanying explanation seems to be a suboptimal choice, as it hinders a clear understanding of this critical component. Providing more clarity and detail on this novel approach would greatly enhance the paper's readability.
>
> **A8:** We had deferred the definition of $\hat{r}_h^k$ to the appendix because (i) the more important ideas are how to update the weight and choose the parameters for the weight, and (ii) we felt that there was not enough space for explaining $\hat{r}_h^k$.
>
> However, in light of the reviewer’s concern, in Section 4 of the revision, we have revised the algorithm description and added the definition of $\hat r_h^k$ as follows: “where the estimated reward $\hat r_h^{k-1}(\phi_{i}(s_h^{k-1}),a_h^{k-1}) = r_h(\phi_{i}(s_h^{k-1}),a_h^{k-1})$, if $i = i_h^k$; and $\hat r_h^{k-1}(\phi_{i}(s_h^{k-1}),a_h^{k-1}) = 0$, otherwise. Note that this is a new variant of the importance sampling method, where the new development lies in estimating the reward by exploiting partial OSI.”
>
> **Q9:** Regarding Theorem 6: Why can't the cases of $\tilde{d} = 1$ and $\tilde{d} \geq 2$ be unified? Why is Theorem 6 unable to encompass the results for the case of $\tilde{d} = 1$? Perhaps presenting the general case for $\tilde{d}$ would better illustrate the contributions of this paper.
>
> **A9:** Thanks for this question. The cases of $\tilde{d} = 1$ and $\tilde{d} \geq 2$ cannot be unified under our design. This is because the leading sub-state (see step-1 of algorithm 4) and the supporting sub-states (see step-4 of algorithm 4) are chosen in two different new ways. Thus, the global weight estimation in (56) and local weight estimation in (57) are fundamentally different, and cannot be reduced to the weight update in the case of $\tilde{d} = 1$. Moreover, due to these two different ideas used for weight updates, the analyses for these two cases (in Appendix E and Appendix F) are fundamentally different. It is unclear if these two cases can be unified, which appear to require a new sub-state querying idea. We leave this as a future work.
>
> Moreover, we choose to present the case with $\tilde{d} = 1$ in the main body because it is easier for readers to follow. Also, the allowed space is not good enough for a detailed and thorough elaboration of our results for the general case with $\tilde{d} \geq 2$, e.g., the algorithm design and regret. Thus, we chose to defer the explanation of the general case to the appendix.
>
> **Q10:** Another minor comment: it would be much more interesting if the lower bound in Theorem 4 depends on $\tilde{d}$.
>
> **A10:** Many Thanks. A lower bound depending on $\tilde{d}$ turns out to be quite challenging for the following reasons. To develop a lower bound that increases when $\tilde{d}$ decreases, one needs to construct special (transitions of) states, such that the sub-states that are not queried keep adding uncertainty to any algorithm. From an analytical point of view, such uncertainty increases the KL divergence between the probability measures under the case with such a lower-bound instance and the case with fictitiously same sub-states. However, in our problem setting, the agent can *query any sub-states freely at each time*, which makes it non-trivial to construct such a lower-bound instance.
>
> We conjecture that a stronger lower bound depending on the query capability would be $\tilde{\Omega}\left( \sqrt{AH} \cdot \vert \tilde{\mathbb{S}} \vert^{(d-\tilde{d})/2} \cdot \sqrt{K} / \alpha \right)$, and leave this as a future open question. Right after Theorem 4 of the revision, we have added this conjecture.
>
> **Q11:** There are a few issues I identified in the main paper. I hope the authors will thoroughly review the paper (including the Appendix) and make comprehensive revisions, aiming to make the paper easy to follow.
>
> **A11:** Many thanks for all your suggestions. We have carefully proofread the paper and revised the paper thoroughly, including the appendices. We hope the revised version is clearer for the reviewer. Please let us know if there are any other unclear places and we are more than happy to address them accordingly.
>
> Finally, we thank the reviewer again for the helpful comments and suggestions for our work. If our response clarifies your concerns to a satisfactory level, we kindly ask the reviewer to consider raising the rating of our work. Certainly, we are more than happy to address any further questions that you may have during the discussion period.

---

> > ### Comment · Reviewer_oftj · 2023-11-23
> >
> > Thank you for your detailed response. Your clarifications and revisions have improved the paper's readability. I have adjusted my score to reflect this positive development and your dedicated efforts. I continue to encourage the authors to further refine the paper for enhanced clarity and ease of understanding.

---

> > > ### Author Response · Authors · 2023-11-23
> > >
> > > We thank the reviewer for the response and increasing the score!

---

### Official Review · Reviewer_yDo3 · 2023-10-31

**Soundness:** 2 fair
**Presentation:** 3 good
**Contribution:** 3 good
**Rating:** 6
**Confidence:** 3

**Summary:**

This paper focuses on the benefits of online state information (OSI) in the learning of POMDPs. Previous work showed that full OSI in hindsight helps accelerate the exploration of unknown POMDPs. The main problem studied in the paper is whether partial OSI is beneficial for POMDPs, and to what extend are the benefits if the answer is yes. The partial OSI setting means the agent can actively query part of the state at each step. For the negative results, the paper shows that partial OSI cannot prevent exponential sample complexity in general by constructing a hard family of POMDPs. For positive results, it identifies two subclasses of partial OSI POMDPs that is tractable and provides sample efficient algorithms to learn the optimal policy with $\sqrt{K}$ regret ($K$ is the number of episodes).

**Strengths:**

1. The setting studied in the paper is common in some real-world problems. The paper also provides some motivating examples for the partial OSI model.

2. The techniques to develop the lower bound instances and the algorithmic design of pessimistic-optimistic exploration strategy for the query policy and execution policy are appreciated.

**Weaknesses:**

1. The exact setting studied in the paper seems to be inconsistent with the motivation of introducing OSI in POMDPs. As in previous work, the OSI in the hindsight is common some applications, such as data center scheduling, where the state is revealed after the action is take (so the effect of the action can be observed), However, the query model in this paper allows the agent to query part of the state before the action is taken. Even the motivating examples (e.g., the autonomous driving) seems to advocate the former setup, which reveals the state after taking actions. Although the conclusions of the paper may still hold in the case that the query happens after taking actions, it is appreciated to discuss about the model that is better aligned with the real world.

2. Some of the claims and proofs in the paper are not clear enough, leaving potential risks for the soundness of the paper. In the definition of tractable class 1, the transition kernel can be factored as the multiplication of $d$ different subtransition according to each substate. Algorithm 3 does not clearly show how to estimate those $d$ subtransitions, instead it looks like the subtransitions are the same for each $i \in [d]$, which enables us to estimate this transition using all the adjacent revealed substates within one episode no matter whether the adjacent steps choose the same dimension of substate or not. This is not achievable when the subtransition for each dimension is different (so the final regret bound would be worse). Another subtle point is the proof of Lemma 1 and Lemma 5. It is not clear with what the expectation in Eqn. 40, Eqn. 42, and Eqn. 43 is taken. It seems that the current definition of the expectation does not make sense for proving the regret bound, but I believe this problem can be fixed and the conclusion is still correct. Due to this caveats in the paper, I cannot ensure the correctness of the conclusions. I strongly recommend the authors to revise the statement and proofs in the paper.

**Questions:**

The tractable class 1 is similar to the factored MDPs, where the transition (and reward) of a large MDP can be factored into several subtransitions with regards to different part of the state. The optimal regret of factored MDPs only has square root dependency on the size of factored state space and action space. Can you provide some comparisons with tractable class 1 with factored MDPs?

Typo:
Section 2.3 Line 3: $\Phi_h \to \hat{\Phi}_h$

---

> ### Author Response · Authors · 2023-11-20
> **Response to Reviewer yDo3 (part 1)**
>
> We thank the reviewer for providing the helpful review! We have addressed the reviewer’s helpful comments and modified the paper accordingly. Please note that in the revised paper, we highlighted our changes by blue-colored texts.
>
> **Q1:** The exact setting studied in the paper seems to be inconsistent with the motivation of introducing OSI in POMDPs. As in previous work, the OSI in the hindsight is common some applications, such as data center scheduling, where the state is revealed after the action is take (so the effect of the action can be observed), However, the query model in this paper allows the agent to query part of the state before the action is taken. Even the motivating examples (e.g., the autonomous driving) seems to advocate the former setup, which reveals the state after taking actions. Although the conclusions of the paper may still hold in the case that the query happens after taking actions, it is appreciated to discuss about the model that is better aligned with the real world.
>
> **A1:** We clarify that our OSI setting shares in common with the previous work on the hindsight information setting in the aspect that some state information is available. However, they are different in the way when the state information is revealed. Particularly, in our problem setting, our process occurs in the order of “query the OSI”, “OSI is revealed”, and “action is taken”.
> Moreover, please note that our two motivating examples indeed serve to justify our setting, i.e., our process occurs in the order of “query the OSI”, “OSI is revealed”, and “action is taken”. Specifically, (i) in autonomous delivery systems, the robot first “queries condition of several paths”, then “observes the conditions of the queried paths (i.e., the partial OSI)”, and finally “takes an action to choose one path to follow (i.e., takes an action)”; (ii) in cognitive multiple access control protocols (the well established wireless communication standards), the agent first “probes a number of channels”, then “observes the conditions of those channels (i.e., the partial OSI)”, and finally “chooses one channel to transfer the data (i.e., takes an action)”.
>
> **Q2:** Some of the claims and proofs in the paper are not clear enough, leaving potential risks for the soundness of the paper. In the definition of tractable class 1, the transition kernel can be factored as the multiplication of $d$ different sub-transition according to each sub-state. Algorithm 3 does not clearly show how to estimate those $d$ sub-transitions, instead it looks like the sub-transitions are the same for each $i \in [d]$, which enables us to estimate this transition using all the adjacent revealed substates within one episode no matter whether the adjacent steps choose the same dimension of sub-state or not. This is not achievable when the subtransition for each dimension is different (so the final regret bound would be worse).
>
> **A2:** We clarify that we do not require the sub-transitions to be the same for each $i \in [d]$ in the paper. Specifically, in the definition of tractable class 1 in Sec. 4, we do not assume the sub-states have the same transition kernels. We also clarify that Algorithm 3 estimates possibly different sub-transitions for different $i \in [d]$. To make these clear in the paper, in the revision, we have changed $\mathbb P_h^k$ in Eqn. (37) to $\mathbb P_h^k(\phi_{\hat{i}}(s') | \phi_{\hat{i}}(s),a)$, which clearly captures the estimates of sub-transitions for each $i \in [d]$ in Algorithm 3.
>
> **Q3:** Another subtle point is the proof of Lemma 1 and Lemma 5. It is not clear with what the expectation in Eqn. 40, Eqn. 42, and Eqn. 43 is taken. It seems that the current definition of the expectation does not make sense for proving the regret bound, but I believe this problem can be fixed and the conclusion is still correct. Due to these caveats in the paper, I cannot ensure the correctness of the conclusions. I strongly recommend the authors to revise the statement and proofs in the paper.
>
> **A3:** We clarify that the expectations in Eqn. (40), Eqn. (42) and Eqn. (43) are taken with respect to the randomness of the algorithm, including the randomness of the query policy and action policy, and the state transition. Many thanks for the suggestion. We have added this explanation in the revision. Moreover, we have revised the statements and proofs in the paper. Please feel free to check.

---

> ### Author Response · Authors · 2023-11-20
> **Response to Reviewer yDo3 (part 2)**
>
> **Q4:** The tractable class 1 is similar to the factored MDPs, where the transition (and reward) of a large MDP can be factored into several sub-transitions with regards to different parts of the state. The optimal regret of factored MDPs only has square root dependency on the size of factored state space and action space. Can you provide some comparisons with tractable class 1 with factored MDPs?
>
> **A4:** Thanks for the question. The key difference between the tractable class 1 and factored MDPs is that in the tractable class 1, the agent needs to actively query only one sub-state, and then observe the information of only this queried sub-state. Hence the problem is still **POMDP**. In contrast, in factored MDPs, the full state is observed, and hence the problem is **MDP**. This is also the main reason that the same regret in factored MDPs cannot be obtained in the tractable class 1. We have added this explanation in Appendix E of the revised paper.
>
> **Q5:** Typo: Section 2.3 Line 3: $\Phi_h \rightarrow \hat{\Phi}_h$.
>
> **A5:** Thanks. In the revision, we have changed $\Phi_h$ to $\hat{\Phi}_h$ in Line 3 of Section 2.3.
>
> Finally, we thank the reviewer again for the helpful comments and suggestions for our work. If our response resolves your concerns to a satisfactory level, we kindly ask the reviewer to consider raising the rating of our work. Certainly, we are more than happy to address any further questions that you may have during the discussion period.

---

> > ### Comment · Reviewer_yDo3 · 2023-11-23
> >
> > Sorry for the delay. I would like to thank the reviewers for clarifying my concerns of the paper. I will increase my score accordingly. However, I still suggest to revise the proofs and presentations to make it more eligible, as other reviewers also mentioned this problem.

---

> > > ### Author Response · Authors · 2023-11-23
> > >
> > > We thank the reviewer for the response and increasing the score!

---

### Official Review · Reviewer_Z1AA · 2023-11-03

**Soundness:** 3 good
**Presentation:** 3 good
**Contribution:** 3 good
**Rating:** 6
**Confidence:** 3

**Summary:**

This paper studies the theoretical hardness and tractability of partially observable Markov decision processes (POMDPs) when only partial online state information (OSI) is available. The setting is different from the standard POMDP case, and also different from the recent work on POMDP with hindsight state information. The authors also provide motivating exampling to justify the proposed framework.

The authors then establish a lower bound that shows the exponential scaling of sample complexity is required to find an optimal policy solution for POMDPs without full OSI in general. However, they also identify two tractable sub-classes of POMDPs with partial OS. New algorithms are also proposed to solve the identified classes.

**Strengths:**

1 POMDP is a useful framework of the interactive decision-making problems, but is also intractable in general. Therefore, identifying interesting tractable sub-class of POMDP with interesting and reasonable structures is an important problem in theoretical RL study. The topic itself is thus very relevant to the community of neurips. Meanwhile, the proposed framework introduces partial side information before the agent makes decision, which is natural in practice and supported by the practical applications, and also is a great complement to existing works.

2 I find that the story is complete: the authors start with the proposed frameworks and its motivating examples, and also the connections with existing frameworks; and then a pessimistic lower bound to motivate further structural assumptions; finally, two algorithms are proposed to solve the identified tractable problems.

3 The proposed algorithm 1 is distinct from the recent popular OMLE/MOPS algorithms, which are popular since GOLF and Bilin-ucb. Instead, the algorithms are more related to the classic algorithms that are crafted to exploit the observable operator structure, but with distinct new ideas to handle the partial side information. To the best my knowledge, the algorithmic designs and some of the analysis techniques are new.

Overall, I feel that the authors have presented a reasonable framework of tractable POMDP problems, with a complete story, thus hitting the bar of acceptance.

**Weaknesses:**

1 The authors mention that the proposed framework can be placed under the general decision-making framework studied in [1,2]. I am wondering whether the identified problems can also be proved to be a sub-class of the tractable problems identified in these works. For instance, do the two classes of problems admit a low dec?

2 I think it would be better if you could explicitly instantiate the motivating examples in section 2 with the proposed framework (e.g. explicitly write down the state mapping with the physical quantities). Also, some of the superscripts are missing in the main paper (e.g. in section 2.3). It is also suggested to provide a notation table for the readers to improve readability.


[1] The statistical complexity of interactive decision making
[2] Unified algorithms for rl with decision-estimation coefficients: No-regret, pac, and reward-free learning

**Questions:**

see weakness

---

> ### Author Response · Authors · 2023-11-20
> **Response to Reviewer Z1AA**
>
> We thank the reviewer for providing the helpful review! We have addressed the reviewer’s helpful comments and modified the paper accordingly. Please note that in the revised paper, we highlighted our changes by blue-colored texts.
>
> **Q1:** The authors mention that the proposed framework can be placed under the general decision-making framework studied in [1,2]. I am wondering whether the identified problems can also be proved to be a sub-class of the tractable problems identified in these works. For instance, do the two classes of problems admit a low DEC?
>
> [1] The statistical complexity of interactive decision making.
>
> [2] Unified algorithms for RL with decision-estimation coefficients: No-regret, PAC, and reward-free learning.
>
> **A1:** Thanks for the question. Although the proposed framework can be placed under the general decision-making framework, bounding the DEC seems to be nontrivial. This is mainly because of the complicated underlying structure of these two classes, e.g., the two policies (query policy and action policy) are interleaved by the observed OSI, and the final reward depends on both the query and the action. Due to such a special event sequence and the complex dependence structure, we have not found a way to see if they admit low DECs. But these are indeed interesting problems to explore in the future.
>
> **Q2:** I think it would be better if you could explicitly instantiate the motivating examples in section 2 with the proposed framework (e.g. explicitly write down the state mapping with the physical quantities).
>
> **A2:** Great suggestion! Yes, we can explicitly instantiate the motivating examples in Sec. 2. Specifically, (i) in autonomous delivery systems, the robot agent first queries and observes the conditions of several paths, e.g., traffic intensity (i.e., the partial OSI). Then, the agent takes an action to choose one path to follow (i.e., the action that will incur a reward); (ii) in cognitive multiple access control protocols, the agent first probes the states of a number of channels, e.g., if the channel is in use or idle. After this query, the states of the sensed channels will be observed, i.e., the partial OSI. Then, the agent chooses one channel to transmit the data (i.e., the action that will incur a reward). We have added these in Sec. 2.
>
> **Q3:** Some of the superscripts are missing in the main paper (e.g. in section 2.3). It is also suggested to provide a notation table for the readers to improve readability.
>
> **A3:** Thanks. We have updated the notations in Sec. 2.3 and added a notation table, i.e., Table 1, at the beginning of the Appendix section in the revision.
>
> Finally, we thank the reviewer again for the helpful comments and suggestions for our work. If our response resolves your concerns to a satisfactory level, we kindly ask the reviewer to consider raising the rating of our work. Certainly, we are more than happy to address any further questions that you may have during the discussion period.

---

### Author Response · Authors · 2023-11-22

Dear reviewers,

Since the end of the discussion period is nearing, we were wondering whether you have further questions. We would like to have the opportunity to discuss and resolve potential further inquiries.

Thanks in advance,

The authors

---

### Meta-Review · Area_Chair_nkSx · 2023-12-06

**Metareview:**

This paper studies the problem of when partial state information can aid sample-efficient learning in Partially Observable Markov Decision Processes (POMDPs), and proposes a specific model coined POMDP with Partial OSI to study this. The main result are the identification of two tractable subclasses of Partial OSI models that enjoy polynomial sample complexities, via different algorithms. This is complemented with exponential lower bounds for arbitrary POMDPs in the worst case.

The reviewers generally agree that the proposed Partial OSI model makes progress towards an important problem, and the two proposed tractable subclasses as well as the algorithms are natural. However, as pointed out by and after discussing with the reviewers, the paper still contains issues in the formulations and some of the results, including (1) the tractable class 1, where it is unclear *why* a fixed query distribution which does not depend on $h$ or the previous interaction history, as outputted in Algorithm 1, suffices to make a near-optimal policy, as pointed out by a reviewer. If that's the case, then it is worth further explanations or a separate short proof; (2) some inconsistencies in the setting itself, in particular whether partial OSI model can be cast as a POMDP can be solved by (efficient) reductions or not, and that class 2 admits an extra observation $\tilde{o}_h$ while class 1 does not.

I encourage the authors to do a thorough revision of the paper, in particular about the setting (to make the prelim, class 1, and class 2 consistent) as well as the algorithm/result for Class 1.

**Justification For Why Not Higher Score:**

There are concerns about the formulations and some of the results that are beyond simple fixes and are worth a further careful revision.

**Justification For Why Not Lower Score:**

N/A

---

### Decision · Program_Chairs · 2024-01-16

Reject